# AtC: Aggregate-then-Calibrate for Human-centered Assessment

**Zejun Xie**[1*], **Xintong Li**[2*], **Guang Wang**[3†], **Desheng Zhang**[1]
[1] Rutgers University    [2] Renmin University of China    [3] Florida State University
[1] {zx180, dz220}@cs.rutgers.edu    [2] lixintong@ruc.edu.cn    [3] guang.wang@fsu.edu

## Abstract

Human-centered assessment tasks, which are essential for systematic decision-making, rely heavily on human judgment and typically lack verifiable ground truth. Existing approaches face a dilemma: methods using only human judgments suffer from heterogeneous expertise and inconsistent rating scales, while methods using only model-generated scores must learn from imperfect proxies or incomplete features. We propose Aggregate-then-Calibrate (`AtC`), a two-stage framework that combines these complementary sources. Stage-1 aggregates heterogeneous comparative judgments into a consensus ranking $\hat{\pi}$ using a rank-aggregation model that accounts for annotator reliability. Stage-2 calibrates any predictive model's scores by an isotonic projection onto the order $\hat{\pi}$, enforcing ordinal consistency while preserving as much of the model's quantitative information as possible. Theoretically, we show: (1) modeling annotator heterogeneity yields strictly more efficient consensus estimation than homogeneity; (2) isotonic calibration enjoys risk bounds even when the consensus ranking is misspecified; and (3) `AtC` asymptotically outperforms model-only assessment. Across semi-synthetic and real-world datasets, `AtC` consistently improves accuracy and robustness over human-only or model-only assessments. Our results bridge judgment aggregation with model-free calibration, providing a principled recipe for human-centered assessment when ground truth is costly, scarce, or unverifiable.

## 1 Introduction

Human-centered assessment tasks are essential for systematic decision-making in domains where ground truth is costly, unobservable, or only available in the future. Consider two scenarios: (1) Delivery platforms seek to estimate worker workload for fair compensation, yet true energy expenditure, though measurable via wearables, is impractical to collect at scale, necessitating reliance on worker judgments (Ahmadabadi et al., 2024; Chatterjee et al., 2020; Ho & Vaughan, 2012; Xie et al., 2025b). (2) Conference committees must evaluate paper quality for acceptance decisions, yet the true quality only becomes observable years later through future impact, requiring committees to aggregate reviewer assessments in the present (Su, 2021; Dolati Neghabadi et al., 2019; Zhang et al., 2022; Tran et al., 2021). These tasks share key characteristics: they seek standardized assessments from human judgments rather than individual preference satisfaction (Chu & Ghahramani, 2005; Chau et al., 2022; Hu et al., 2022), and while ground truth exists conceptually, such as workload or future impact, it cannot be directly observed when decisions are needed (Alur et al., 2024; Xu et al., 2024).

Existing approaches tend to rely either solely on human judgments or solely on model predictions. Methods that use only annotator-provided comparative judgments, such as judgment aggregation algorithms (Chen et al., 2013; Zhao et al., 2018; Jin et al., 2020), can weight annotators by expertise but remain confounded by inconsistent rating scales. For example, a lenient expert and a strict novice might agree on an item's relative quality yet assign very different absolute scores (Ito & Kashima, 2024; Khurana et al., 2024). Conversely, methods that use only model-generated scores from proxy labels face a supervision crisis: the true target values may correlate with latent factors that are impractical to measure at scale, such as cognitive load in task difficulty estimation (Xu et al.,

---

*    Equal contribution and shared co-first authorship.
†    Corresponding author.

2016; Guo et al., 2023; Wang et al., 2023; Lai et al., 2022). This forces models to learn from noisy surrogates, propagating systematic biases into the assessments.

Intuitively, human judgments are easy to solicit but lack a universal scale, whereas model predictions are consistent by design but require supervised signals (Cowgill, 2018; Dawes et al., 1989). Our goal is to combine the strengths of both sources: use the scale from the model assessment and the structure from human judgments (Hemmer et al., 2025). A key **insight** is that people are typically more reliable at comparing items than at absolute scoring (also known as Weber–Fechner laws in psychophysics (Pardo-Vazquez et al., 2019)). We therefore extract only the ordinal information from the noisy human inputs, rather than trusting their raw rating values or aggregated scores.

We propose a two-stage framework called Aggregate-then-Calibrate (`AtC`) that integrates the complementary advantages of human judgments and model assessments. In the aggregation stage, `AtC` employs a heterogeneous rank aggregation approach (Jin et al., 2019; Chen et al., 2013; Kuhlman & Rundensteiner, 2020) to derive a consensus ranking of items from multiple annotators, explicitly accounting for each annotator's individual scale and reliability. In the calibration stage, `AtC` uses isotonic regression (Dykstra & Robertson, 1982; Fielding, 2018; Su, 2021) to adjust an initial predictive model's assessments so that they align with the consensus ranking. This two-step approach ensures that the final assessments are monotonic with respect to the human consensus judgments. By combining the ordinal structure (from human) with the scoring scale (from the model), we theoretically and empirically demonstrate that `AtC` leverages the best of both: the human judgments provide a reliable ranking backbone, and the model provides consistent scoring, resulting in assessments that outperform either approach alone.

Our contributions are summarized as follows:

- **Conceptual:** We formalize a specific class of human-centered assessment problems where judgment aggregation (not preference optimization) is central for systematic decision-making. `AtC` addresses these via a human-model complementarity: aggregated comparisons provide ordinal constraints, while models estimate metrically scaled scores. Moreover, it can be generalized to any predictive model and thus enables the use of any off-the-shelf predictive model without modification.

- **Theoretical:** We provide a comprehensive analysis for `AtC` from three perspectives. First, to the best of our knowledge, Theorem 3.3 offers the first proof that heterogeneous rank aggregation models are strictly more statistically efficient than homogeneous methods when annotator abilities vary, validating our approach for the initial aggregation stage. Second, the risk analysis in Theorem 3.5 makes novel contributions to isotonic regression theory beyond prior work (Zhang, 2002; Chatterjee et al., 2015; Bellec, 2018) by accounting for projection onto a random cone and managing biased effective noise, which are two necessary conditions for our problem setting. Finally, these components culminate in our optimality guarantee in Theorem 3.9, formally demonstrating the power of integrating human ordinal judgments with model-based scores.

- **Empirical:** `AtC` consistently outperforms human-only and model-only assessments across semi-synthetic and real-world datasets, demonstrating superior accuracy and robustness under varying levels of data degradation.

## 2 Aggregate-then-Calibrate Framework

### 2.1 Overview

We consider a human-centered assessment task for $n$ items using inputs from $m$ human annotators and a predictive model $p$. Let $\mathbf{s} \in \mathbb{R}^n$ denote the true (unobserved) latent scores of the $n$ items, representing their ideal ground-truth quality (e.g., the true proficiency of a student's answer or the true merit of a product). Since $\mathbf{s}$ cannot be directly observed, we collect two complementary sources of information to estimate it:

- **Human Judgments:** Annotators provide comparative assessments (e.g., pairwise preferences or ratings) for various subsets of the items. These judgments are aggregated to produce a consensus score vector, which we denote by $\widetilde{\mathbf{s}} \in \mathbb{R}^n$. The vector $\widetilde{\mathbf{s}}$ represents the aggregated consensus over item scores derived purely from the given human input. We model $\widetilde{\mathbf{s}}$ as a noisy observation of the true scores $\mathbf{s}$; specifically, we assume $\widetilde{\mathbf{s}} = \mathbf{s} + \widetilde{\boldsymbol{\epsilon}}$, where $\widetilde{\boldsymbol{\epsilon}} \sim \mathcal{N}(0, \widetilde{\sigma}^2 I_n)$. In practice, $\widetilde{\mathbf{s}}$ is not

observed directly; instead, we estimate this consensus via the Stage-1 procedure. Let $\mathbf{s}^* \in \mathbb{R}^n$ denote the Stage-1 estimate of $\widetilde{\mathbf{s}}$ (as described in Stage-1 below). Let $\widehat{\pi}$ be the consensus ranking of the $n$ items, obtained by sorting the entries of $\mathbf{s}^*$ from lowest to highest. This ranking $\widehat{\pi}$ reflects the collective ordering of items according to the annotators' comparative judgments.

- **Model Assessments:** A predictive model $p$ (e.g., a machine learning model analyzing item features) produces an initial score vector $\mathbf{s}_p \in \mathbb{R}^n$ for the items. We do not assume that $\mathbf{s}_p$ is perfectly calibrated or aligned with the human judgments; indeed, the model may have been trained on proxy labels or may even be unsupervised due to the lack of ground truth. We model $\mathbf{s}_p$ as a biased observation of the true scores, subject to an unknown systematic offset: specifically, $\mathbf{s}_p = \mathbf{s} + \boldsymbol{\nu}$, where $\boldsymbol{\nu} \in \mathbb{R}^n$ is an unknown error (or bias) vector. Given the relationship $\widetilde{\mathbf{s}} = \mathbf{s} + \widetilde{\boldsymbol{\epsilon}}$ from above, we can relate the model scores to the consensus by writing $\mathbf{s}_p = \widetilde{\mathbf{s}} - \widetilde{\boldsymbol{\epsilon}} + \boldsymbol{\nu}$. For simplicity, we assume that the model's noise is independent of the annotators' noise (i.e., $\boldsymbol{\nu}$ is independent of $\widetilde{\boldsymbol{\epsilon}}$).

Our goal is to produce a final calibrated score vector $\widehat{\mathbf{s}} \in \mathbb{R}^n$ for the $n$ items. The calibrated scores $\widehat{\mathbf{s}}$ should (i) respect the consensus ordering $\widehat{\pi}$ (to maintain agreement with collective human judgment), and (ii) remain as close as possible to the model's original scores $\mathbf{s}_p$ (to preserve the model's useful quantitative information). We formalize this goal by defining the AtC estimator as the projection of $\mathbf{s}_p$ onto the set of score vectors that are consistent with $\widehat{\pi}$. Let $\widehat{\mathcal{M}} := \{\mathbf{y} \in \mathbb{R}^n : y_{\widehat{\pi}(1)} \leq y_{\widehat{\pi}(2)} \leq \cdots \leq y_{\widehat{\pi}(n)}\}$ be the set of all length-$n$ vectors that are nondecreasing according to the ranking $\widehat{\pi}$. The AtC estimate is then given by:

$$\widehat{\mathbf{s}} \;=\; \Pi_{\widehat{\mathcal{M}}}(\mathbf{s}_p) \;=\; \arg\min_{\mathbf{y} \in \widehat{\mathcal{M}}} \|\mathbf{y} - \mathbf{s}_p\|_2^2, \tag{2.1}$$

where $\Pi_{\widehat{\mathcal{M}}}(\mathbf{s}_p)$ denotes the Euclidean projection of $\mathbf{s}_p$ onto the constraint set $\widehat{\mathcal{M}}$. By construction, $\widehat{\mathbf{s}}$ is the closest vector to $\mathbf{s}_p$ (in $\ell_2$-distance) that does not violate the consensus ordering. In other words, $\widehat{\mathbf{s}}$ is obtained by an isotonic regression of the model's scores onto the ordering $\widehat{\pi}$. Figure 1 illustrates the roles of $\widetilde{\mathbf{s}}$, $\mathbf{s}_p$, and $\widehat{\mathbf{s}}$ in AtC, with pseudocode provided in Appendix A.

## 2.2 STAGE-1: JUDGMENT AGGREGATION UNDER HETEROGENEOUS THURSTONE MODEL

In Stage-1, we aggregate the human judgments into a single consensus ranking. We formulate this rank aggregation problem as finding a latent score vector (the consensus) that best explains the annotators' comparative judgments. Each annotator $u \in \{1, \ldots, m\}$ provides pairwise preferences over certain item pairs (for example, $i \succ j$ indicates annotator $u$ prefers item $i$ over item $j$). A key design choice is how to handle observed inconsistencies when different annotators judge the same items. When couriers assess delivery route difficulty, their judgments differ—but this reflects heterogeneity in expertise rather than fundamental subjective preferences. Rather than filtering novice opinions, our approach learns each annotator's reliability $\gamma_u$ from data and applies optimal weighting to produce consensus closer to the latent target.

**Preliminary: Heterogeneous Thurstone Model (HTM).** We assume these preferences follow a Thurstone choice model, extended to allow annotator-specific noise levels—this is HTM (Jin et al., 2020). In particular, given any two items $i$ and $j$ that annotator $u$ compares, the probability that $u$ prefers $i$ to $j$ is modeled as: $\Pr\{u : i \succ j\} = F(\gamma_u(\mathbf{s}_i - \mathbf{s}_j))$, where $F$ is a fixed symmetric cumulative distribution function (e.g., the standard normal CDF in Thurstone's original formulation, or a logistic CDF in a Bradley–Terry model), and $\gamma_u > 0$ is annotator $u$'s precision (or consistency) parameter. A larger $\gamma_u$ means annotator $u$ is less noisy (more reliable) in their pairwise comparisons, whereas a smaller $\gamma_u$ implies higher variance (lower consistency) for that annotator. To infer the consensus scores from the observed comparisons, maximum-likelihood estimation (MLE) is performed under the HTM.

Let $\mathcal{D}_u$ denote the set of observed pairwise comparisons from annotator $u$, where $i \succ j \in \mathcal{D}_u$ means $u$ preferred item $i$ over $j$. The log-likelihood of all annotators' data, given parameters $(\mathbf{s}, \gamma)$, is: $\ell(\mathbf{s}, \gamma) = \sum_{u=1}^m \sum_{i \succ j \in \mathcal{D}_u} \log F(\gamma_u(\mathbf{s}_i - \mathbf{s}_j))$, with appropriate constraints on $(\mathbf{s}, \gamma)$ for identifiability (e.g., one can fix $\frac{1}{n}\sum_{i=1}^n \mathbf{s}_i = 0$ or constrain $\frac{1}{m}\sum_{u=1}^m \gamma_u = 1$). The consensus score estimates $\mathbf{s}^* = (\mathbf{s}_1^*, \ldots, \mathbf{s}_n^*)$ and the annotator reliabilities $\gamma$ are obtained by maximizing this log-likelihood: $\max_{\mathbf{s}, \gamma} \ell(\mathbf{s}, \gamma)$, i.e. by finding the parameters that best explain the observed comparisons. This optimization can be carried out with iterative methods; for example, one can

alternate between updating the score estimates $\mathbf{s}$ and the annotator precisions $\gamma$ until convergence (each update can use gradient ascent on $\ell$ or other numerical routines).

The result of Stage-1 is an estimated consensus score vector $\mathbf{s}^*$. Sorting the components of $\mathbf{s}^*$ from lowest to highest yields the consensus ranking $\widehat{\pi}$. In other words, $\widehat{\pi}$ is the permutation of item indices such that $\mathbf{s}^*_{\widehat{\pi}(1)} \le \mathbf{s}^*_{\widehat{\pi}(2)} \le \cdots \le \mathbf{s}^*_{\widehat{\pi}(n)}$. This Stage-1 outcome $\widehat{\pi}$ reconciles the varying, potentially inconsistent human judgments into a single ranked list of items.

## 2.3 STAGE-2: ASSESSMENT CALIBRATION VIA ISOTONIC REGRESSION

In Stage-2, we leverage the consensus ranking $\widehat{\pi}$ from Stage-1 to calibrate the model's raw scores $\mathbf{s}_p$. The key idea is to adjust the model's scores so that the final outputs $\widehat{\mathbf{s}}$ respect the consensus ordering $\widehat{\pi}$ while remaining as faithful as possible to $\mathbf{s}_p$. We use the term calibration here to mean rescaling or aligning the model-generated scores with the human-derived ranking constraints. This notion of calibration differs from its conventional meaning in probability estimation or psychology: instead of aligning predicted probabilities with observed frequencies, here we enforce an ordinal consistency between $\mathbf{s}_p$ and the consensus ranking. In practice, we implement this calibration by projecting $\mathbf{s}_p$ onto the monotonicity constraint set defined by $\widehat{\pi}$, as formalized in Eq. (2.1).

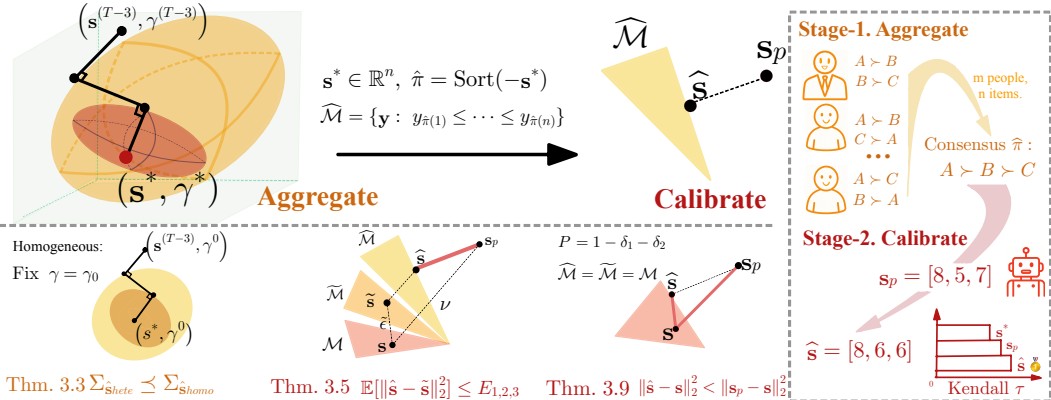

Figure 1: AtC Framework and Its Theoretical Guarantees.
Stage-1 aggregates human judgments into a ranking $\widehat{\pi}$; Stage-2 calibrates model scores $\mathbf{s}_p$ via isotonic projection onto $\mathcal{M}$ to obtain $\widehat{\mathbf{s}}$. Theorems 3.3, 3.5, and 3.9 provide efficiency, risk, and superiority guarantees.

**Preliminary: Isotonic regression.** As shown in Eq. (2.1), the calibrated output can be obtained by solving an isotonic regression problem: we seek the adjusted score vector $\widehat{\mathbf{s}}$ in $\widehat{\mathcal{M}}$ that minimizes the squared error to $\mathbf{s}_p$. Equivalently, $\widehat{\mathbf{s}}$ is the solution to $\min_{\mathbf{x} \in \widehat{\mathcal{M}}} \sum_{i=1}^n (x_i - \mathbf{s}_{p,i})^2$, which is exactly the projection defined in Eq. (2.1). This optimization finds the closest score vector to $\mathbf{s}_p$ that does not violate the consensus ranking order. The solution to this isotonic regression is efficiently computed via the Pool-Adjacent-Violators (PAV) algorithm (De Leeuw et al., 2010). PAV iteratively scans through the scores sorted by $\widehat{\pi}$ to detect any adjacent pair that is out of order (i.e., violates the required non-decreasing condition), and fixes such violations by averaging the offending scores. This process repeats until no more violations remain. The result is the calibrated score vector $\widehat{\mathbf{s}}$, which by construction satisfies $\widehat{\mathbf{s}}_{\widehat{\pi}(1)} \le \widehat{\mathbf{s}}_{\widehat{\pi}(2)} \le \cdots \le \widehat{\mathbf{s}}_{\widehat{\pi}(n)}$ (i.e., it respects the consensus ranking).

Notably, if the model's score vector $\mathbf{s}_p$ already happens to be perfectly consistent with $\widehat{\pi}$, the isotonic regression will leave it unchanged. Otherwise, the model's scores are adjusted in a minimally invasive way (with respect to squared-error) to remove any ranking inconsistencies. The calibrated scores $\widehat{\mathbf{s}}$ can be interpreted as a revised set of model-generated scores that combine the model's quantitative estimates with the human-derived ordinal constraints.

## 3 MAIN THEORETICAL RESULTS

As illustrated in Figure 1, AtC's intuitive design offers strong theoretical guarantees for human-centered assessment across three perspectives: heterogeneity, robustness, and optimality. In this section, we present the main results, with related work discussions in Section 5 and detailed assumptions and proofs in Appendix B, C, and D.

## 3.1 Heterogeneous Efficiency Guarantee

We now demonstrate the efficiency advantage of the HTM estimator over an estimator derived from a misspecified homogeneous model. When annotators exhibit varying levels of reliability, a correctly specified HTM is expected to yield score estimates with lower asymptotic variance compared to a homogeneous model that incorrectly assumes uniform annotator reliability. The following lemmas formalize the asymptotic covariance structures of the two estimators.

**Lemma 3.1.** Under regularity conditions in (White, 1982), the score vector estimator $\widehat{\mathbf{s}}_{hete}$ obtained from the MLE of the HTM has an asymptotic covariance matrix given by:

$$\Sigma_{\widehat{\mathbf{s}}_{hete}} = \frac{1}{N}\left(I_{ss} - I_{s\gamma}\,I_{\gamma\gamma}^{-1}I_{\gamma s}\right)^{+}, \tag{3.1}$$

where $N = \sum_{u=1}^{m} k_u$ is the total number of observations. The matrices $I_{ss}$, $I_{\gamma\gamma}$ and $I_{s\gamma}$ are the Fisher information sub-matrices for $\mathbf{s}$ and $\boldsymbol{\gamma}$. $(I_{ss} - I_{s\gamma}\,I_{\gamma\gamma}^{-1}I_{\gamma s})^{+}$ is the Schur complement of the full Fisher information with respect to $\mathbf{s}$, representing the effective Fisher information for $\mathbf{s}$ after accounting for the estimation of $\boldsymbol{\gamma}$. The pseudoinverse addresses the identifiability constraint $\mathbf{1}^{\top}\mathbf{s} = 0$.

**Lemma 3.2.** Consider a homogeneous Thurstone model assuming a known constant annotator accuracy $\gamma_0$. The quasi-MLE $\widehat{\mathbf{s}}_{homo}$, derived from this potentially misspecified model, has an asymptotic covariance matrix given by the classic sandwich form:

$$\Sigma_{\widehat{\mathbf{s}}_{homo}} = \frac{1}{N}\left(A_{homo}(\mathbf{s}_*)^{+}B_{homo}(\mathbf{s}_*)A_{homo}(\mathbf{s}_*)^{+}\right), \tag{3.2}$$

where $\mathbf{s}_*$ is the probability limit of $\widehat{\mathbf{s}}_{homo}$. The matrix $A_{homo}(\mathbf{s}_*)$ is the negative of the expected Hessian and $B_{homo}(\mathbf{s}_*)$ is the expected outer product of the scores from the misspecified model's log-likelihood, with expectations taken under the true heterogeneous data-generating process. The pseudoinverse also addresses identifiability.

Lemmas 3.1 and 3.2 allow us to formally compare the asymptotic efficiency of the two estimation strategies. The following theorem states that the estimator from the correctly specified HTM is asymptotically superior. The proof is given in Appendix B.4 and B.5.

**Theorem 3.3. (Heterogeneous Efficiency Guarantee).** Let $\Sigma_{\widehat{\mathbf{s}}_{hete}}$ and $\Sigma_{\widehat{\mathbf{s}}_{homo}}$ be the asymptotic covariance matrices of the score estimators as defined in Lemmas 3.1 and 3.2. Then,

$$\Sigma_{\widehat{\mathbf{s}}_{hete}} \preceq \Sigma_{\widehat{\mathbf{s}}_{homo}}, \tag{3.3}$$

where $\preceq$ denotes the Loewner order. This implies that the matrix $\Sigma_{\widehat{\mathbf{s}}_{homo}} - \Sigma_{\widehat{\mathbf{s}}_{hete}}$ is positive semi-definite. Furthermore, if the true annotator accuracies $\boldsymbol{\gamma}^*$ are not all equal (i.e., genuine heterogeneity exists), then this inequality is strict within the identifiable subspace, meaning $\Sigma_{\widehat{\mathbf{s}}_{homo}} - \Sigma_{\widehat{\mathbf{s}}_{hete}}$ is positive definite on this subspace.

Theorem 3.3 demonstrates that when annotator heterogeneity is present, the correctly specified HTM yields a strictly more efficient estimator of the true scores than the misspecified homogeneous model. Explicitly modeling annotator-specific differences provides a demonstrable advantage in estimation precision. The proof is in Appendix B.6.

## 3.2 Robustness Guarantee under Model Misspecification

We next analyze how errors from the Stage-1 rank aggregation affect the final calibrated output in Stage-2: an upper bound on the expected squared error $\mathbb{E}[\|\widehat{\mathbf{s}} - \widetilde{\mathbf{s}}\|_2^2]$. This bound quantifies the performance of the two-stage procedure, where $\widehat{\mathbf{s}}$ is the projection of model predictions $\mathbf{s}_p$ onto the human-consensus-derived cone $\widehat{\mathcal{M}}$, aiming to estimate the subjective optimal scores $\widetilde{\mathbf{s}}$. Our analysis accounts for the misspecification of $\widehat{\mathcal{M}}$ and the systematic bias $\boldsymbol{\nu}$ in $\mathbf{s}_p$ relative to $\widetilde{\mathbf{s}}$.

We first establish a sharp oracle inequality for least squares estimators, leveraging a lemma adapted from (Bellec, 2018, Proposition 2.1). This result bounds the estimation error based on the effective noise properties and the constraint set's geometry.

**Lemma 3.4** (Oracle Inequality with Biased Effective Noise). Let $\widetilde{\mathbf{s}} \in \mathbb{R}^n$ be the target score vector (itself random, with $\mathbb{E}[\widetilde{\mathbf{s}}] = \mathbf{s}$). Let $\widehat{\mathcal{M}} \subset \mathbb{R}^n$ be a closed convex set. The observations for projection

are $\mathbf{s}_p$, and the effective noise relating $\mathbf{s}_p$ to $\widetilde{\mathbf{s}}$ is $\boldsymbol{\xi} = \mathbf{s}_p - \widetilde{\mathbf{s}} \sim \mathcal{N}(\boldsymbol{\nu}, \widetilde{\sigma}^2 I_n)$. The least squares estimator is $\widehat{\mathbf{s}} = \Pi_{\widehat{\mathcal{M}}}(\mathbf{s}_p)$. For any $\boldsymbol{u} \in \widehat{\mathcal{M}}$, the squared estimation error satisfies, almost surely:

$$\|\widehat{\mathbf{s}} - \widetilde{\mathbf{s}}\|_2^2 - \|\boldsymbol{u} - \widetilde{\mathbf{s}}\|_2^2 \leq \frac{1}{n} \left( \sup_{\boldsymbol{\theta} \in \mathcal{T}_{\widehat{\mathcal{M}},\boldsymbol{u}} : \|\boldsymbol{\theta}\|_2 \leq 1} \boldsymbol{\xi}^T \boldsymbol{\theta} \right)^2, \tag{3.4}$$

where $\mathcal{T}_{\widehat{\mathcal{M}},\boldsymbol{u}}$ is the tangent cone to $\widehat{\mathcal{M}}$ at $\boldsymbol{u}$.

Taking expectations and choosing $\boldsymbol{u} = \Pi_{\widehat{\mathcal{M}}}(\widetilde{\mathbf{s}})$ leads to a decomposition of the total expected risk into three principal components:

$$\mathbb{E}[\|\widehat{\mathbf{s}} - \widetilde{\mathbf{s}}\|_2^2] \leq \underbrace{\mathbb{E}_{\widehat{\mathcal{M}},\widetilde{\mathbf{s}}} \left[ \|\Pi_{\widehat{\mathcal{M}}}(\widetilde{\mathbf{s}}) - \widetilde{\mathbf{s}}\|_2^2 \right]}_{\text{Projection Error}:=\mathbb{E}_1} + \underbrace{\mathbb{E}_{\widehat{\mathcal{M}},\widetilde{\mathbf{s}}} \left[ \frac{2\widetilde{\sigma}^2}{n} \delta(\mathcal{T}_{\widehat{\mathcal{M}},\Pi_{\widehat{\mathcal{M}}}(\widetilde{\mathbf{s}})}) \right]}_{\text{Statistical Error}:=\mathbb{E}_2} + \underbrace{\mathbb{E}_{\widehat{\mathcal{M}},\widetilde{\mathbf{s}}} \left[ \frac{2}{n} \|\Pi_{\mathcal{T}_{\widehat{\mathcal{M}},\Pi_{\widehat{\mathcal{M}}}(\widetilde{\mathbf{s}})}}(\boldsymbol{\nu})\|_2^2 \right]}_{\text{Bias Error}:=\mathbb{E}_3}. \tag{3.5}$$

Term $\mathbb{E}_1$ reflects errors from projecting $\widetilde{\mathbf{s}}$ onto the potentially incorrect cone $\widehat{\mathcal{M}}$. Term $\mathbb{E}_2$ arises from the zero-mean component of $\boldsymbol{\xi}$, where $\delta(\cdot)$ is the statistical dimension of the tangent cone. Term $\mathbb{E}_3$ captures error due to the systematic bias $\boldsymbol{\nu}$ (details are in Appendix C.1). Further bounding these components yields our main robustness theorem.

**Theorem 3.5. (Robustness Guarantee).** The total expected squared error of $\widehat{\mathbf{s}}$ for $\widetilde{\mathbf{s}}$ is bounded by:

$$\mathbb{E}[\|\widehat{\mathbf{s}} - \widetilde{\mathbf{s}}\|_2^2] \leq \mathbb{E}_{\widetilde{\mathbf{s}}}[n\mathrm{Var}(\widetilde{\mathbf{s}})] \, \mathbb{E}\left[\mathrm{Inv}(\widehat{\pi}, \widetilde{\pi})\right] + \frac{2\widetilde{\sigma}^2(\ln(n) + \gamma_E + O(1/n))}{n} + \frac{2\|\boldsymbol{\nu}\|_2^2}{n}. \tag{3.6}$$

Here, $\mathbb{E}[\mathrm{Inv}(\widehat{\pi}, \widetilde{\pi})]$ is the expected ranking inversions from Stage-1. $\widetilde{\sigma}^2$ is the variance of $\widetilde{\mathbf{s}}$, $\boldsymbol{\nu}$ is the fixed systematic bias, and $\gamma_E$ is the Euler-Mascheroni constant.

The first term in the bound of Theorem 3.5, which accounts for errors due to Stage-1 ranking inaccuracies, is directly related to the probability of the estimated ranking $\widehat{\pi}$ differing from the subjective optimal ranking $\widetilde{\pi}$. This probability can be further bounded as follows:

**Corollary 3.6** (Bound on Ranking Misspecification Probability)**.** For any two items $j, k$ such that $\widetilde{\mathbf{s}}_j < \widetilde{\mathbf{s}}_k$, let $\Delta_{kj} = \widetilde{\mathbf{s}}_k - \widetilde{\mathbf{s}}_j$ be the true score gap. Let $\Sigma_{\widehat{\mathbf{s}}}$ denote the covariance matrix of the estimation error $\widehat{\mathbf{s}} - \widetilde{\mathbf{s}}$ in Lemma 3.1, and $\sigma_{X_{jk}}^2 = (\boldsymbol{e}_j - \boldsymbol{e}_k)^\top \Sigma_{\widehat{\mathbf{s}}} (\boldsymbol{e}_j - \boldsymbol{e}_k)$. The probability that the estimated ranking $\pi$ differs from the subjective optimal ranking $\widetilde{\pi}$ (i.e., $\widehat{\mathcal{M}} \neq \widetilde{\mathcal{M}}$) is bounded by:

$$\delta_1 := P(\widehat{\pi} \neq \widetilde{\pi}) \leq \sum_{j,k:\widetilde{\mathbf{s}}_j < \widetilde{\mathbf{s}}_k} \frac{\sigma_{X_{jk}}}{\Delta_{kj}\sqrt{2\pi}} \exp\left( -\frac{(\Delta_{kj})^2}{2\,\sigma_{X_{jk}}^2} \right). \tag{3.7}$$

To conclude, the bound highlights in Theorem 3.5 three key error sources: (i) Stage-1 ranking inaccuracies, quantified by the expected number of inversions $\mathbb{E}[\mathrm{Inv}(\widehat{\pi}, \widetilde{\pi})]$ and scaled by the target's variability, which rapidly diminishes with larger score gaps relative to first-stage noise; (ii) statistical uncertainty from the zero-mean effective noise, decaying around $(\ln n)/n$; (iii) systematic observation bias $\boldsymbol{\nu}$, decaying as $1/n$. Notably, even perfect first-stage ranking does not eliminate errors from $\widetilde{\sigma}^2$ and $\boldsymbol{\nu}$, while poor initial rankings can significantly inflate the total error, though mitigation of these factors can still lead to effective error reduction. The proof is in Appendix C.3.

### 3.3 OPTIMALITY GUARANTEE

Finally, we establish that the AtC yields a calibrated output that exceeds the uncalibrated model's output in accuracy with high probability. Intuitively, if the Stage-1 aggregation produces an almost-correct ordering of items and Stage-2 calibration correctly removes the bias, then the final result will be closer to the ground truth than using the original model predictions alone. Recall from Corollary 3.6 shows that $\delta_1$ decays exponentially in the gap-to-noise ratio $\frac{\sigma_{X_{jk}}}{\Delta_{kj}}$ for each pair. If every true gap is not too small compared to the associated noise, then $\delta_1$ becomes negligible. The following proposition shows that Stage-1 produces the correct order of items with high probability.

**Proposition 3.7.** Let $\Delta_{\min} = \min_{j<k} \Delta_{kj}$ and $\sigma_{\max} = \max_{j<k} \sigma_{X_{jk}}$. If $\frac{\sigma_{\max}}{\Delta_{\min}} \leq \varepsilon$ for some sufficiently small $\varepsilon > 0$ (equivalently, the Stage–1 sample size satisfies $mk \gg 1/(\Delta_{\min}\varepsilon)^2$), then $\delta_1 = o(1)$; i.e. the estimated ranking space $\widehat{\mathcal{M}}$ will equal the true ranking space $\mathcal{M}$.

At the same time, Stage-2 may still fail if the projection is badly perturbed by the noise $\widetilde{\epsilon} \sim \mathcal{N}(0, \widetilde{\sigma}^2 I_n)$. Let's define the event $B = \{\mathcal{M} = \widetilde{\mathcal{M}}\}$, then

**Lemma 3.8.**

$$\delta_2 := P(B^c) \leq \sum_{j,k:\, s_k > s_j} \frac{\widetilde{\sigma}}{\Delta_{kj}\sqrt{\pi}} \exp\left(-\frac{\Delta_{kj}^2}{4\widetilde{\sigma}^2}\right). \tag{3.8}$$

Lemma 3.8 shows that the chance of calibration error decays rapidly once the post–aggregation score gaps exceed the noise level $\sigma$.

Because the pairwise comparisons used in Stage-1 are independent of the Gaussian perturbations $\epsilon$ in Stage-2, the events in Corollary 3.6 and Lemma 3.8 are independent.

**Theorem 3.9. (Optimality Guarantee).** Let $\delta_1$ and $\delta_2$ be as in Corollary 3.6 and Lemma 3.8. Then, with probability at least $1 - \delta_1 - \delta_2$,

$$\left\|\widehat{\mathbf{s}} - \mathbf{s}\right\|_2^2 < \left\|\mathbf{s}_p - \mathbf{s}\right\|_2^2, \tag{3.9}$$

i.e. the calibrated output $\widehat{\mathbf{s}}$ is strictly closer to the ground truth $\mathbf{s}$ than the uncalibrated model prediction $\mathbf{s}_p$. As the Stage–1 sample size grows and $\Delta_{\min}$ dominates the noise, both $\delta_1$ and $\delta_2$ approach 0, so the probability of improvement approaches 1.

Theorem 3.9 confirms that, once Stage-1 has gathered enough pairwise comparisons to rank items reliably, the monotone projection in Stage-2 invariably drives the (possibly biased) model scores toward the ground truth. In effect, `AtC` protects against misspecification: even if both aggregated consensus and predictive model are imperfect, enforcing the human–derived rank guarantees a strictly better estimator with high probability.

## 4 EXPERIMENTS

We evaluate `AtC` on both semi-synthetic and real-world datasets, guided by 6 research questions:

- (**RQ1**) Does `AtC`'s output $\widehat{\mathbf{s}}$ outperform human-only assessment $\mathbf{s}^*$ and model-only assessment $\mathbf{s}_p$?
- (**RQ2**) Do heterogeneous rank aggregation methods outperform homogeneous ones, and is `AtC`'s design of using the ranking $\widehat{\pi}$ from $\mathbf{s}^*$ rather than directly using the $\mathbf{s}^*$ values correct?
- (**RQ3**) Is `AtC` robust to aggregation errors?
- (**RQ4**) Is `AtC` robust to model assessments ($\mathbf{s}_p$) errors?
- (**RQ5**) Does `AtC`'s assumption that rankings are more reliable than ratings hold true in practice?
- (**RQ6**) How well does `AtC` perform on a real-world task?

Table 1: Semi-synthetic results on Reading Level dataset.
**Bold** indicate the best performance among human-only assessment ($\mathbf{s}^*$), model-only assessment ($\mathbf{s}_p$), and `AtC` assessment ($\widehat{\mathbf{s}}$), respectively. The **red** denotes the optimal performance across all methods for the given metric.

| Stage-1 Method | Kendall $\tau\uparrow$ $\mathbf{s}^*/\mathbf{s}_p/\widehat{\mathbf{s}}$ | Wasserstein$\downarrow$ $\mathbf{s}^*/\mathbf{s}_p/\widehat{\mathbf{s}}$ | KS$\downarrow$ $\mathbf{s}^*/\mathbf{s}_p/\widehat{\mathbf{s}}$ | MSE$\downarrow$ $\mathbf{s}^*/\mathbf{s}_p/\widehat{\mathbf{s}}$ |
|---|---|---|---|---|
| HRA-G | 0.375 / 0.399 / **0.410** | 2.250 / 2.831 / **0.839** | 0.500 / 0.300 / **0.163** | 8.658 / 29.00 / **8.122** |
| HRA-E | 0.375 / 0.399 / **0.403** | 2.243 / 2.831 / **0.827** | 0.498 / 0.300 / **0.163** | 8.658 / 29.00 / **8.191** |
| HRA-N | 0.368 / 0.399 / **0.399** | 2.351 / 2.831 / **0.738** | 0.563 / 0.300 / **0.192** | 8.985 / 29.00 / **7.919** |
| CrowdBT | 0.354 / **0.399** / 0.399 | 2.150 / 2.831 / **0.843** | 0.455 / 0.300 / **0.269** | 8.301 / 29.00 / **7.555** |
| CrowdTCV | 0.339 / **0.399** / 0.372 | 2.272 / 2.831 / **1.015** | 0.506 / **0.300** / 0.312 | 8.689 / 29.00 / **7.809** |
| BTL | 0.340 / **0.399** / 0.373 | 2.186 / 2.831 / **0.894** | 0.461 / 0.300 / **0.300** | 7.764 / 29.00 / **8.097** |
| TCV | 0.338 / **0.399** / 0.373 | 2.343 / 2.831 / **0.914** | 0.547 / 0.300 / **0.300** | 8.943 / 29.00 / **8.030** |

**Datasets.** We consider two datasets in our experiments. **(1) Reading-level (semi-synthetic dataset)** (Chen et al., 2013) contains pairwise comparisons of text documents based on reading difficulty. 490 documents with known reading levels, 624 annotators providing 12,728 pairwise judgments. **(2) Dots-activity (real-world dataset)** (Kemmer et al., 2020), a benchmark for collective judgment aggregation. The task involves estimating the number of dots in images. The dataset contains two types judgments (ranking and rating) from 300 participants on 30 distinct images, which yield 8700 pairwise comparisons.

**Baselines and Metrics.** For the semi-synthetic dataset, we compare 7 different aggregation methods: 3 HRA variants based on different noise distribution assumptions (HRA-G/E/N), 2 heterogeneous methods (CrowdBT/TCV) (Chen et al., 2013), and 2 homogeneous baselines (BTL/TCV) (Bradley & Terry, 1952). For the real-world dataset, we additionally compare against 3 human-centered assessment methods: GPPL (Chu & Ghahramani, 2005), Rank-SVM (Joachims, 2002), and BARCW (Li et al., 2022). We evaluate using 4 metrics: Kendall $\tau$ measures ranking accuracy; Wasserstein distance and KS statistic quantify distributional alignment between predicted and ground-truth scores; MSE captures absolute prediction error. Detailed descriptions are provided in Appendix E.

**Semi-Synthetic Evaluation.** We first evaluate on semi-synthetic setting where ground-truth item scores are available. We simulate $m$ annotators who provide pairwise comparisons, apply various Stage-1 aggregation methods (w/ or w/o modeling annotator heterogeneity) to obtain a consensus score $\mathbf{s}^*$, and then train a predictive model to produce an initial model score $\mathbf{s}_p$ for each item (using a portion of the ground truth with added noise to emulate model error). We calibrate $\mathbf{s}_p$ via AtC to obtain the output $\widehat{\mathbf{s}}$.

In Table 1, we observe that $\widehat{\mathbf{s}}$ consistently outperforms both $\mathbf{s}^*$ and $\mathbf{s}_p$ across nearly all metrics and methods (the $\widehat{\mathbf{s}}$ entries are almost always **bolded** as best).[1] Figure 2 further illustrates this improvement: the $\widehat{\mathbf{s}}$ values align much more closely with the true distribution of $s$, whereas the uncalibrated $\mathbf{s}^*$ and $\mathbf{s}_p$ scores deviate significantly (same conclusion for all methods in Appendix E.4). To verify that this improvement stems from $\mathbf{s}_p$ rather than simple rescaling, we conducted experiments matching $\mathbf{s}^*$'s range to ground truth (Appendix E.5). In summary, the AtC score dominates the human-only and model-only assessment, confirming that combining the two sources produces more accurate judgments (answering **RQ1**).

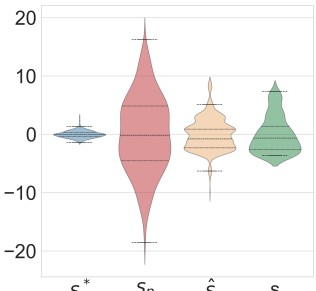

Figure 2: Score Distribution of HRA-E

We examine the effect of modeling annotator heterogeneity on the consensus ranking quality. In Table 1, the methods that account for annotator heterogeneity (HRA-G/E/N, and CrowdBT/TCV) produce better (or the **best**) calibrated scores $\widehat{\mathbf{s}}$ than the homogeneous models (BTL, TCV). Furthermore, we observe that for $\mathbf{s}^*$, the heterogeneous HRA models achieve more accurate rankings (higher Kendall $\tau$), despite performing worse on distance metrics compared to 4 baselines. Notably, after calibration with $\mathbf{s}_p$, the resulting $\widehat{\mathbf{s}}$ scores from heterogeneous models surpass 4 baselines across Wasserstein distance and KS statistic, confirming that our design choice to prioritize ranking information from $\mathbf{s}^*$ is indeed effective (answering **RQ2**).

Figure 3 shows AtC's robustness to ranking errors in $\mathbf{s}^*$. Our normalized results show that as we introduce pairwise judgment inversions, AtC degrades gradually until a critical threshold. With moderate noise (up to 400 inversions), Kendall $\tau$ decreases steadily while MSE increases, but AtC still produces meaningful outputs by leveraging model signals. However, beyond approximately 500 inversions, performance collapses completely—Kendall $\tau$ becomes negative, variance approaches zero, and the output distribution flattens (shown by declining block count). This synchronized degradation across all metrics confirms that while AtC can tolerate considerable noise in consensus rankings, extremely corrupted inputs will eventually render the calibration ineffective (answering **RQ3**).

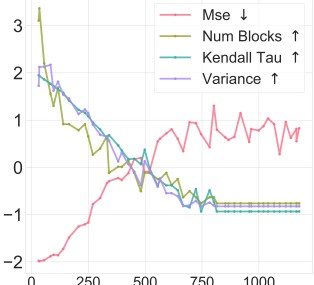

Figure 3: Robustness Analysis

---

[1]The Kendall's $\tau$ values for $\widehat{\mathbf{s}}$ and $\mathbf{s}^*$ differ due to tie-creation operation; see Appendix E.3 for details.

Table 2: Real-world results on Dots dataset.

| Stage-1 Method | Kendall $\tau\uparrow$ $\mathbf{s}^*$ / $\mathbf{s}_p$ / $\widehat{\mathbf{s}}$ | Wasserstein$\downarrow$ $\mathbf{s}^*$ / $\mathbf{s}_p$ / $\widehat{\mathbf{s}}$ | KL$\downarrow$ $\mathbf{s}^*$ / $\mathbf{s}_p$ / $\widehat{\mathbf{s}}$ | MSE$\downarrow$ $\mathbf{s}^*$ / $\mathbf{s}_p$ / $\widehat{\mathbf{s}}$ |
|---|---|---|---|---|
| AtC HRA-G | 0.917 / 0.923 / **0.940** | 6.97 / 2.53 / **2.53** | 123.13 / 0.881 / **0.860** | 65.08 / 11.75 / **9.61** |
| HRA-E | 0.922 / 0.922 / **0.943** | 6.98 / 2.53 / **2.53** | 123.16 / 0.881 / **0.861** | 65.16 / 11.75 / **9.59** |
| HRA-N | 0.917 / 0.923 / **0.934** | 7.05 / 2.53 / **2.53** | 125.50 / 0.881 / **0.859** | 66.44 / 11.75 / **9.75** |
| GPPL | – / – / 0.931 | – / – / 64.50 | – / – / 126.62 | – / – / 4220.36 |
| Rank-SVM | – / – / 0.923 | – / – / 61.20 | – / – / 133.70 | – / – / 3814.49 |
| BARCW | – / – / 0.940 | – / – / 64.62 | – / – / 24.09 | – / – / 4236.16 |

**Real-World Evaluation.** We evaluate the validity and robustness of the AtC framework on the Dots-activity dataset, a benchmark for counting dots in images. We assume models can only process corrupted images due to various constraints, while humans can still make reliable inferences based on experience, making this a valuable human-centered assessment problem. We introduce various image corruptions to simulate real-world scenarios with noisy or partially occluded data, then generate predictive scores $\mathbf{s}_p$ using OpenCV's contour detection (Suzuki & Abe, 1985) as an imperfect predictor. As illustrated in Figure 4, we apply 4 corruption types (Hendrycks & Dietterich, 2019): global Gaussian blur and three localized corruptions (blur, noise, and whiteout), with degradation intensity controlled by a hyperparameter to evaluate performance across different corruption levels.

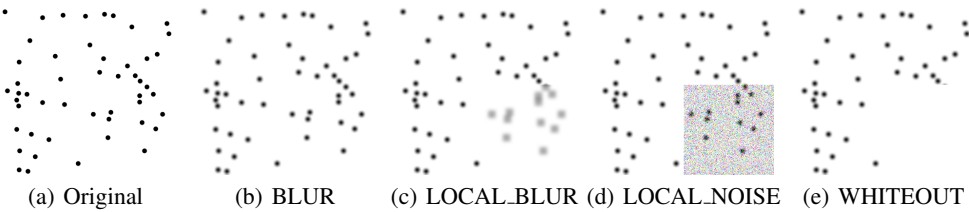

(a) Original      (b) BLUR      (c) LOCAL_BLUR    (d) LOCAL_NOISE    (e) WHITEOUT

Figure 4: Examples of image corruption types applied to the Dots-activity dataset.

Table 2 summarizes performance across AtC and 3 human-centered assessment baselines (GPPL, Rank-SVM, BARCW); comprehensive results including all Stage-1 methods appear in Appendix E.6. Figure 5 presents the core robustness results for HRA-E, with similar trends for other baseline methods shown in Appendix E.7. The radar charts intuitively visualize the degradation of Kendall's Tau correlation as damage intensity increases (clockwise from the top axis). Each chart compares the initial model ($\mathbf{s}_p$), the uncalibrated human consensus ($\mathbf{s}^*$), and the calibrated AtC outputs based on rankings ($\widehat{\mathbf{s}}$ (Rank)) and ratings ($\widehat{\mathbf{s}}$ (Rate)). Across the 4 types of corruption, our proposed primary method $\widehat{\mathbf{s}}$ (Rank) demonstrates remarkable resilience. While the performance of the raw objective model $\mathbf{s}_p$ degrades sharply with increasing noise, particularly under localized corruptions. This shows that by anchoring the model's scores to the stable structure provided by human consensus, AtC effectively mitigates the impact of noise on the predictor (answering **RQ4**). Furthermore, these results underscore the superiority of using ordinal (ranking) information for calibration. In every scenario, $\widehat{\mathbf{s}}$ (Rank) outperforms $\widehat{\mathbf{s}}$ (Rate), the alternative calibrated using aggregated cardinal ratings. This suggests that the consensus ranking ($\mathbf{s}^*$) provides a more robust calibration framework than aggregated numerical estimates, which are susceptible to individual biases and inconsistent scales (answering **RQ5**). In summary, the AtC framework demonstrates strong real-world performance by maintaining high accuracy even under severe input degradation (answering **RQ6**).

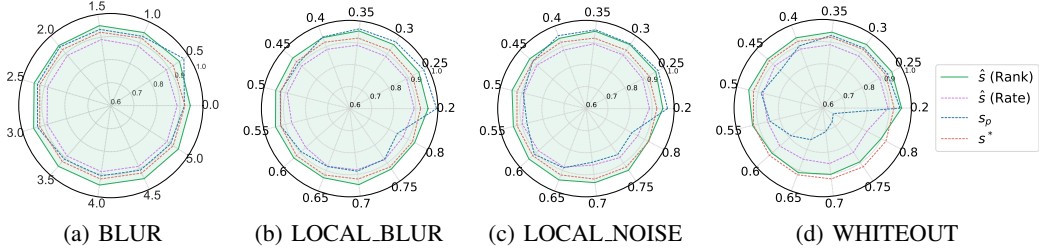

(a) BLUR      (b) LOCAL_BLUR      (c) LOCAL_NOISE      (d) WHITEOUT

Figure 5: HRA-E's performance (Kendall $\tau\uparrow$) under 4 corruption types as damage intensity increases.

## 5 ADDITIONAL RELATED WORK

**Judgement Aggregation and Rank Aggregation.** Inferring a consensus from multiple noisy annotators is a common problem in crowdsourcing and psychometrics (Murphy & Martin, 2003; Liu et al., 2019; Hu et al., 2022). Classic models, like Dawid & Skene's (Dawid & Skene, 1979) model estimate a latent true label or score while capturing annotator error rates, with extensions incorporating bias and expertise. Ranking models, such as Bradley–Terry or Thurstone–Mosteller variants, account for heterogeneous annotator reliability (Pan et al., 2022). Our Stage-1 algorithm accounts for rater heterogeneity to estimate the consensus ranking $\mathbf{s}^*$, which is supported by Theorem 3.3 and experiments, outperforms homogeneous annotator assumptions. Unlike prior methods focused solely on consensus, $\mathtt{AtC}$ uses it as an ordinal scaffold for predictive model calibration, enhancing downstream efficiency with formal guarantees on predictive performance.

**Model Calibration and Isotonic Regression.** Calibrating model outputs to better align with empirical outcomes is a common practice in machine learning, especially for probabilistic predictions. Isotonic regression in particular is a non-parametric technique that enforces output monotonicity and has been used to calibrate scores to observed binary labels or probabilities (Luss et al., 2012; Baan et al., 2022). In $\mathtt{AtC}$, we novelly apply isotonic regression to calibrate a continuous model predictions to a rank-based target derived from human judgments. This usage differs from standard calibration, as we are mapping model scores to an ordering rather than absolute ground-truth values. Prior work on isotonic regression and calibration (Su, 2021; Chatterjee et al., 2015; Bellec, 2018) has not, to our knowledge, combined isotonic regression with a crowd-sourced ranking constraint. Theorem 3.5 provides calibration error analysis with imperfect rankings, demonstrating $\mathtt{AtC}$'s ability to effectively calibrate using inferred consensus rather than fixed ground-truth labels. Unlike (Bellec, 2018), our innovation lies in modeling subjective optimum $\widetilde{\mathbf{s}}$ as a random variable with allowable non-zero bias $\boldsymbol{\nu}$, enabling monotonic calibration when only relative judgments are available. We choose $L_2$ projection instead of $L_1$ or $L_\infty$ alternatives because it corresponds to maximum likelihood estimation under Gaussian noise and ensures unique solutions due to strict convexity (Stout, 2023).

**Human-AI Complementarity.** Our work is also related to methods that integrate human and machine decision-making to improve prediction quality. Such approaches seek to optimally fuse human labels and algorithmic learning (Raykar et al., 2010; Yang et al., 2016; Li et al., 2018; Steyvers et al., 2022). Notably, calibration via human judgment has emerged as a practical entry point for real-world assessment, such as logistics area difficulty assessment (Xie et al., 2025b). $\mathtt{AtC}$ extends this line of work in two ways: it generalizes the framework beyond domain-specific settings, and provides an optimality theorem (Theorem 3.9) that explains when and why the two-stage pipeline outperforms either component alone, which gap is common across prior human-in-the-loop methods (Xu et al., 2024; Alur et al., 2024). Unlike (Pan et al., 2022; Li et al., 2022), which impose linear model constraints, $\mathtt{AtC}$ places no restrictions on the aggregation method or the predictive model, making the design modular and broadly applicable.

## 6 DISCUSSION AND CONCLUSION

We introduced $\mathtt{AtC}$, a two-stage framework that addresses the fundamental challenge of human-centered evaluation when ground truth is costly, unobservable, or future-dependent. Our theoretical analysis establishes three key results: (1) heterogeneous annotator modeling yields more efficient consensus estimation than homogeneous assumptions, (2) isotonic calibration provides risk bounds even under misspecified rankings, and (3) $\mathtt{AtC}$ asymptotically outperforms model-only assessment. Empirically, $\mathtt{AtC}$ demonstrates consistent improvements in accuracy and robustness across semi-synthetic and real-world datasets, particularly under data degradation conditions common in practice. The framework suits human-centered assessment tasks where model-only evaluation faces limitations due to unobservable or costly ground truth, such as when models receive degraded inputs while humans access complete information, or when collecting human comparisons is more cost-effective than measuring ground truth directly. Future work can extend $\mathtt{AtC}$ in several directions: multi-dimensional assessments in which items are judged along multiple criteria, and domains with expert polarization that require multiple distinct consensus rankings. Another direction is to extend our formulation to LLM-as-a-judge settings, where generalizing the heterogeneous annotator to language model evaluators would help build automated and adaptive assessment systems. We believe that addressing these challenges will broaden the impact of collaborative human-AI assessment systems.

ETHICS STATEMENT

This work raises no specific ethics concerns. All datasets used are publicly available and accessed under their respective licenses. We do not collect new human-subject data and do not alter existing human data; our study reorganizes and validates existing resources from a methodological perspective. No personally identifiable information is processed, and no sensitive applications are targeted.

REPRODUCIBILITY STATEMENT

We release code and datasets at `https://github.com/CAMELLIAxt/12500_AtC_supp`. The paper specifies the problem setup and assumptions; the appendix provides complete proofs, algorithm pseudocode, and experiment details.

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

## THE USE OF LARGE LANGUAGE MODELS (LLMs)

We used LLMs solely as general-purpose writing aids: (i) light prose polishing (grammar and clarity), (ii) checking LaTeX math syntax and typesetting, and (iii) generating boilerplate LaTeX table templates. LLMs did *not* contribute to research ideation, problem formulation, algorithm or theorem development, dataset curation, experimental design, or result interpretation. All technical content and decisions were authored and verified by the authors.

## A  ALGORITHM PSEUDOCODE

We provide the pseudocode here to describe the algorithm steps for Stage-1 and Stage-2.

---

**Algorithm 1** Heterogeneous Rank Aggregation

1: **Input:** Comparative judgment data $\{i \succ j \in \mathcal{D}_u : u = 1, \ldots, m\}$ from $m$ annotators.
2: **Output:** Consensus score vector $\widetilde{\mathbf{s}}$ and consensus ranking $\widehat{\pi}$.
3: Initialize item scores $\mathbf{s}_i^{(0)} \leftarrow 0$ for all $i \in [n]$, annotator parameters $\gamma_u^{(0)} \leftarrow 1$ for all $u \in [m]$.
4: **for** $t = 1, 2, \ldots$ **until** convergence **do**
5:  Update score estimates:
  $\mathbf{s}^{(t)} \leftarrow \mathbf{s}^{(t-1)} + \eta_s \nabla_{\mathbf{s}} \ell\Big(\mathbf{s}^{(t-1)}, \gamma^{(t-1)}\Big).$
6:  Identifiability: $\mathbf{s}^{(t)} = (\mathbf{I} - \mathbf{1}\mathbf{1}^\top/n)\mathbf{s}^{(t)}.$
7:  Update annotator parameters:
  $\gamma_u^{(t)} \leftarrow \gamma_u^{(t-1)} + \eta_\gamma \nabla_{\gamma_u} \ell\Big(\mathbf{s}^{(t-1)}, \{\gamma_u^{(t-1)}\}\Big)$
  for all $u \in [m]$.
8:  (Optionally, project $\{\gamma_u^{(t)}\}$ onto its feasible set, e.g., $\frac{1}{m}\sum_u \gamma_u^{(t)} = 1$.)
9: **end for**
10: $\mathbf{s}^* \leftarrow \mathbf{s}^{(t_{\text{final}})}$
11: $\widehat{\pi} \leftarrow \text{rank\_order}(\mathbf{s}^*)$
12: **return** $\widehat{\pi}$.

---

**Algorithm 2** Isotonic Regression Calibration

1: **Input:** Model score vector $\mathbf{s}_p = (\mathbf{s}_{p,1}, \ldots, \mathbf{s}_{p,n})$; consensus ranking $\widehat{\pi}$ over $n$ items.
2: **Output:** Calibrated score vector $\widehat{\mathbf{s}} = (\widehat{\mathbf{s}}_1, \ldots, \widehat{\mathbf{s}}_n)$ that is monotonic w.rt. $\widehat{\pi}$.
3: Relabel item indices of $\mathbf{s}_p$ according to $\widehat{\pi}$ (so that index 1 corresponds to $\widehat{\pi}(1)$, etc.). Let $(y_1, \ldots, y_n)$ be the reordered scores, where $y_1 = \mathbf{s}_{p,\widehat{\pi}(1)}, y_2 = \mathbf{s}_{p,\widehat{\pi}(2)}, \ldots, y_n = \mathbf{s}_{p,\widehat{\pi}(n)}.$
4: Initialize $\widehat{\mathbf{s}}_i \leftarrow y_i$ for $i = 1, \ldots, n$.
5: **repeat**
6:  **for** $i = 1$ to $n - 1$ **do**
7:   **if** $\widehat{\mathbf{s}}_i > \widehat{\mathbf{s}}_{i+1}$ **then**
8:    $\widehat{\mathbf{s}}_i, \widehat{\mathbf{s}}_{i+1} \leftarrow \frac{\widehat{\mathbf{s}}_i + \widehat{\mathbf{s}}_{i+1}}{2}$
9:   **end if**
10:  **end for**
11: **until** $\widehat{\mathbf{s}}_1 \leq \widehat{\mathbf{s}}_2 \leq \cdots \leq \widehat{\mathbf{s}}_n$
12: Undo the reordering: for each item $j$, set $\widehat{\mathbf{s}}_j$ to the value of the calibrated score assigned to item $j$'s position in the sorted order.
13: **return** $\widehat{\mathbf{s}}$.

---

## B  ASYMPTOTIC EFFICIENCY OF THE HETEROGENEOUS THURSTONE MODEL ESTIMATOR

This section details the asymptotic properties of the score estimators derived from the Heterogeneous Thurstone Model (HTM) and compares its efficiency against an estimator from a misspecified homogeneous model.

### B.1  MODEL DEFINITIONS AND PROBLEM STATEMENT

We consider the task of estimating item scores $\mathbf{s} \in \mathbb{R}^n$ from pairwise comparison data. The comparisons are provided by $m$ users, each user $u$ performs $k_u$ comparisons, leading to a total of $N = \sum_{u=1}^m k_u$ observations $Y = \{Y_{ul}\}_{u=1..m, l=1..k_u}$. Each $Y_{ul} \in \{0, 1\}$ represents the outcome of a comparison between items $i_l$ and $j_l$ by user $u$.

### B.1.1 THE **HETERO**GENEOUS THURSTONE MODEL

We assume the true data generating process follows a HTM. Under this model, the probability that user $u$ prefers item $i_l$ over item $j_l$ is given by:

$$P(Y_{ul} = 1|\mathbf{s}, \gamma_u) = F(\gamma_u a_{l,u}^\top \mathbf{s}), \tag{B.1}$$

where $\mathbf{s} \in \mathbb{R}^n$ is the vector of true item scores, $\gamma_u \in \mathbb{R}^+$ is the accuracy parameter for user $u$, $a_{l,u} = e_{i_l} - e_{j_l}$ is a difference vector with $e_i$ being the $i$-th standard basis vector, and $F(\cdot)$ is a known link function, typically a Cumulative Distribution Function (CDF) such as the logistic function (leading to the Heterogeneous Bradley-Terry-Luce model, HBTL) or the Normal CDF (leading to the Heterogeneous Thurstone Case V model, HTCV).

The complete parameter vector for the HTM is $\theta = (\mathbf{s}, \gamma) \in \mathbb{R}^{n+m}$, where $\gamma = (\gamma_1, ..., \gamma_m)^\top$. The log-likelihood for a single observation $(Y_{ul}, a_{l,u})$ given parameters $(\mathbf{s}, \gamma_u)$ is denoted by $\ell(Y_{ul}|\mathbf{s}, \gamma_u) = \log P(Y_{ul}|\mathbf{s}, \gamma_u)$. The average log-likelihood for all $N$ observations is:

$$\mathcal{L}_{hete}(\theta; Y) = \frac{1}{N} \sum_{u=1}^{m} \sum_{l=1}^{k_u} \ell(Y_{ul}|\mathbf{s}, \gamma_u). \tag{B.2}$$

The Maximum Likelihood Estimator (MLE) for the HTM is:

$$\widehat{\theta}_{hete} = (\widehat{\mathbf{s}}_{hete}, \widehat{\gamma}_{hete}) = \arg\max_\theta \mathcal{L}_{hete}(\theta; Y). \tag{B.3}$$

We are primarily interested in the score estimator $\widehat{\mathbf{s}}_{hete}$.

### B.1.2 THE **HOMO**GENEOUS THURSTONE MODEL

For comparison, we consider a misspecified HTM. This model incorrectly assumes that all users have the same accuracy parameter, $\gamma_u = \gamma_0$ for all $u$, where $\gamma_0$ is a fixed constant or a single parameter to be estimated. For simplicity and to highlight the effect of ignoring heterogeneity, we assume $\gamma_0$ is a known constant (e.g., $\gamma_0 = 1$).

The probability of preference under this model is:

$$P(Y_{ul} = 1|\mathbf{s}, \gamma_0) = F(\gamma_0 a_{l,u}^\top \mathbf{s}) \tag{B.4}$$

$$\mathcal{L}_{homo}(\mathbf{s}; Y, \gamma_0) = \frac{1}{N} \sum_{u=1}^{m} \sum_{l=1}^{k_u} \log P(Y_{ul}|\mathbf{s}, \gamma_0) \tag{B.5}$$

The estimator obtained by maximizing $\widehat{\mathbf{s}}_{homo} = \arg\max_\mathbf{s} \mathcal{L}_{homo}(\mathbf{s}; Y, \gamma_0)$, the misspecified likelihood, is a Quasi-Maximum Likelihood Estimator (QMLE)Greene (2003)[Page 734].

The primary objective of Theorem 3.3 is to demonstrate that, when the true data generating process is heterogeneous, the estimator $\widehat{\mathbf{s}}_{hete}$ derived from the correctly specified HTM is asymptotically more efficient than the estimator $\widehat{\mathbf{s}}_{homo}$ derived from the misspecified homogeneous model. This will be shown by comparing their asymptotic covariance matrices.

### B.2 ASSUMPTIONS

To establish the asymptotic properties of the estimators $\widehat{\mathbf{s}}_{hete}$ and $\widehat{\mathbf{s}}_{homo}$, we impose the following assumptions:

**(A1)** *(Independent Observations)* The pairwise comparison outcomes $Y_{ul}$ are independent conditional on the true parameters.

**(A2)** *(True Data Generating Process)* The data are generated by an HTM with true parameters $\theta_0 = (\mathbf{s}^*, \gamma^*) \in \mathbb{R}^{n+m}$.

  • To ensure identifiability of $\mathbf{s}^*$, a constraint such as $\mathbf{1}^\top \mathbf{s}^* = 0$ is imposed, where $\mathbf{1}$ is a vector of ones.

- The vector $\gamma^* = (\gamma_1^*, ..., \gamma_m^*)^\top$ contains at least two distinct values.
- The link function $F(\cdot)$ is a known, twice continuously differentiable CDF.

**(A3)** *(Regularity Conditions)* Both the HTM and the homogeneous model satisfy standard regularity conditions ((White, 1982, Assumptions 1-6)), necessary for the consistency and asymptotic normality of MLE and QMLE respectively. These conditions include compactness of the parameter space, measurability and smoothness of the log-likelihood functions, existence of relevant expectations, unique identifiability of the parameters (or QMLE limits), and non-singularity of key matrices in the identifiable parameter space. Such conditions are generally met by common link functions like the logistic (Gumbel noise) or Normal CDF, provided the parameter space is well-defined and the comparison design is not degenerate. Furthermore, for the correctly specified HTM, Assumption 7 of White (1982) is assumed to hold at $\theta_0$, ensuring the information matrix equivalence.

**(A4)** *(Estimators)* $\widehat{\theta}_{hete} = (\widehat{\mathbf{s}}_{hete}, \widehat{\gamma}_{hete})$ is the MLE for the HTM. $\widehat{\mathbf{s}}_{homo}$ is the QMLE for the homogeneous model (with fixed $\gamma_0 = 1$).

## B.3 Core Idea and Asymptotic Framework

The efficiency comparison between $\widehat{\mathbf{s}}_{hete}$ and $\widehat{\mathbf{s}}_{homo}$ is grounded in the asymptotic theory of MLE and QMLE. The core idea is that an estimator derived from a correctly specified model should be asymptotically more efficient than an estimator derived from a misspecified model.

Under the stated assumptions, the MLE $\widehat{\mathbf{s}}_{hete}$ from the correctly specified HTM is consistent for the true score vector $\mathbf{s}^*$, asymptotically normal, and asymptotically efficient. Its asymptotic covariance matrix, denoted $\Sigma_{\widehat{\mathbf{s}}_{hete}}$, achieves the Cramér-Rao Lower Bound (CRLB) for estimating $\mathbf{s}^*$ under the HTM. This matrix is derived from the inverse (or pseudoinverse, to account for identifiability constraints) of the Fisher information matrix associated with the HTM, specifically from the Schur complement corresponding to the $\mathbf{s}$ parameters.

Conversely, the QMLE $\widehat{\mathbf{s}}_{homo}$ from the misspecified homogeneous model will converge to a parameter vector $\mathbf{s}_*$ that minimizes the Kullback-Leibler divergence between the true data generating process and the misspecified model; generally, $\mathbf{s}_* \neq \mathbf{s}^*$ when true heterogeneity exists. The asymptotic covariance matrix of $\widehat{\mathbf{s}}_{homo}$, denoted $\Sigma_{\widehat{\mathbf{s}}_{homo}}$, is given by the "sandwich" formula, which accounts for the misspecification. Due to the model misspecification, the standard information matrix equivalence does not hold for the homogeneous model, and its QMLE is typically not asymptotically efficient for $\mathbf{s}^*$. Moreover, the asymptotic covariance matrix of an estimator from a correctly specified model will be no larger than that of an estimator from a misspecified model in the Loewner sense. We aim to formalize this by showing that $\Sigma_{\widehat{\mathbf{s}}_{homo}} - \Sigma_{\widehat{\mathbf{s}}_{hete}}$ is a positive semi-definite matrix, and strictly positive definite in the identifiable subspace when true heterogeneity is present.

To formally compare the efficiency of the estimators $\widehat{\mathbf{s}}_{hete}$ and $\widehat{\mathbf{s}}_{homo}$, we first derive their respective asymptotic covariance matrices by deriving Lemma 3.1 and Lemma 3.2.

## B.4 Proof of Lemma 3.1

The estimator $\widehat{\mathbf{s}}_{hete}$ is derived from the HTM. The average negative log-likelihood function for the HTM, with parameters $\theta = (\mathbf{s}, \gamma)$, is given by

$$\mathcal{L}(\mathbf{s}, \gamma) = -\frac{1}{N} \sum_{u=1}^{m} \sum_{l=1}^{k_u} \log P(Y_{ul}|\mathbf{s}, \gamma_u) \tag{B.6}$$

where $N = \sum_{u=1}^{m} k_u$ is the total number of observations, and $P(Y_{ul}|\mathbf{s}, \gamma_u) = F(\gamma_u a_{l,u}^\top \mathbf{s})$ if we denote $g(x; Y) = -\log P(Y|x)$ as the negative log-likelihood for a single observation. For simplicity, we adopt the notation from prior work where $g(x) = -\log F(x)$ is used when the outcome $Y_{ul}$ is 1 or $F(x)$ represents the probability of a specific outcome that leads to the form in the loss function.

The score functions, which are the first-order partial derivatives of $\mathcal{L}(\mathbf{s}, \gamma)$, are:

$$\nabla_{\mathbf{s}}\mathcal{L}(\mathbf{s}, \gamma) = \frac{1}{N} \sum_{u=1}^{m} \sum_{l=1}^{k_u} g'\left(\gamma_u a_{l,u}^{\top}\mathbf{s}\right) \gamma_u a_{l,u} \tag{B.7}$$

$$\nabla_{\gamma}\mathcal{L}(\mathbf{s}, \gamma) = \frac{1}{N} \begin{bmatrix} \sum_{l=1}^{k_1} g'\left(\gamma_1 a_{l,1}^{\top}\mathbf{s}\right) a_{l,1}^{\top}\mathbf{s} \\ \vdots \\ \sum_{l=1}^{k_m} g'\left(\gamma_m a_{l,m}^{\top}\mathbf{s}\right) a_{l,m}^{\top}\mathbf{s} \end{bmatrix} \tag{B.8}$$

The Hessian matrix comprises the second-order partial derivatives:

$$\nabla_{\mathbf{s}}^2\mathcal{L}(\mathbf{s}, \gamma) = \frac{1}{N} \sum_{u=1}^{m} \sum_{l=1}^{k_u} g''\left(\gamma_u a_{l,u}^{\top}\mathbf{s}\right) (\gamma_u)^2 a_{l,u} a_{l,u}^{\top} \tag{B.9}$$

$$\nabla_{\gamma}^2\mathcal{L}(\mathbf{s}, \gamma) = \frac{1}{N}\mathrm{diag}\left(\left[\sum_{l=1}^{k_u} g''\left(\gamma_u a_{l,u}^{\top}\mathbf{s}\right) (a_{l,u}^{\top}\mathbf{s})^2\right]_{u=1}^{m}\right) \tag{B.10}$$

$$(\nabla_{\mathbf{s}}\nabla_{\gamma}\mathcal{L}(\mathbf{s}, \gamma))_{i,u} = \frac{1}{N} \sum_{l=1}^{k_u} (a_{l,u})_i \left[g''\left(\gamma_u a_{l,u}^{\top}\mathbf{s}\right) (a_{l,u}^{\top}\mathbf{s})\gamma_u + g'\left(\gamma_u a_{l,u}^{\top}\mathbf{s}\right)\right] \tag{B.11}$$

The Fisher information matrix for a single observation $I_{hete}(\theta_0)$, evaluated at the true parameters $\theta_0 = (\mathbf{s}^*, \gamma^*)$, is defined as the negative expectation of the Hessian of the log-likelihood for that observation. For the average log-likelihood $\mathcal{L}$, the corresponding expected Hessian (or information scaled by $1/N$) is $I(\mathbf{s}^*, \gamma^*) = -E_{Y|\mathbf{s}^*,\gamma^*}[\nabla^2\mathcal{L}(\mathbf{s}^*, \gamma^*)]$. The blocks of this matrix are:

$$I_{ss} = -E_{Y|\mathbf{s}^*,\gamma^*}[\nabla_{\mathbf{s}}^2\mathcal{L}(\mathbf{s}^*, \gamma^*)] \tag{B.12}$$

$$(I_{\gamma\gamma})_{uu} = -E_{Y|\mathbf{s}^*,\gamma^*}[(\nabla_{\gamma}^2\mathcal{L}(\mathbf{s}^*, \gamma^*))_{uu}] \tag{B.13}$$

$$(I_{s\gamma})_{i,u} = -E_{Y|\mathbf{s}^*,\gamma^*}[(\nabla_{\mathbf{s}}\nabla_{\gamma}\mathcal{L}(\mathbf{s}^*, \gamma^*))_{i,u}] \tag{B.14}$$

and $I_{\gamma s} = I_{s\gamma}^{\top}$.

The log-likelihood for a single comparison $Y_{ul}$ given $\mathbf{s}^*$ and $\gamma_u^*$ is:

$$\ell_{ul}(\mathbf{s}^*, \gamma_u^*) = Y_{ul} \log F(x_{ul}^*) + (1 - Y_{ul})\log(1 - F(x_{ul}^*)), \tag{B.15}$$

where $x_{ul}^* = \gamma_u^* a_{l,u}^{\top}\mathbf{s}^*$. Let $g(x; Y) = -[Y \log F(x) + (1 - Y)\log(1 - F(x))]$. Under standard regularity conditions for MLE and assuming the model is correctly specified (Assumption A3):

$$E_{Y_{ul}|x_{ul}^*}[g'(x_{ul}^*; Y_{ul})] = 0, \tag{B.16}$$

$$-E_{Y_{ul}|x_{ul}^*}[g''(x_{ul}^*; Y_{ul})] = W(x_{ul}^*), \tag{B.17}$$

where $W(x) = \frac{(F'(x))^2}{F(x)(1-F(x))}$.

Substituting these expectations, the blocks of the Fisher information matrix (scaled by $1/N$) become:

$$I_{ss} = \frac{1}{N} \sum_{u=1}^{m} \sum_{l=1}^{k_u} W(x_{ul}^*)(\gamma_u^*)^2 a_{l,u} a_{l,u}^{\top} \tag{B.18}$$

$$(I_{\gamma\gamma})_{uu} = \frac{1}{N} \sum_{l=1}^{k_u} W(x_{ul}^*)(a_{l,u}^{\top}\mathbf{s}^*)^2 \tag{B.19}$$

$$(I_{s\gamma})_{i,u} = \frac{1}{N} \sum_{l=1}^{k_u} W(x_{ul}^*)(a_{l,u}^{\top}\mathbf{s}^*)\gamma_u^*(a_{l,u})_i \tag{B.20}$$

The asymptotic covariance matrix of the MLE $\widehat{\theta}_{hete} = (\widehat{\mathbf{s}}_{hete}, \widehat{\gamma}_{hete})$ is $N^{-1}[I_{hete}(\theta_0)]^{-1}$, where $I_{hete}(\theta_0)$ is the Fisher information for a single observation. The matrix $I(\mathbf{s}^*, \gamma^*)$ derived above corresponds to $N^{-1}I_{hete}(\theta_0)$ if we interpret the sums as averages per observation. We define:

$$S_{total} = N \cdot S = N \cdot (I_{ss} - I_{s\gamma}I_{\gamma\gamma}^{-1}I_{\gamma s}), \tag{B.21}$$

where $I_{ss}, I_{s\gamma}, I_{\gamma\gamma}$ are the blocks of $N \cdot I(\mathbf{s}^*, \gamma^*)$ (i.e., sums without the $1/N$ scaling).

The asymptotic covariance matrix for $\sqrt{N}(\widehat{\mathbf{s}}_{hete} - \mathbf{s}^*)$ is then the top-left $(n \times n)$ block of $[I_{hete}(\theta_0)]^{-1}$. Using the formula for the inverse of a partitioned matrix, this block is:

$$(S_{total}/N)^+ = S^+, \tag{B.22}$$

where $S = I'_{ss} - I'_{s\gamma}(I'_{\gamma\gamma})^{-1}I'_{\gamma s}$, and $I'_{ss}, I'_{s\gamma}, I'_{\gamma\gamma}$ are the blocks of the Fisher information for a single observation (i.e., the expressions equation B.18-equation B.20 without the $1/N$ factor and with sums replaced by expectations or a single representative term if all $k_u = k$ and $N = mk$).

Due to the identifiability constraint:

$$\mathbf{1}^\top \mathbf{s}^* = 0, \tag{B.23}$$

the score $\mathbf{s}$ is estimable only up to an additive constant. This implies that the Fisher information matrix $I_{hete}(\theta_0)$ is singular, and specifically, its block $S$ is singular with $S\mathbf{1} = \mathbf{0}$. Let $S = U\Lambda U^\top$ be the spectral decomposition of $S$, where $U$ is an orthogonal matrix of eigenvectors and $\Lambda = \text{diag}(\lambda_1, ..., \lambda_{n-1}, 0)$ is the diagonal matrix of eigenvalues, with $\lambda_1, ..., \lambda_{n-1} > 0$. The Moore-Penrose pseudoinverse is:

$$S^+ = U\Lambda^+ U^\top = \sum_{i=1}^{n-1} \frac{1}{\lambda_i} u_i u_i^\top, \tag{B.24}$$

where $\Lambda^+ = \text{diag}(1/\lambda_1, 1/\lambda_2, ..., 1/\lambda_{n-1}, 0)$. And this $S^+$ represents the inverse of $S$ restricted to the identifiable subspace:

$$V_{\perp \mathbf{1}} = \{\mathbf{v} \in \mathbb{R}^n : \mathbf{1}^\top \mathbf{v} = 0\}. \tag{B.25}$$

The asymptotic covariance matrix for $\sqrt{N}(\widehat{\mathbf{s}}_{hete} - \mathbf{s}^*)$ is therefore $S^+$. This matrix is the Cramér-Rao Lower Bound for estimating $\mathbf{s}^*$ under the HTM, considering the estimation of $\gamma^*$ and the identifiability constraint on $\mathbf{s}^*$. $\square$

## B.5 PROOF OF LEMMA 3.2

The estimator $\widehat{\mathbf{s}}_{homo}$ is obtained as the QMLE from the HTM, which assumes a common accuracy parameter $\gamma_0$ for all users. The average log-likelihood for this model is:

$$\mathcal{L}_{homo}(\mathbf{s}; Y, \gamma_0) = \frac{1}{N} \sum_{u=1}^{m} \sum_{l=1}^{k_u} \log P(Y_{ul}|\mathbf{s}, \gamma_0) \tag{B.26}$$

where $P(Y_{ul}|\mathbf{s}, \gamma_0) = F(\gamma_0 a_{l,u}^\top \mathbf{s})$. Under Assumption (A2), the true data generating process is an HTM with heterogeneous accuracies $\gamma^* = (\gamma_1^*, ..., \gamma_m^*)^\top$, where not all $\gamma_u^*$ are equal to $\gamma_0$. Thus, the homogeneous model is misspecified. According to White (1982, Theorem 2.2), the QMLE $\widehat{\mathbf{s}}_{homo}$ converges in probability to a parameter vector $\mathbf{s}_*$, which is the unique minimizer of the Kullback-Leibler divergence between the true data generating process and the family of distributions defined by the misspecified homogeneous model. Equivalently, $\mathbf{s}_*$ maximizes the expected misspecified log-likelihood:

$$\mathbf{s}_* = \arg\max_{\mathbf{s}} E_{Y|\mathbf{s}^*, \gamma^*}[\log P(Y_{ul}|\mathbf{s}, \gamma_0)] \tag{B.27}$$

Due to the model misspecification , it is generally true that $\mathbf{s}_* \neq \mathbf{s}^*$.

Furthermore, according to White (1982)[Theorem 3.2], the QMLE $\widehat{\mathbf{s}}_{homo}$ is asymptotically normally distributed:

$$\sqrt{N}(\widehat{\mathbf{s}}_{homo} - \mathbf{s}_*) \xrightarrow{d} N(\mathbf{0}, C_{homo}(\mathbf{s}_*)) \tag{B.28}$$

where $C_{Homo}(\mathbf{s}_*)$ is the "sandwich" covariance matrix given by

$$C_{homo}(\mathbf{s}_*) = [A_{homo}(\mathbf{s}_*)]^{-1} B_{homo}(\mathbf{s}_*)[A_{homo}(\mathbf{s}_*)]^{-1} \qquad \text{(B.29)}$$

The matrices $A_{homo}(\mathbf{s}_*)$ and $B_{homo}(\mathbf{s}_*)$ are defined for a single observation under the true data generating process $(\mathbf{s}^*, \gamma^*)$, evaluated at $\mathbf{s}_*$:

$$A_{homo}(\mathbf{s}_*) = E_{Y|\mathbf{s}^*,\gamma^*}[\nabla_{\mathbf{s}}^2 \log P(Y_{ul}|\mathbf{s}_*, \gamma_0)] \qquad \text{(B.30)}$$

$$B_{homo}(\mathbf{s}_*) = E_{Y|\mathbf{s}^*,\gamma^*}[(\nabla_{\mathbf{s}} \log P(Y_{ul}|\mathbf{s}_*, \gamma_0))(\nabla_{\mathbf{s}} \log P(Y_{ul}|\mathbf{s}_*, \gamma_0))^\top] \qquad \text{(B.31)}$$

The expectation $E_{Y|\mathbf{s}^*,\gamma^*}[\cdot]$ is taken with respect to the true probability distribution of $Y_{ul}$ determined by $\mathbf{s}^*$ and $\gamma^*$. Due to the model misspecification, the information matrix equivalence White (1982)[Theorem 3.3] generally does not hold for the homogeneous model, meaning $A_{homo}(\mathbf{s}_*) \neq -B_{homo}(\mathbf{s}_*)$.

Moreover, to account for the identifiability constraint $\mathbf{1}^\top \mathbf{s} = 0$, the inverses in the sandwich formula are replaced by pseudoinverses. Thus, the asymptotic covariance matrix for $\sqrt{N}(\widehat{\mathbf{s}}_{homo} - \mathbf{s}_*)$ is more precisely written as:

$$\text{AsyVar}(\sqrt{N}(\widehat{\mathbf{s}}_{homo} - \mathbf{s}_*)) = [A_{homo}(\mathbf{s}_*)]^+ B_{homo}(\mathbf{s}_*)[A_{homo}(\mathbf{s}_*)]^+ \qquad \text{(B.32)}$$

□

## B.6 PROOF OF THEOREM 3.3

We aim to compare the asymptotic efficiencies of $\widehat{\mathbf{s}}_{hete}$ and $\widehat{\mathbf{s}}_{homo}$ by examining their respective asymptotic covariance matrices. From Lemma 3.1, the asymptotic covariance matrix for $\sqrt{N}(\widehat{\mathbf{s}}_{hete} - \mathbf{s}^*)$ is $S^+$. This matrix represents the CRLB for estimating $\mathbf{s}^*$ under the correctly specified HTM, within the identifiable subspace defined by $\mathbf{1}^\top \mathbf{s} = 0$. Thus,

$$\text{AsyVar}(\sqrt{N}\widehat{\mathbf{s}}_{hete}) = S^+ \qquad \text{(B.33)}$$

From Lemma 3.2, the asymptotic covariance matrix for $\sqrt{N}(\widehat{\mathbf{s}}_{homo} - \mathbf{s}_*)$ is given by the sandwich formula:

$$\text{AsyVar}(\sqrt{N}\widehat{\mathbf{s}}_{homo}) = [A_{homo}(\mathbf{s}_*)]^+ B_{homo}(\mathbf{s}_*)[A_{homo}(\mathbf{s}_*)]^+ \qquad \text{(B.34)}$$

where $\mathbf{s}_*$ is the probability limit of $\widehat{\mathbf{s}}_{homo}$ under the misspecified homogeneous model, and $A_{homo}(\mathbf{s}_*)$ and $B_{homo}(\mathbf{s}_*)$ are defined as in Lemma 3.2.

To prove the first part of Theorem 3.3, we compare the matrices from equation B.34 and equation B.33. According to the theory of estimation under misspecified models, the asymptotic covariance matrix of a QMLE from a misspecified model cannot be smaller (in the Loewner sense) than the CRLB achieved by an MLE from the correctly specified model, when both are estimating parameters related to the true data generating process. The matrix $S^+$ is the CRLB for $\mathbf{s}^*$ under the true HTM. Therefore, it must hold that:

$$[A_{homo}(\mathbf{s}_*)]^+ B_{homo}(\mathbf{s}_*)[A_{homo}(\mathbf{s}_*)]^+ \succeq S^+ \qquad \text{(B.35)}$$

This inequality establishes that:

$$N \cdot \Sigma_{\widehat{\mathbf{s}}_{homo}} \succeq N \cdot \Sigma_{\widehat{\mathbf{s}}_{hete}}. \qquad \text{(B.36)}$$

For the second part of the theorem, concerning the strict inequality, Assumption (A2) states that true heterogeneity exists, meaning the true user accuracies $\gamma_u^*$ are not all equal to the fixed $\gamma_0$ used in the homogeneous model. This implies that the homogeneous model is genuinely misspecified. Under such misspecification, the information matrix equivalence $A_{homo}(\mathbf{s}_*) = -B_{homo}(\mathbf{s}_*)$ does not hold. The QMLE $\widehat{\mathbf{s}}_{homo}$ fails to incorporate user-specific accuracy information, leading to a loss of statistical efficiency compared to the MLE $\widehat{\mathbf{s}}_{hete}$ from the correctly specified HTM. This efficiency loss manifests as a strict inequality in equation B.35 when considering the identifiable subspace $V_{\perp \mathbf{1}} = \{\mathbf{v} \in \mathbb{R}^n : \mathbf{1}^\top \mathbf{v} = 0\}$. Specifically, for any non-zero vector $\mathbf{v} \in V_{\perp \mathbf{1}}$,

$$\mathbf{v}^\top \left([A_{homo}(\mathbf{s}_*)]^+ B_{homo}(\mathbf{s}_*)[A_{homo}(\mathbf{s}_*)]^+\right) \mathbf{v} > \mathbf{v}^\top S^+ \mathbf{v} \qquad \text{(B.37)}$$

This occurs because the misspecification prevents the sandwich covariance matrix from collapsing to the simpler inverse Fisher information form and results in a larger variance for any linear combination of parameters within the identifiable subspace, compared to the CRLB achieved by the correctly specified model. Thus, $N \cdot \Sigma_{\widehat{\mathbf{s}}_{homo}} - N \cdot \Sigma_{\widehat{\mathbf{s}}_{hete}}$ is strictly positive definite on $V_{\perp \mathbf{1}}$. □

# C ANALYSIS OF LSE UNDER MODEL MISSPECIFICATION AND NON-ZERO MEAN NOISE

## C.1 PRELIMINARIES: CONES AND STATISTICAL DIMENSION

This section introduces fundamental concepts related to convex cones, tangent cones, and statistical dimension, which are essential for the subsequent analysis.

**Definition C.1** (Convex Cone). A set $K \subset \mathbb{R}^n$ is a **convex cone** if for any $\boldsymbol{v}_1, \boldsymbol{v}_2 \in K$ and any non-negative scalars $t_1, t_2 \geq 0$, the linear combination $t_1 \boldsymbol{v}_1 + t_2 \boldsymbol{v}_2$ is also in $K$. If, additionally, $K$ is a closed set, it is a closed convex cone.

**Definition C.2** (Tangent Cone). Let $K \subset \mathbb{R}^n$ be a closed convex set and let $\boldsymbol{u} \in K$. The **tangent cone** to $K$ at $\boldsymbol{u}$, denoted by $\mathcal{T}_{K,\boldsymbol{u}}$, is defined as:

$$\mathcal{T}_{K,\boldsymbol{u}} = \overline{\{t(\boldsymbol{v} - \boldsymbol{u}) : t > 0, \boldsymbol{v} \in K\}} \tag{C.1}$$

where $\overline{\{\cdot\}}$ denotes the closure of the set. Intuitively, the tangent cone $\mathcal{T}_{K,\boldsymbol{u}}$ comprises all feasible directions from $\boldsymbol{u}$ scaled by non-negative scalars, along which one can move infinitesimally while remaining within $K$.

If $K$ is a closed convex cone, the definition of the tangent cone at $\boldsymbol{u} \in K$ simplifies due to the cone property (i.e., if $\boldsymbol{x} \in K$, then $t\boldsymbol{x} \in K$ for $t \geq 0$). For a closed convex cone $K$ and $\boldsymbol{u} \in K$, the tangent cone is given by :

$$\mathcal{T}_{K,\boldsymbol{u}} = \overline{K + \text{span}(\{-\boldsymbol{u}\})} = \overline{\{\boldsymbol{v} - t\boldsymbol{u} : \boldsymbol{v} \in K, t \geq 0\}} \tag{C.2}$$

where the sum is the Minkowski sum.

**Definition C.3** (Isotonic Cone). The isotonic cone (or non-decreasing cone) in $\mathbb{R}^n$, denoted by $\mathcal{M}$, is defined as:

$$\mathcal{M} := \{\boldsymbol{x} = (x_1, \ldots, x_n)^T \in \mathbb{R}^n : x_1 \leq x_2 \leq \cdots \leq x_n\}$$

The set $\mathcal{M}$ is a closed convex polyhedral cone. For any $\boldsymbol{u} \in \mathcal{M}$, the tangent cone $\mathcal{T}_{\mathcal{M},\boldsymbol{u}}$ captures the local directional constraints at $\boldsymbol{u}$. If $u_i < u_{i+1}$ for some $i$, the tangent cone allows more freedom in the $i$-th and $(i+1)$-th coordinates compared to when $u_i = u_{i+1}$, where the constraint $x_i \leq x_{i+1}$ becomes active for directions in the cone.

**Definition C.4** (Statistical Dimension). Let $K \subset \mathbb{R}^n$ be a closed convex cone. Its statistical dimension, denoted by $\delta(K)$, is defined as:

$$\delta(K) := \mathbb{E}\left[|\Pi_K(\mathbf{g})|_2^2\right] \tag{C.3}$$

where $\mathbf{g} \sim \mathcal{N}(\mathbf{0}, I_n)$ is a standard Gaussian vector in $\mathbb{R}^n$, and $\Pi_K(\mathbf{g})$ is the Euclidean projection of $\boldsymbol{g}$ onto $K$. Equivalently, the statistical dimension can also be expressed as:

$$\delta(K) = \mathbb{E}\left[\mathbf{g}^T \Pi_K(\mathbf{g})\right] = \mathbb{E}\left[\left(\sup_{\boldsymbol{\theta} \in K : |\boldsymbol{\theta}|_2 \leq 1} \mathbf{g}^T \boldsymbol{\theta}\right)^2\right] \tag{C.4}$$

The statistical dimension provides a measure of the "complexity" of the cone from a statistical perspective, often related to the effective number of parameters in a model constrained to the cone.

## C.2 PROBLEM SETTING AND RELATION TO PRIOR WORK

Let $\mathbf{s} \in \mathbb{R}^n$ be an underlying ground truth score vector. The primary target of our estimation in the second stage is a related vector $\widetilde{\mathbf{s}} \in \mathbb{R}^n$, which can be thought of as a "subjective optimal point" or a noisy version of $\mathbf{s}$. Specifically, $\widetilde{\mathbf{s}}$ is generated as:

$$\widetilde{\mathbf{s}} = \mathbf{s} + \widetilde{\boldsymbol{\epsilon}}, \tag{C.5}$$

where $\widetilde{\boldsymbol{\epsilon}} \sim \mathcal{N}(\mathbf{0}, \widetilde{\sigma}^2 I_n)$. Thus, $\widetilde{\mathbf{s}}$ is a random vector with $\mathbb{E}[\widetilde{\mathbf{s}}] = \mathbf{s}$. We assume the true isotonic cone relevant to our target $\widetilde{\mathbf{s}}$ is $\widetilde{\mathcal{M}} := \mathcal{M}_{\pi(\widetilde{\mathbf{s}})}$ (or $\mathcal{M}_{\pi(\mathbf{s})}$, if the true ordering is dictated by the mean $\mathbf{s}$; this distinction should be noted, but for now, we assume $\widetilde{\mathbf{s}} \in \widetilde{\mathcal{M}}$ in expectation or with high probability).

In the first stage of our overall procedure, a separate algorithm (e.g., Algorithm 1 from Jin et al. (2020)) processes auxiliary data to produce an estimate of latent scores, $\mathbf{s}^* \in \mathbb{R}^n$. The ranking induced by $\mathbf{s}^*$, denoted $\widehat{\pi} := \pi(\mathbf{s}^*)$, defines a (random) isotonic cone $\widehat{\mathcal{M}} := \mathcal{M}_{\widehat{\pi}}$. This cone $\widehat{\mathcal{M}}$ serves as the constraint set for the second stage.

The observations for the second stage, denoted by $\mathbf{s}_p \in \mathbb{R}^n$, are related to the ground truth $\mathbf{s}$ and a fixed, unknown systematic bias vector $\boldsymbol{\nu} \in \mathbb{R}^n$ as follows:

$$\mathbf{s}_p = \mathbf{s} + \boldsymbol{\nu}. \tag{C.6}$$

Using the relationship $\mathbf{s} = \widetilde{\mathbf{s}} - \widetilde{\boldsymbol{\epsilon}}$, we can express $\mathbf{s}_p$ in terms of our target $\widetilde{\mathbf{s}}$:

$$\mathbf{s}_p = (\widetilde{\mathbf{s}} - \widetilde{\boldsymbol{\epsilon}}) + \boldsymbol{\nu} = \widetilde{\mathbf{s}} + \boldsymbol{\nu} - \widetilde{\boldsymbol{\epsilon}}$$

The final estimator for the target $\widetilde{\mathbf{s}}$ is obtained by projecting $\mathbf{s}_p$ onto the estimated cone $\widehat{\mathcal{M}}$:

$$\widehat{\mathbf{s}} := \Pi_{\widehat{\mathcal{M}}}(\mathbf{s}_p)$$

Our objective is to analyze the risk of this estimator with respect to $\widetilde{\mathbf{s}}$:

$$\mathbb{E}[\|\widehat{\mathbf{s}} - \widetilde{\mathbf{s}}\|_2^2]. \tag{C.7}$$

The expectation is over all sources of randomness: $\widetilde{\boldsymbol{\epsilon}}$ (which makes $\widetilde{\mathbf{s}}$ random) and $\widehat{\mathbf{s}}_{\mathrm{alg1}}$ (which makes $\widehat{\mathcal{M}}$ random).

The effective noise when estimating $\widetilde{\mathbf{s}}$ from $\mathbf{s}_p$ is:

$$\boldsymbol{\xi} := \mathbf{s}_p - \widetilde{\mathbf{s}} = \boldsymbol{\nu} - \widetilde{\boldsymbol{\epsilon}}, \tag{C.8}$$

in which $\boldsymbol{\xi} \sim \mathcal{N}(\boldsymbol{\nu}, \widetilde{\sigma}^2 I_n)$, as $\mathbb{E}[\boldsymbol{\xi}] = \boldsymbol{\nu}$ and $\mathrm{Cov}(\boldsymbol{\xi}) = \mathrm{Cov}(-\widetilde{\boldsymbol{\epsilon}}) = \widetilde{\sigma}^2 I_n$.

## C.3 PROOF OF THEOREM 3.5

In stage-1, the target is $\widetilde{\mathbf{s}}$, and our observation is $\mathbf{s}_p$. The effective noise is $\boldsymbol{\xi} = \mathbf{s}_p - \widetilde{\mathbf{s}} \sim \mathcal{N}(\boldsymbol{\nu}, \widetilde{\sigma}^2 I_n)$.

By Lemma 3.4, we have:

$$\|\widehat{\mathbf{s}} - \widetilde{\mathbf{s}}\|_2^2 - \|\boldsymbol{u} - \widetilde{\mathbf{s}}\|_2^2 \le \frac{1}{n} \left( \sup_{\boldsymbol{\theta} \in \mathcal{T}_{\widehat{\mathcal{M}}, \boldsymbol{u}} : \|\boldsymbol{\theta}\|_2 \le 1} \boldsymbol{\xi}^T \boldsymbol{\theta} \right)^2 \tag{C.9}$$

Taking expectation with respect to $\boldsymbol{\xi}$ (conditional on $\widehat{\mathcal{M}}$ and $\widetilde{\mathbf{s}}$), and choosing $\boldsymbol{u} = \Pi_{\widehat{\mathcal{M}}}(\widetilde{\mathbf{s}})$, we have:

$$\mathbb{E}_{\boldsymbol{\xi}}[\|\widehat{\mathbf{s}} - \widetilde{\mathbf{s}}\|_2^2 | \widehat{\mathcal{M}}, \widetilde{\mathbf{s}}] \le \|\Pi_{\widehat{\mathcal{M}}}(\widetilde{\mathbf{s}}) - \widetilde{\mathbf{s}}\|_2^2 + \frac{1}{n} \mathbb{E}_{\boldsymbol{\xi}} \left[ \left( \sup_{\boldsymbol{\theta} \in W'} \boldsymbol{\xi}^T \boldsymbol{\theta} \right)^2 \middle| \widehat{\mathcal{M}}, \widetilde{\mathbf{s}} \right] \tag{C.10}$$

where $W' = \{\boldsymbol{\theta} \in \mathcal{T}_{\widehat{\mathcal{M}}, \Pi_{\widehat{\mathcal{M}}}(\widetilde{\mathbf{s}})} : \|\boldsymbol{\theta}\|_2 \le 1\}$. Let $\boldsymbol{\epsilon}' = \boldsymbol{\xi} - \boldsymbol{\nu} = -\widetilde{\boldsymbol{\epsilon}} \sim \mathcal{N}(\mathbf{0}, \widetilde{\sigma}^2 I_n)$. The expectation term is bounded using $(a + b)^2 \le 2a^2 + 2b^2$:

$$\mathbb{E}_{\boldsymbol{\xi}} \left[ \left( \sup_{\boldsymbol{\theta} \in W'} ((\boldsymbol{\xi} - \boldsymbol{\nu})^T \boldsymbol{\theta} + \boldsymbol{\nu}^T \boldsymbol{\theta}) \right)^2 \middle| \widehat{\mathcal{M}}, \widetilde{\mathbf{s}} \right] \le 2 \mathbb{E}_{\boldsymbol{\epsilon}'} \left[ \left( \sup_{\boldsymbol{\theta} \in W'} (\boldsymbol{\epsilon}')^T \boldsymbol{\theta} \right)^2 \middle| \widehat{\mathcal{M}}, \widetilde{\mathbf{s}} \right] + 2 \left( \sup_{\boldsymbol{\theta} \in W'} \boldsymbol{\nu}^T \boldsymbol{\theta} \right)^2 \tag{C.11}$$

$$= 2 \widetilde{\sigma}^2 \delta(\mathcal{T}_{\widehat{\mathcal{M}}, \Pi_{\widehat{\mathcal{M}}}(\widetilde{\mathbf{s}})}) + 2 \|\Pi_{\mathcal{T}_{\widehat{\mathcal{M}}, \Pi_{\widehat{\mathcal{M}}}(\widetilde{\mathbf{s}})}}(\boldsymbol{\nu})\|_2^2 \tag{C.12}$$

Thus, given $\widehat{\mathcal{M}}$ and $\widetilde{\mathbf{s}}$, the conditional risk is:

$$\mathbb{E}_{\boldsymbol{\xi}}[\|\widehat{\mathbf{s}} - \widetilde{\mathbf{s}}\|_2^2 | \widehat{\mathcal{M}}, \widetilde{\mathbf{s}}] \le \|\Pi_{\widehat{\mathcal{M}}}(\widetilde{\mathbf{s}}) - \widetilde{\mathbf{s}}\|_2^2 + \frac{2\widetilde{\sigma}^2}{n} \delta(\mathcal{T}_{\widehat{\mathcal{M}}, \Pi_{\widehat{\mathcal{M}}}(\widetilde{\mathbf{s}})}) + \frac{2}{n} \|\Pi_{\mathcal{T}_{\widehat{\mathcal{M}}, \Pi_{\widehat{\mathcal{M}}}(\widetilde{\mathbf{s}})}}(\boldsymbol{\nu})\|_2^2 \tag{C.13}$$

Taking the full expectation over $\widehat{\mathcal{M}}$ and $\widetilde{\mathbf{s}}$:

$$\mathbb{E}[\|\widehat{\mathbf{s}} - \widetilde{\mathbf{s}}\|_2^2] \le \mathbb{E}_{\widehat{\mathcal{M}}, \widetilde{\mathbf{s}}} \left[ \|\Pi_{\widehat{\mathcal{M}}}(\widetilde{\mathbf{s}}) - \widetilde{\mathbf{s}}\|_2^2 \right] \tag{C.14}$$

$$+ \mathbb{E}_{\widehat{\mathcal{M}}, \widetilde{\mathbf{s}}} \left[ \frac{2\widetilde{\sigma}^2}{n} \delta(\mathcal{T}_{\widehat{\mathcal{M}}, \Pi_{\widehat{\mathcal{M}}}(\widetilde{\mathbf{s}})}) \right] \tag{C.15}$$

$$+ \mathbb{E}_{\widehat{\mathcal{M}}, \widetilde{\mathbf{s}}} \left[ \frac{2}{n} \|\Pi_{\mathcal{T}_{\widehat{\mathcal{M}}, \Pi_{\widehat{\mathcal{M}}}(\widetilde{\mathbf{s}})}}(\boldsymbol{\nu})\|_2^2 \right] \tag{C.16}$$

We aim to bound the risk $\mathbb{E}[\|\widehat{\mathbf{s}} - \widetilde{\mathbf{s}}\|_2^2]$, where $\widehat{\mathbf{s}} = \Pi_{\widehat{\mathcal{M}}}(\mathbf{s}_p)$ is the estimator for the target $\widetilde{\mathbf{s}}$. Recall the key relationships:

- Target for estimation: $\widetilde{\mathbf{s}} = \mathbf{s} + \widetilde{\epsilon}$, where $\widetilde{\epsilon} \sim \mathcal{N}(\mathbf{0}, \widetilde{\sigma}^2 I_n)$.
- Observation: $\mathbf{s}_p = \mathbf{s} + \boldsymbol{\nu} = \widetilde{\mathbf{s}} - \widetilde{\epsilon} + \boldsymbol{\nu}$.
- Effective noise for estimating $\widetilde{\mathbf{s}}$ from $\mathbf{s}_p$: $\boldsymbol{\xi} = \mathbf{s}_p - \widetilde{\mathbf{s}} = \boldsymbol{\nu} - \widetilde{\epsilon} \sim \mathcal{N}(\boldsymbol{\nu}, \widetilde{\sigma}^2 I_n)$.
- $\widehat{\mathcal{M}} = \mathcal{M}_{\pi(\mathbf{s}^*)}$, where $\mathbf{s}^*$ is the output of a first-stage algorithm, estimating $\widetilde{\mathbf{s}}$.
- We assume the expected ordering relevant for $\widetilde{\mathbf{s}}$ is $\pi(\mathbb{E}[\widetilde{\mathbf{s}}]) = \pi(\mathbf{s})$. Let $\widetilde{\mathcal{M}} = \mathcal{M}_{\pi(\widetilde{\mathbf{s}})}$. The error term arises if $\widehat{\mathcal{M}} \neq \widetilde{\mathcal{M}}$ and $\widetilde{\mathbf{s}} \notin \widehat{\mathcal{M}}$.

From Lemma 3.4 (or Eq. equation C.14-equation C.16 in previous discussions), the total risk is bounded by:

$$\mathbb{E}[\|\widehat{\mathbf{s}} - \widetilde{\mathbf{s}}\|_2^2] \leq \underbrace{\mathbb{E}_{\widehat{\mathcal{M}}, \widetilde{\mathbf{s}}}\left[\|\Pi_{\widehat{\mathcal{M}}}(\widetilde{\mathbf{s}}) - \widetilde{\mathbf{s}}\|_2^2\right]}_{\text{Term 1: Projection Error from Misspecified Cone}} \tag{C.17}$$

$$+ \underbrace{\mathbb{E}_{\widehat{\mathcal{M}}, \widetilde{\mathbf{s}}}\left[\frac{2\widetilde{\sigma}^2}{n}\delta(\mathcal{T}_{\widehat{\mathcal{M}}, \Pi_{\widehat{\mathcal{M}}}(\widetilde{\mathbf{s}})})\right]}_{\text{Term 2: Statistical Error (Zero-Mean Noise Component)}} \tag{C.18}$$

$$+ \underbrace{\mathbb{E}_{\widehat{\mathcal{M}}, \widetilde{\mathbf{s}}}\left[\frac{2}{n}\|\Pi_{\mathcal{T}_{\widehat{\mathcal{M}}, \Pi_{\widehat{\mathcal{M}}}(\widetilde{\mathbf{s}})}}(\boldsymbol{\nu})\|_2^2\right]}_{\text{Term 3: Bias Error Component}} \tag{C.19}$$

We now bound each term.

**Bounding Term 1: Projection Error from Misspecified Cone.** Term 1 is $\mathbb{E}_{\widehat{\mathcal{M}}, \widetilde{\mathbf{s}}}\left[\|\Pi_{\widehat{\mathcal{M}}}(\widetilde{\mathbf{s}}) - \widetilde{\mathbf{s}}\|_2^2\right]$. This term captures the error due to projecting the target $\widetilde{\mathbf{s}}$ onto the cone $\widehat{\mathcal{M}}$, which is estimated from $\mathbf{s}^*$ and may differ from the true cone associated with $\widetilde{\mathbf{s}}$ (or its mean $\mathbf{s}$).

Let $\widetilde{\pi} = \pi(\widetilde{\mathbf{s}})$ be the true ranking corresponding to the target of the first-stage algorithm. Let $\widetilde{\mathcal{M}} = \mathcal{M}_{\widetilde{\pi}}$. The event of an incorrect cone $\widehat{\mathcal{M}}$ (relative to the first stage's own target) is $A^c \equiv \{\widehat{\pi} \neq \widetilde{\pi}\}$, where $\widehat{\pi} = \pi(\mathbf{s}^*)$.

$$\mathbb{E}_{\widehat{\mathcal{M}}, \widetilde{\mathbf{s}}}\left[\|\Pi_{\widehat{\mathcal{M}}}(\widetilde{\mathbf{s}}) - \widetilde{\mathbf{s}}\|_2^2\right] = \mathbb{E}_{\widetilde{\mathbf{s}}}\left[\mathbb{E}_{\widehat{\mathcal{M}}|\widetilde{\mathbf{s}}}\left[\|\Pi_{\widehat{\mathcal{M}}}(\widetilde{\mathbf{s}}) - \widetilde{\mathbf{s}}\|_2^2\right]\right] \tag{C.20}$$

If $\widetilde{\mathbf{s}} \in \widehat{\mathcal{M}}$ (i.e., the random target happens to satisfy the random cone's constraints), then $\|\Pi_{\widehat{\mathcal{M}}}(\widetilde{\mathbf{s}}) - \widetilde{\mathbf{s}}\|_2^2 = 0$. This occurs if $\pi(\mathbf{s}^*)$ is compatible with $\widetilde{\mathbf{s}}$. We follow the previous decomposition:

$$\mathbb{E}_{\widehat{\mathcal{M}}|\widetilde{\mathbf{s}}}\left[\|\Pi_{\widehat{\mathcal{M}}}(\widetilde{\mathbf{s}}) - \widetilde{\mathbf{s}}\|_2^2\right] = \mathbb{E}_{\widehat{\mathcal{M}}|\widetilde{\mathbf{s}}}\left[\|\Pi_{\widehat{\mathcal{M}}}(\widetilde{\mathbf{s}}) - \widetilde{\mathbf{s}}\|_2^2\big|A^c\right] P(A^c) \tag{C.21}$$

(assuming $\|\Pi_{\widetilde{\mathcal{M}}}(\widetilde{\mathbf{s}}) - \widetilde{\mathbf{s}}\|_2^2$ is negligible or zero if $\pi(\mathbf{s}) = \widetilde{\pi}$). The term $\mathbb{E}_{\widehat{\mathcal{M}}|\widetilde{\mathbf{s}}}\left[\|\Pi_{\widehat{\mathcal{M}}}(\widetilde{\mathbf{s}}) - \widetilde{\mathbf{s}}\|_2^2\big|A^c\right]$ is bounded by

$$\sup_{\widehat{\mathcal{M}}} \|\Pi_{\widehat{\mathcal{M}}}(\widetilde{\mathbf{s}}) - \widetilde{\mathbf{s}}\|_2^2 \leq n\text{Var}(\widetilde{\mathbf{s}}), \tag{C.22}$$

where $\text{Var}(\widetilde{\mathbf{s}})$ is the sample variance of the components of the specific realization of $\widetilde{\mathbf{s}}$. Taking expectation over $\widetilde{\mathbf{s}}$:

$$\mathbb{E}_{\widehat{\mathcal{M}}, \widetilde{\mathbf{s}}}\left[\|\Pi_{\widehat{\mathcal{M}}}(\widetilde{\mathbf{s}}) - \widetilde{\mathbf{s}}\|_2^2\right] \leq \mathbb{E}_{\widetilde{\mathbf{s}}}[n\text{Var}(\widetilde{\mathbf{s}})]P(A^c) \tag{C.23}$$

The term $\mathbb{E}_{\widetilde{\mathbf{s}}}[n\text{Var}(\widetilde{\mathbf{s}})]$ is related to the expected dispersion of $\widetilde{\mathbf{s}}$. Since $\widetilde{\mathbf{s}} = \mathbf{s} + \widetilde{\epsilon}$, $\mathbb{E}[\widetilde{\mathbf{s}}_i] = \mathbf{s}_i$.

$$\mathbb{E}_{\widetilde{\mathbf{s}}}\left[\sum_{i=1}^n (\widetilde{\mathbf{s}}_i - \bar{\widetilde{\mathbf{s}}})^2\right] = \mathbb{E}_{\widetilde{\mathbf{s}}}\left[\sum_{i=1}^n \widetilde{\mathbf{s}}_i^2 - n\bar{\widetilde{\mathbf{s}}}^2\right] \tag{C.24}$$

This can be further analyzed, but for a simpler bound, we can use:

$$\mathbb{E}_{\widetilde{\mathbf{s}}}[n\mathrm{Var}(\widetilde{\mathbf{s}})] \le \mathbb{E}_{\widetilde{\mathbf{s}}}[\|\widetilde{\mathbf{s}}\|_2^2] = \|\mathbf{s}\|_2^2 + n\widetilde{\sigma}^2. \tag{C.25}$$

For a potentially looser but simpler worst-case, if the range of $\widetilde{\mathbf{s}}_i$ is bounded, $n\mathrm{Var}(\widetilde{\mathbf{s}})$ can be bounded by a term related to the range, e.g., $n(R_{\widetilde{\mathbf{s}}})^2/4$ if $\widetilde{\mathbf{s}}_i \in [a,b]$ and $R_{\widetilde{\mathbf{s}}} = b - a$. A bound that depends on the properties of the ground truth $\mathbf{s}$ and noise $\widetilde{\sigma}^2$ is needed here. For now, we denote it as $\mathbb{E}[n\mathrm{Var}(\widetilde{\mathbf{s}})]$.

The probability of a ranking error $P(A^c) = P(\pi(\mathbf{s}^*) \ne \pi(\widetilde{\mathbf{s}}))$ is:

$$P(A^c) \le \sum_{j,k:\, \widetilde{\mathbf{s}}_j < \widetilde{\mathbf{s}}_k} P(\mathbf{s}_j^* > \mathbf{s}_k^*) \tag{C.26}$$

Let $\eta = \mathbf{s}^* - \widetilde{\mathbf{s}} \sim \mathcal{N}(0, \Sigma_{\mathbf{s}^*})$, where $\Sigma_{\mathbf{s}^*} = \frac{1}{mk}S^+$. Let $\widetilde{\Delta}_{kj} = \widetilde{\mathbf{s}}_k - \widetilde{\mathbf{s}}_j$, and $\sigma^*_{X_{jk}}{}^2 = (\boldsymbol{e}_j - \boldsymbol{e}_k)^\top \Sigma_{\mathbf{s}^*}(\boldsymbol{e}_j - \boldsymbol{e}_k)$. Using the Chernoff-Cramer bound:

$$P(\mathbf{s}_j^* > \mathbf{s}_k^*) \le \frac{\sigma^*_{X_{jk}}}{(\widetilde{\Delta}_{kj})\sqrt{2\pi}} \exp\left(-\frac{(\widetilde{\Delta}_{kj})^2}{2\sigma^*_{X_{jk}}{}^2}\right) \tag{C.27}$$

This completes the proof of Corollary 3.6. Thus,

$$\text{Term 1} \le \mathbb{E}_{\widetilde{\mathbf{s}}}[n\mathrm{Var}(\widetilde{\mathbf{s}})] \sum_{j,k:\widetilde{\mathbf{s}}_j < \widetilde{\mathbf{s}}_k} \frac{\sigma^*_{X_{jk}}}{(\widetilde{\Delta}_{kj})\sqrt{2\pi}} \exp\left(-\frac{(\widetilde{\Delta}_{kj})^2}{2\sigma^*_{X_{jk}}{}^2}\right) \tag{C.28}$$

Alternatively, using the expected number of inversions $Inv(\pi^*, \widetilde{\pi})$ for the first stage:

$$\sum_{j,k:\widetilde{\mathbf{s}}_j < \widetilde{\mathbf{s}}_k} P(\mathbf{s}_j^* > \mathbf{s}_k^*) = \mathbb{E}[Inv(\pi^*, \widetilde{\pi})] \tag{C.29}$$

Then,

$$\text{Term 1} \le \mathbb{E}_{\widetilde{\mathbf{s}}}[n\mathrm{Var}(\widetilde{\mathbf{s}})] \cdot \mathbb{E}[Inv(\pi^*, \widetilde{\pi})] \tag{C.30}$$

**Bounding Term 2: Statistical Error.** Term 2 from the main risk decomposition Eq. equation C.15 is given by:

$$\text{Term 2} = \mathbb{E}_{\widehat{\mathcal{M}}, \widetilde{\mathbf{s}}}\left[\frac{2\widetilde{\sigma}^2}{n}\delta(\mathcal{T}_{\widehat{\mathcal{M}}, \Pi_{\widehat{\mathcal{M}}}(\widetilde{\mathbf{s}})})\right] \tag{C.31}$$

Here, $\widehat{\mathcal{M}} = \mathcal{M}_{\pi(\mathbf{s}^*)}$ is the estimated isotonic cone, which is always a monotone cone . The point $\Pi_{\widehat{\mathcal{M}}}(\widetilde{\mathbf{s}})$ is an element of $\widehat{\mathcal{M}}$.

A fundamental property of tangent cones is that for any closed convex set $K$ and any point $\boldsymbol{u} \in K$, the statistical dimension of the tangent cone $\mathcal{T}_{K,\boldsymbol{u}}$ is bounded by the statistical dimension of the set $K$ itself:

$$\delta(\mathcal{T}_{K,\boldsymbol{u}}) \le \delta(K) \tag{C.32}$$

This result is standard in the analysis of such problems (see, e.g., discussions related to statistical dimension in Amelunxen et al. (2014)). In our case, $K = \widehat{\mathcal{M}}$. Since $\widehat{\mathcal{M}}$ is always an isotonic cone, its statistical dimension is the same as that of the standard isotonic cone $\mathcal{M}_n^\uparrow$. It is known (cf. (Amelunxen et al., 2014, Eq. (D.12))) that the statistical dimension of the standard isotonic cone $\mathcal{M}_n^\uparrow$ is given by the $n$-th harmonic number $H_n$:

$$\delta(\mathcal{M}_n^\uparrow) = \sum_{k=1}^{n} \frac{1}{k} = H_n \tag{C.33}$$

Therefore, for any realization of $\widehat{\mathcal{M}}$ and $\widetilde{\mathbf{s}}$:

$$\delta(\mathcal{T}_{\widehat{\mathcal{M}}, \Pi_{\widehat{\mathcal{M}}}(\widetilde{\mathbf{s}})}) \le \delta(\widehat{\mathcal{M}}) = \delta(\mathcal{M}_n^\uparrow) = H_n \tag{C.34}$$

The harmonic number $H_n$ has well-known asymptotic expansions. For $n \geq 1$:

$$H_n = \ln(n) + \gamma_E + \frac{1}{2n} - \frac{1}{12n^2} + O\left(\frac{1}{n^4}\right) \tag{C.35}$$

where $\gamma_E \approx 0.57721$ is the Euler-Mascheroni constant. Thus, we can use the upper bound:

$$H_n \leq \ln(n) + \gamma_E + \frac{1}{2n} \tag{C.36}$$

For a simpler, more common bound in such analyses, $H_n \approx \ln(n) + \gamma_E$, or $H_n \leq \ln(n) + 1$ for $n \geq 1$. The bound $\log(en) = \ln(n) + 1$ used in Bellec (2018) (Eq. 1.29) is a convenient and slightly looser upper bound for $H_n$. Using $H_n \leq \ln(n) + \gamma_E + O(1/n)$:

$$\text{Term 2} \leq \mathbb{E}_{\widehat{\mathcal{M}},\widetilde{\mathbf{s}}}\left[\frac{2\widetilde{\sigma}^2}{n}H_n\right] = \frac{2\widetilde{\sigma}^2 H_n}{n} = \frac{2\widetilde{\sigma}^2}{n}\left(\ln(n) + \gamma_E + O\left(\frac{1}{n}\right)\right) \tag{C.37}$$

Alternatively, using the $\log(en)$ bound for consistency with some literature:

$$\text{Term 2} \leq \frac{2\widetilde{\sigma}^2 \log(en)}{n} \tag{C.38}$$

This provides a bound for the statistical error component that depends on the variance $\widetilde{\sigma}^2$ of the zero-mean noise component $\widetilde{\epsilon}$ and logarithmically on $n$.

**Bounding Term 3: Bias Error Component.** Term 3 from the main risk decomposition Eq. equation C.16 is given by:

$$\text{Term 3} = \mathbb{E}_{\widehat{\mathcal{M}},\widetilde{\mathbf{s}}}\left[\frac{2}{n}\|\Pi_{\mathcal{T}_{\widehat{\mathcal{M}},\Pi_{\widehat{\mathcal{M}}}(\widetilde{\mathbf{s}})}}(\boldsymbol{\nu})\|_2^2\right] \tag{C.39}$$

This term arises from the fixed systematic bias $\boldsymbol{\nu}$ in the effective noise $\boldsymbol{\xi} = \boldsymbol{\nu} - \widetilde{\epsilon}$. The expectation $\mathbb{E}_{\widehat{\mathcal{M}},\widetilde{\mathbf{s}}}$ averages over the randomness of the estimated cone $\widehat{\mathcal{M}}$ and the randomness of the target $\widetilde{\mathbf{s}}$, which influences the point $\Pi_{\widehat{\mathcal{M}}}(\widetilde{\mathbf{s}})$ at which the tangent cone is evaluated.

Let $\boldsymbol{u}_{\text{proj}} = \Pi_{\widehat{\mathcal{M}}}(\widetilde{\mathbf{s}})$. We need to bound $\|\Pi_{\mathcal{T}_{\widehat{\mathcal{M}},\boldsymbol{u}_{\text{proj}}}}(\boldsymbol{\nu})\|_2^2$. A fundamental property of projection onto a closed convex set is that it does not increase the $L_2$ norm. Therefore,

$$\|\Pi_{\mathcal{T}_{\widehat{\mathcal{M}},\boldsymbol{u}_{\text{proj}}}}(\boldsymbol{\nu})\|_2^2 \leq \|\boldsymbol{\nu}\|_2^2 \tag{C.40}$$

This inequality holds for any realization of $\widehat{\mathcal{M}}$ and $\widetilde{\mathbf{s}}$. Substituting this into the expression for Term 3:

$$\text{Term 3} \leq \mathbb{E}_{\widehat{\mathcal{M}},\widetilde{\mathbf{s}}}\left[\frac{2}{n}\|\boldsymbol{\nu}\|_2^2\right] \tag{C.41}$$

As $\boldsymbol{\nu}$ is a fixed vector, $\|\boldsymbol{\nu}\|_2^2$ is a constant. Then the expectation operator does not change it:

$$\text{Term 3} \leq \frac{2\|\boldsymbol{\nu}\|_2^2}{n} \tag{C.42}$$

This provides an upper bound for the bias error component, indicating that it scales with the squared norm of the fixed bias vector $\boldsymbol{\nu}$ and decreases with $1/n$. The factor of 2 arises from the use of the inequality $(a + b)^2 \leq 2a^2 + 2b^2$ when separating the zero-mean noise and bias components in the derivation of Lemma 3.4.

**Combined Risk Bound.** Combining the bounds for the three terms:

$$\begin{aligned}
\mathbb{E}[\|\widehat{\mathbf{s}} - \widetilde{\mathbf{s}}\|_2^2] \leq &\mathbb{E}_{\widetilde{\mathbf{s}}}[n\text{Var}(\widetilde{\mathbf{s}})] \cdot \mathbb{E}[Inv(\pi^*, \widetilde{\pi})] \\
&+ \frac{2\widetilde{\sigma}^2 \log(en)}{n} \\
&+ \frac{2\|\boldsymbol{\nu}\|_2^2}{n}
\end{aligned}$$

where $\mathbb{E}[Inv(\pi^*, \widetilde{\pi})]$ is the expected number of inversions produced by the first-stage algorithm $\mathbf{s}^*$ with respect to its true target $\widetilde{\mathbf{s}}$, and can be expressed as:

$$\mathbb{E}[Inv(\pi^*, \widetilde{\pi})] = \sum_{j,k:\widetilde{\mathbf{s}}_j < \widetilde{\mathbf{s}}_k} P(\mathbf{s}_j^* > \mathbf{s}_k^*)$$

with each $P(\mathbf{s}_j^* > \mathbf{s}_k^*)$ bounded using the Chernoff-Cramer bound:

$$P(\mathbf{s}_j^* > \mathbf{s}_k^*) \leq \frac{\sqrt{(\boldsymbol{e}_j - \boldsymbol{e}_k)^\top \Sigma_{\mathbf{s}^*}(\boldsymbol{e}_j - \boldsymbol{e}_k)}}{(\widetilde{\mathbf{s}}_k - \widetilde{\mathbf{s}}_j)\sqrt{2\pi}} \exp\left(-\frac{(\widetilde{\mathbf{s}}_k - \widetilde{\mathbf{s}}_j)^2}{2(\boldsymbol{e}_j - \boldsymbol{e}_k)^\top \Sigma_{\mathbf{s}^*}(\boldsymbol{e}_j - \boldsymbol{e}_k)}\right)$$

and $\Sigma_{\mathbf{s}^*}$ is the asymptotic covariance matrix of $\mathbf{s}^*$ in Lemma 3.1.

**Additional discussion.** The core distinctions remain similar to previous discussions, with the key aspect being that our target for estimation $\widetilde{\mathbf{s}}$ is a noisy version of the underlying ground truth $\mathbf{s}$, our observations $\mathbf{s}_p$ are related to $\widetilde{\mathbf{s}}$ via an effective noise term $\mathcal{N}(\boldsymbol{\nu}, \widetilde{\sigma}^2 I_n)$, and the projection is onto a random cone $\widehat{\mathcal{M}}$. (1) Differs from traditional isotonic regression which assumes a fixed cone and zero-mean noise relative to its direct target. (2) Differs from fixed model misspecification as our cone is random and our effective observation noise has a fixed bias $\boldsymbol{\nu}$ and variance $\widetilde{\sigma}^2 I_n$ relative to the target $\widetilde{\mathbf{s}}$.

# D  PROBABILITY OF ESTIMATION IMPROVEMENT VIA PROJECTION

In this section, we establish a probabilistic bound demonstrating that, under certain conditions related to the accuracy of the estimated cone $\widehat{\mathcal{M}}$, the projection-based estimator $\widehat{\mathbf{s}}$ provides a strictly better estimate of the ground truth $\mathbf{s}$ than the direct observation $\mathbf{s}_p$.

## D.1  PROOF OF COROLLARY 3.6

**Lemma D.1** (Restatement of Corollary 3.6). Let $\mathbf{s}^*$ be the estimate from the first-stage algorithm for its true target $\widetilde{\mathbf{s}}$. For any two items $j, k$ with $\widetilde{\mathbf{s}}_j < \widetilde{\mathbf{s}}_k$, let $\widetilde{\Delta}_{kj} = \widetilde{\mathbf{s}}_k - \widetilde{\mathbf{s}}_j$ be the true score gap for the first-stage target, and let $\sigma_{X_{jk}}^{*2} = (\boldsymbol{e}_j - \boldsymbol{e}_k)^T \Sigma_{\mathbf{s}^*}(\boldsymbol{e}_j - \boldsymbol{e}_k)$ be the variance of the estimated score difference from the first stage, where $\Sigma_{\mathbf{s}^*}$ is the asymptotic covariance of $\mathbf{s}^* - \widetilde{\mathbf{s}}$. Then the probability that the estimated cone $\widehat{\mathcal{M}} = \mathcal{M}_{\pi(\mathbf{s}^*)}$ differs from the true cone for the first-stage target $\widetilde{\mathcal{M}} = \mathcal{M}_{\pi(\widetilde{\mathbf{s}})}$ is bounded by

$$\delta_1 := P(\pi(\mathbf{s}^*) \neq \pi(\widetilde{\mathbf{s}})) \leq \sum_{j,k:\widetilde{\mathbf{s}}_j < \widetilde{\mathbf{s}}_k} \frac{\sigma_{X_{jk}}^*}{\widetilde{\Delta}_{kj}\sqrt{2\pi}} \exp\left(-\frac{(\widetilde{\Delta}_{kj})^2}{2\sigma_{X_{jk}}^{*2}}\right). \tag{D.1}$$

*Proof.* The event $\pi(\mathbf{s}^*) \neq \pi(\widetilde{\mathbf{s}})$ occurs if and only if there exists at least one pair of items $(j, k)$ such that their true order according to $\widetilde{\mathbf{s}}$ is $\widetilde{\mathbf{s}}_j < \widetilde{\mathbf{s}}_k$, but their estimated order is $\mathbf{s}_j^* > \mathbf{s}_k^*$. Using the union bound over all such pairs:

$$P(\pi(\mathbf{s}^*) \neq \pi(\widetilde{\mathbf{s}})) \leq \sum_{j,k:\widetilde{\mathbf{s}}_j < \widetilde{\mathbf{s}}_k} P(\mathbf{s}_j^* > \mathbf{s}_k^*)$$

Let $\eta = \mathbf{s}^* - \widetilde{\mathbf{s}}$. We assume $\eta \sim \mathcal{N}(\mathbf{0}, \Sigma_{\mathbf{s}^*})$. The probability of a single incorrect pairwise ranking is $P(\mathbf{s}_j^* > \mathbf{s}_k^*) = P(\eta_j - \eta_k > \widetilde{\mathbf{s}}_k - \widetilde{\mathbf{s}}_j)$. Let $X_{jk} = \eta_j - \eta_k$. Then $X_{jk} \sim \mathcal{N}(0, \sigma_{X_{jk}}^{*2})$. The probability becomes $P(X_{jk} > \widetilde{\Delta}_{kj})$. Applying the Chernoff-Cramer bound (see Appendix C.3 or a similar result), $P(Z > t) \leq \frac{1}{t\sqrt{2\pi}} e^{-t^2/2}$ for $Z \sim \mathcal{N}(0,1)$ and $t > 0$. Let $t_{jk} = \widetilde{\Delta}_{kj}/\sigma_{X_{jk}}^*$.

$$P(X_{jk} > \widetilde{\Delta}_{kj}) \leq \frac{\sigma_{X_{jk}}^*}{\widetilde{\Delta}_{kj}\sqrt{2\pi}} \exp\left(-\frac{(\widetilde{\Delta}_{kj})^2}{2\sigma_{X_{jk}}^{*2}}\right)$$

Summing over all relevant pairs yields the stated bound for $\delta_1$. $\qquad\square$

## D.2 PROOF OF LEMMA 3.8

**Lemma D.2** (Restatement of Lemma 3.8). Let $\mathbf{s} \in \mathbb{R}^n$ be the ground truth score and $\widetilde{\mathbf{s}} = \mathbf{s} + \widetilde{\boldsymbol{\epsilon}}$, where $\widetilde{\boldsymbol{\epsilon}} \sim \mathcal{N}(\mathbf{0}, \widetilde{\sigma}^2 I_n)$. Let $B^c$ be the event $B^c := \{\pi(\widetilde{\mathbf{s}}) \neq \pi(\mathbf{s})\}$. Then,

$$\delta_2 := P(B^c) \leq \sum_{j,k:\, s_k > s_j} \frac{\widetilde{\sigma}}{(\Delta_{kj}(\mathbf{s}))\sqrt{\pi}} \exp\left(-\frac{(\Delta_{kj}(\mathbf{s}))^2}{4\widetilde{\sigma}^2}\right) \tag{D.2}$$

where $\Delta_{kj}(\mathbf{s}) = \mathbf{s}_k - \mathbf{s}_j$.

*Proof.* The event $B^c \equiv \{\pi(\widetilde{\mathbf{s}}) \neq \pi(\mathbf{s})\}$ occurs if there exists at least one pair of items $(j, k)$ such that their true order according to $\mathbf{s}$ is (without loss of generality) $\mathbf{s}_j < \mathbf{s}_k$, but their order according to $\widetilde{\mathbf{s}}$ is $\widetilde{\mathbf{s}}_j > \widetilde{\mathbf{s}}_k$. Using the union bound over all such pairs $(j, k)$ where $\mathbf{s}_j < \mathbf{s}_k$:

$$P(B^c) \leq \sum_{j,k:\, \mathbf{s}_j < \mathbf{s}_k} P(\widetilde{\mathbf{s}}_j > \widetilde{\mathbf{s}}_k)$$

Consider a single pair $(j, k)$ such that $\mathbf{s}_j < \mathbf{s}_k$. The event $\widetilde{\mathbf{s}}_j > \widetilde{\mathbf{s}}_k$ is equivalent to $(\mathbf{s}_j + \widetilde{\epsilon}_j) > (\mathbf{s}_k + \widetilde{\epsilon}_k)$, which simplifies to:

$$\widetilde{\epsilon}_j - \widetilde{\epsilon}_k > \mathbf{s}_k - \mathbf{s}_j$$

Let $Y_{jk} = \widetilde{\epsilon}_j - \widetilde{\epsilon}_k$. Since $\widetilde{\epsilon}_j$ and $\widetilde{\epsilon}_k$ are independent and identically distributed as $\mathcal{N}(0, \widetilde{\sigma}^2)$, their difference $Y_{jk}$ follows a normal distribution:

$$Y_{jk} \sim \mathcal{N}(0, \mathrm{Var}(\widetilde{\epsilon}_j) + \mathrm{Var}(\widetilde{\epsilon}_k)) = \mathcal{N}(0, 2\widetilde{\sigma}^2)$$

Let $\Delta_{kj}(\mathbf{s}) = \mathbf{s}_k - \mathbf{s}_j$. Since $\mathbf{s}_j < \mathbf{s}_k$, we have $\Delta_{kj}(\mathbf{s}) > 0$. The probability of interest for a single pair is $P(Y_{jk} > \Delta_{kj}(\mathbf{s}))$. Let $Z = Y_{jk}/\sqrt{2\widetilde{\sigma}^2}$. Then $Z \sim \mathcal{N}(0, 1)$.

$$P(Y_{jk} > \Delta_{kj}(\mathbf{s})) = P\left(Z > \frac{\Delta_{kj}(\mathbf{s})}{\sqrt{2\widetilde{\sigma}^2}}\right)$$

Applying the Chernoff-Cramer bound $P(Z > t) \leq \frac{1}{t\sqrt{2\pi}} e^{-t^2/2}$ with $t = \frac{\Delta_{kj}(\mathbf{s})}{\sqrt{2\widetilde{\sigma}^2}}$:

$$P(Y_{jk} > \Delta_{kj}(\mathbf{s})) \leq \frac{\sqrt{2\widetilde{\sigma}^2}}{\Delta_{kj}(\mathbf{s})\sqrt{2\pi}} \exp\left(-\frac{(\Delta_{kj}(\mathbf{s}))^2}{2 \cdot (2\widetilde{\sigma}^2)}\right) \tag{D.3}$$

$$= \frac{\widetilde{\sigma}}{\Delta_{kj}(\mathbf{s})\sqrt{\pi}} \exp\left(-\frac{(\Delta_{kj}(\mathbf{s}))^2}{4\widetilde{\sigma}^2}\right) \tag{D.4}$$

Summing this probability over all pairs $(j, k)$ such that $\mathbf{s}_j < \mathbf{s}_k$ (which is equivalent to summing over pairs $j, k$ such that $\mathbf{s}_k > \mathbf{s}_j$ by swapping indices if necessary, ensuring $\Delta_{kj}(\mathbf{s})$ remains positive) gives the stated bound for $\delta_2$. $\qquad\square$

## D.3 PROOF OF THEOREM 3.9

**Theorem D.3** (Restatement of Theorem 3.9). Let $\delta_1 = P(\pi(\mathbf{s}^*) \neq \pi(\widetilde{\mathbf{s}}))$ be the probability of a ranking error in the first-stage estimation relative to its true target $\widetilde{\mathbf{s}}$ (as defined in Corollary 3.6). Let $\delta_2 = P(\pi(\widetilde{\mathbf{s}}) \neq \pi(\mathbf{s}))$ be the probability that the ranking of the subjective optimal point $\widetilde{\mathbf{s}}$ differs from the ranking of the ground truth $\mathbf{s}$ (as defined in Lemma D.2). Assume that the true ranking for the first-stage target is consistent with the ground truth ranking, i.e., $\pi(\widetilde{\mathbf{s}}) = \pi(\mathbf{s})$. Then, with probability at least $1 - \delta_1 - \delta_2$,

$$\|\widehat{\mathbf{s}} - \mathbf{s}\|_2^2 < \|\mathbf{s}_p - \mathbf{s}\|_2^2, \tag{D.5}$$

provided that $\mathbf{s}_p \notin \mathcal{M}_{\pi(\mathbf{s})}$.

*Proof.* Consider the following events:

- Let $A$ be the event that the estimated cone $\widehat{\mathcal{M}}$ correctly identifies the cone $\mathcal{M}_{\widetilde{\mathbf{s}}}$ derived from the subjective optimal point $\widetilde{\mathbf{s}}$, i.e., $A := \{\widehat{\mathcal{M}} = \mathcal{M}_{\widetilde{\mathbf{s}}}\}$.

- Let $B$ be the event that the cone $\mathcal{M}_{\widetilde{\mathbf{s}}}$ correctly identifies the true cone $\mathcal{M}_{\mathbf{s}}$ derived from the ground truth $\mathbf{s}$, i.e., $B := \{\mathcal{M}_{\widetilde{\mathbf{s}}} = \mathcal{M}_{\mathbf{s}}\}$.

The probability $P(A^c) = P(\widehat{\mathcal{M}} \neq \mathcal{M}_{\widetilde{\mathbf{s}}}) = P(\pi(\mathbf{s}^*) \neq \pi(\widetilde{\mathbf{s}})) = 1 - \delta_1$ Thus, $P(A) = 1 - \delta_1$. And $P(B^c) = P(\mathcal{M}_{\pi(\widetilde{\mathbf{s}})} \neq \mathcal{M}_{\pi(\mathbf{s})}) = P(\pi(\widetilde{\mathbf{s}}) \neq \pi(\mathbf{s})) = \delta_2$, as defined in Lemma D.2. Thus,$P(B) = 1 - \delta_2$. Now we let $C$ be the event that $\widehat{\mathcal{M}} = \mathcal{M}_{\mathbf{s}}$. Event $C$ occurs with probability at least $1 - \delta_1 - \delta_2$.

When event $C$ occurs, since $\mathbf{s}$ defines the cone $\mathcal{M}_{\mathbf{s}}$, it follows that $\mathbf{s} \in \mathcal{M}_{\mathbf{s}}$.Thus,

$$\mathbf{s} \in \widehat{\mathcal{M}}. \tag{D.6}$$

By the fundamental property of projection onto a closed convex set $\widehat{\mathcal{M}}$, for any vector $\boldsymbol{x} \in \widehat{\mathcal{M}}$, we have:

$$\|\mathbf{s}_p - \boldsymbol{x}\|_2^2 \geq \|\mathbf{s}_p - \widehat{\mathbf{s}}\|_2^2 + \|\boldsymbol{x} - \widehat{\mathbf{s}}\|_2^2 \tag{D.7}$$

Then we can set $\boldsymbol{x} = \mathbf{s}$:

$$\|\mathbf{s}_p - \mathbf{s}\|_2^2 \geq \|\mathbf{s}_p - \widehat{\mathbf{s}}\|_2^2 + \|\mathbf{s} - \widehat{\mathbf{s}}\|_2^2 \tag{D.8}$$

This inequality holds when event $C$ occurs. For the strict inequality $\|\mathbf{s} - \widehat{\mathbf{s}}\|_2^2 < \|\mathbf{s}_p - \mathbf{s}\|_2^2$ to hold, we require that the term $\|\mathbf{s}_p - \widehat{\mathbf{s}}\|_2^2$ is strictly positive. This occurs if and only if $\mathbf{s}_p \neq \widehat{\mathbf{s}}$, which is true if $\mathbf{s}_p \notin \widehat{\mathcal{M}}$. Thus, if event $C$ occurs and it is also true that $\mathbf{s}_p \notin \mathcal{M}_{\mathbf{s}}$, then $\|\mathbf{s}_p - \widehat{\mathbf{s}}\|_2^2 > 0$. In this situation, we have:

$$\|\mathbf{s} - \widehat{\mathbf{s}}\|_2^2 < \|\mathbf{s}_p - \mathbf{s}\|_2^2 \tag{D.9}$$

This strict inequality holds when event $C$ occurs and the condition $\mathbf{s}_p \notin \widehat{\mathcal{M}}$ is met. The condition $\mathbf{s}_p \notin \mathcal{M}_{\mathbf{s}}$ means that the biased observation $\mathbf{s} + \boldsymbol{\nu}$ violates the true ordering of $\mathbf{s}$, which is generally true if $\boldsymbol{\nu}$ is non-zero and not perfectly aligned with the cone $\mathcal{M}_{\mathbf{s}}$ in a way that preserves membership. $\qquad\square$

**Remark D.4** (Conditions for Small Error Probabilities $\delta_1$ and $\delta_2$)**.** The overall risk bound depends critically on the magnitudes of $\delta_1$ and $\delta_2$. These probabilities become small under favorable conditions:

- **For $\delta_1$ (First-stage ranking accuracy):** The probability $P(\mathbf{s}_j^* > \mathbf{s}_k^*)$ for $\widetilde{\mathbf{s}}_j < \widetilde{\mathbf{s}}_k$ is small if the ratio $\frac{(\widetilde{\Delta}_{kj})^2}{{\sigma_{X_{jk}}^*}^2}$ is large. This occurs when:

    1. The true gaps $\widetilde{\Delta}_{kj} = \widetilde{\mathbf{s}}_k - \widetilde{\mathbf{s}}_j$ between scores (for the first-stage target $\widetilde{\mathbf{s}}$) are large, making items more distinguishable.
    2. The variance of the score difference estimates ${\sigma_{X_{jk}}^*}^2 = (\boldsymbol{e}_j - \boldsymbol{e}_k)^T \Sigma_{\mathbf{s}^*}(\boldsymbol{e}_j - \boldsymbol{e}_k)$ is small. This implies that the first-stage algorithm provides precise estimates of score differences. The matrix $\Sigma_{\mathbf{s}^*} = \frac{1}{mk}S^+$ indicates that precision increases with more users ($m$) or more comparisons per user ($k$).

    Thus, $\delta_1$ is small if the first-stage estimation is based on sufficient data and the items it aims to rank are well-separated.

- **For $\delta_2$ (Ranking consistency between $\widetilde{\mathbf{s}}$ and its mean $\mathbf{s}$):** The probability $P(\widetilde{\mathbf{s}}_j > \widetilde{\mathbf{s}}_k)$ for $\mathbf{s}_j < \mathbf{s}_k$ is small if the ratio $\frac{(\Delta_{kj}(\mathbf{s}))^2}{4\widetilde{\sigma}^2}$ is large. This occurs when:

    1. The true gaps $\Delta_{kj}(\mathbf{s}) = \mathbf{s}_k - \mathbf{s}_j$ in the ground truth scores $\mathbf{s}$ are large.
    2. The variance $\widetilde{\sigma}^2$ of the subjective noise $\widetilde{\boldsymbol{\epsilon}}$ (which makes $\widetilde{\mathbf{s}}$ deviate from $\mathbf{s}$) is small.

    Thus, $\delta_2$ is small if the ground truth scores $\mathbf{s}$ are well-separated and the process generating the subjective optimal point $\widetilde{\mathbf{s}}$ from $\mathbf{s}$ has low noise.

# E    ADDITIONAL INFORMATION OF EXPERIMENTS

## E.1    DATASETS AND SETTINGS

**Reading Level:** This dataset contains pairwise comparisons of text documents based on reading difficulty. The dataset includes 490 documents with known reading levels, serving as ground truth difficulty scores. A total of 624 annotators provided 12728 pairwise judgments, with each comparison indicating which document is easier to read. The dataset structure includes judge information, judgment outcomes (where "A" indicates document A is easier than document B), document identification numbers, and the corresponding reading levels for each document pair. We construct a semi-synthetic setting by generating pairwise comparisons from an underlying Bradley-Terry/Thurstone model using the true item scores, while adding annotator-specific noise to simulate heterogeneity. This provides us with realistic but controlled pairwise preference data. We then obtain an "oracle" objective score $\mathbf{s}_p$ for each item by perturbing its true score with additional noise or bias, mimicking the output of an imperfect predictive model. This noisy $\mathbf{s}_p$ serves as the initial model assessment for Stage 2 of `AtC`. The availability of ground truth in these tasks allows us to quantitatively evaluate $\mathbf{s}^*$, $\mathbf{s}_p$, and $\widehat{\mathbf{s}}$ under various metrics.

**Dots-activity:** This dataset (Kemmer et al., 2020) contains judgments from 300 participants on 30 distinct images, yielding 8700 pairwise comparisons for estimating dot counts. We transformed this into a human-centered assessment scenario as follows: humans observe **original complete images** to provide comparative judgments, while the objective model (OpenCV's contour detection) processes **corrupted images** with varying degradation levels (global blur, localized blur/noise/whiteout). The task is to estimate dot counts in the **original images**, so ground truth remains unchanged regardless of corruption applied to model inputs. This setup simulates realistic scenarios where human judgment helps calibrate models that receive degraded inputs (e.g., low-quality sensors, occluded views), while humans access complete information through experience or alternative channels. To our knowledge, this represents the first open-source study using real human data in this field; given the scarcity of suitable datasets, this transformation provides our best validation approach on authentic data.

## E.2    RANK AGGREGATION BASELINES

We compare several methods for aggregating the crowdsourced pairwise judgments (Stage-1), including both homogeneous models and heterogeneity-aware models:

- **BTL (Bradley-Terry-Luce):** A classic model for pairwise comparisons that assumes each item $i$ has a latent score $\mathbf{s}_i$ such that the probability of $i$ beating $j$ is $\Pr(i \succ j) = \frac{\exp(\mathbf{s}_i)}{\exp(\mathbf{s}_i) + \exp(\mathbf{s}_j)}$. We fit the scores $s$ by maximum likelihood, given all pairwise preferences. BTL assumes all annotators are identical and noise is homogeneous (every comparison is an independent sample from the same underlying distribution).

- **Thurstone Case V (TCV):** An alternative pairwise comparison model proposed by Thurstone, which assumes differences in item scores are normally distributed. In Case V, $\Pr(i \succ j) = \Phi(\mathbf{s}_i - \mathbf{s}_j)$ where $\Phi$ is the standard normal CDF. Like BTL, the homogeneous TCV model assumes a single common noise level for all annotators. We fit item scores by MLE under this probit-based model. BTL and TCV typically produce similar rankings; TCV can be slightly more robust to outliers in some cases (due to the normal vs. logistic noise assumption).

- **CrowdBT:** A variant of the Bradley-Terry model that incorporates annotator-specific reliability parameters. In CrowdBT (Chen et al., 2013), each annotator $u$ has an associated consistency weight or bias parameter that influences the comparisons they provide. Intuitively, this model down-weights votes from inconsistent annotators and up-weights those from reliable annotators, yielding a better aggregate score $\mathbf{s}^*$. We implement CrowdBT following the approach of (Chen et al., 2013), including a "virtual annotator" regularization strategy to stabilize the reliability estimates.

- **CrowdTCV:** A heterogeneity-aware extension of Thurstone Case V that incorporates annotator-specific precision parameters. Each annotator $u$ is assumed to have their own precision $\gamma_u$ in the Thurstone model, such that $\Pr(u : i \succ j) = \Phi(\gamma_u(\mathbf{s}_i - \mathbf{s}_j))$. This model was introduced by (Jin et al., 2020). We refer to our implementation simply as CrowdTCV. Like CrowdBT, it learns which annotators are more consistent (higher $\gamma_u$) and which are noisier (lower $\gamma_u$), improving the quality of the aggregated $\mathbf{s}^*$.

- **Heterogeneous Rank Aggregation (HRA) variants:** We adopt the heterogeneous rank aggregation framework of Jin et al. (2020), which explicitly models annotator-specific noise in the Bradley-Terry and Thurstone settings. Specifically, in the heterogeneous Bradley-Terry-Luce (HBTL) variant, $\Pr(u : i \succ j) = \sigma(\gamma_u(\mathbf{s}_i - \mathbf{s}_j))$ where $\sigma$ is the logistic CDF, whereas in the heterogeneous Thurstone Case V (HTCV) variant, the logistic is replaced by the normal CDF $\Phi$. Both item scores $\mathbf{s}$ and annotator precisions $\{\gamma_u\}$ are jointly estimated from data via iterative optimization. These methods achieve strong aggregation performance by accounting for individual differences: an annotator with a very small $\gamma_u$ (unreliable) contributes nearly random comparisons, which the model down-weights, whereas an annotator with large $\gamma_u$ is given more influence. We use the term "HRA" to refer to this class of heterogeneity-aware aggregators, with HBTL and HTCV as representative implementations (corresponding to HRA-E and HRA-N in Table 1).

Each aggregation method produces an output consensus score vector $\mathbf{s}^*$ (determined up to an arbitrary scale, which we normalize for evaluation). The $\mathbf{s}^*$'s rank $\widehat{\pi}$ serve as inputs to Stage-2 of `AtC`.

### E.3 EVALUATION METRICS

We evaluate the quality of the predicted scores and rankings using the following metrics. Let $\widehat{\mathbf{s}} = (\widehat{\mathbf{s}}_1, \ldots, \widehat{\mathbf{s}}_n)$ be the scores produced by a given method and $\mathbf{s} = (\mathbf{s}_1, \ldots, \mathbf{s}_n)$ be the ground-truth scores (when available). For distribution-based metrics, let $p$ denote the true distribution of scores (or true outcome values) and $q$ denote the distribution derived from $\widehat{\mathbf{s}}$.

- **Kendall's Tau ($\tau$):** A rank correlation coefficient between the ordering induced by $\widehat{\mathbf{s}}$ and the true ordering. It is defined as $\tau = \dfrac{C - D}{\binom{n}{2}}$, where $C$ is the number of concordant item pairs and $D$ is the number of discordant pairs when comparing the rankings of $\widehat{\mathbf{s}}$ and $\mathbf{s}$. We have $\tau = 1$ if the two rankings are identical, $\tau = 0$ if the rankings are uncorrelated, and $\tau = -1$ if one ranking is the exact reverse of the other. Higher $\tau$ indicates better agreement with the ground-truth ordering.

- **MSE:** The root-mean-squared error of the scores. We compute $\mathrm{MSE} = \sqrt{\dfrac{1}{n}\sum_{i=1}^{n}(\widehat{\mathbf{s}}_i - \mathbf{s}_i)^2}$ . This is the square root of the mean squared error (MSE), expressed in the same units as the scores. A lower L2 error indicates that the predicted scores $\widehat{\mathbf{s}}$ are numerically closer to the true scores $\mathbf{s}$.

- **Wasserstein Distance:** Also known as the Earth Mover's Distance, this measures the distance between two distributions. Treating the set of predicted scores $\{\widehat{\mathbf{s}}_i\}$ and true scores $\{\mathbf{s}_i\}$ as empirical distributions, the Wasserstein-1 distance is defined as $W_1(\widehat{\mathbf{s}}, \mathbf{s}) = \displaystyle\int_{-\infty}^{\infty} \left| F_{\widehat{\mathbf{s}}}(x) - F_{\mathbf{s}}(x) \right| dx$, where $F_{\widehat{\mathbf{s}}}$ and $F_{\mathbf{s}}$ are the cumulative distribution functions (CDFs) of the predicted and true score distributions. In practice, we compute $W_1$ by sorting the scores and finding the area between the two empirical CDF curves. Smaller $W_1$ implies that the distribution of predicted scores is closer to that of the true scores.

- **Kolmogorov–Smirnov (KS) Statistic:** Another measure of distributional discrepancy between $\widehat{\mathbf{s}}$ and $\mathbf{s}$. The KS statistic is $D_{\mathrm{KS}} = \sup_x \left| F_{\widehat{\mathbf{s}}}(x) - F_{\mathbf{s}}(x) \right|$, the maximum absolute difference between the CDF of $\widehat{\mathbf{s}}$ and the CDF of $\mathbf{s}$. This represents the largest gap between the two cumulative distributions. Lower KS values (closer to 0) indicate a better alignment of the predicted score distribution with the true distribution.

- **Kullback–Leibler (KL) Divergence:** For tasks where we compare probability distributions, we use KL divergence to measure how one distribution diverges from another. Let $p = \{p_i\}$ be the true distribution and $q = \{q_i\}$ be the predicted distribution derived from $\widehat{\mathbf{s}}$ (e.g. by normalizing or exponentiating the $\widehat{\mathbf{s}}$ values). The KL divergence of $q$ from $p$ is defined as $D_{\mathrm{KL}}(p \parallel q) = \sum_{i=1}^{n} p_i \log \dfrac{p_i}{q_i}$ . We treat lower KL as better (with $D_{\mathrm{KL}} = 0$ indicating $q$ exactly equals $p$). Note that KL is an asymmetric measure and is undefined if there exists some $i$ with $p_i > 0$ but $q_i = 0$; in our implementation we add a small $\epsilon$ to predicted probabilities to avoid zeros.

**Remark.** An important phenomenon in $\texttt{AtC}$ is that $\hat{\mathbf{s}}$ can achieve higher Kendall's $\tau$ than $\mathbf{s}^*$ despite both inducing the same ranking $\hat{\pi}$. The key is that $\hat{\mathbf{s}}$ often contains **ties** introduced by the Pool-Adjacent-Violators (PAVA) algorithm, which are absent in $\mathbf{s}^*$. When $\mathbf{s}_p$ violates the consensus ranking $\hat{\pi}$, PAVA resolves conflicts by averaging violating scores, creating ties. This improves $\tau$ by converting discordant pairs into tied pairs, which Kendall's $\tau$ penalizes less severely than discordant pairs (Brancotte et al., 2015).

Consider an illustrative example where ground truth is $\mathbf{s} = [10, 20, 30, 40]$ with order A < B < C < D. Stage-1 produces $\mathbf{s}^* = [1, 2, 5, 4]$, inducing the ranking A < B < D < C where items C and D are inverted. The model generates $\mathbf{s}_p = [12, 22, 29, 31]$ with the correct order. PAVA detects the violation at positions C and D, then averages them to produce $\hat{\mathbf{s}} = [12, 22, 30, 30]$. Computing Kendall's $\tau$ against ground truth yields $\tau(\mathbf{s}^*, \mathbf{s}) = (5 - 1)/6 \approx 0.67$ but $\tau(\hat{\mathbf{s}}, \mathbf{s}) = (5 - 0)/6 \approx 0.83$. The tie at positions C and D eliminates the discordant pair, improving correlation despite identical ordinal constraints from Stage-1.

### E.4 ESTIMATED SCORE DISTRIBUTIONS ON READING LEVEL DATASET

This section provides visualizations of the estimated item scores obtained by different methods on the Reading Level dataset. We present violin plots to illustrate the distribution of scores for each item.

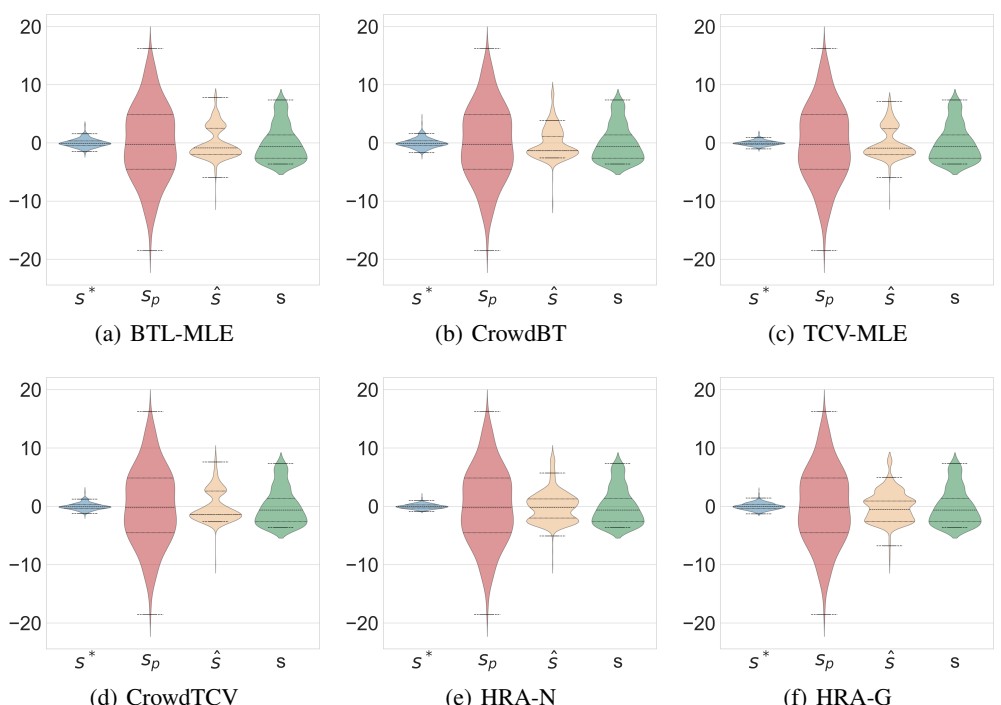

Figure 6: Violin plots of estimated scores for different baseline algorithms.

### E.5 IMPACT OF SCALE INCONSISTENCY ON READING LEVEL DATASET

A common concern in human judgment aggregation is whether scale inconsistency between consensus scores and ground truth inflates distributional divergence metrics. To address this, we conducted experiments scaling $\mathbf{s}^*$ to match the range of ground truth $\mathbf{s}$ using linear transformation:

$$\mathbf{s}^*_{scaled} = \frac{\mathbf{s}^* - \min(\mathbf{s}^*)}{\max(\mathbf{s}^*) - \min(\mathbf{s}^*)} \cdot (\max(\mathbf{s}) - \min(\mathbf{s})) + \min(\mathbf{s}).$$

Table 3 presents results on the Reading Level dataset comparing four variants: unscaled consensus ($\mathbf{s}^*$), scaled consensus ($\mathbf{s}^*_{scaled}$), model predictions ($\mathbf{s}_p$), and $\texttt{AtC}$ output ($\hat{\mathbf{s}}$). The results show that

$\mathbf{s}^*_{scaled}$ substantially improves performance over $\mathbf{s}^*$ (e.g., Wasserstein distance drops from 2.25 to 1.44 for HRA-G), confirming that scale inconsistency is a major issue. However, $\widehat{\mathbf{s}}$ still outperforms $\mathbf{s}^*_{scaled}$ across all metrics and methods, demonstrating that isotonic calibration provides value beyond simple rescaling. This validates that projection onto $\widehat{\mathcal{M}}$'s surface yields superior calibration compared to linear transformations that only address scale but not ordinal alignment with model predictions.

Table 3: Scale Inconsistency Analysis on Reading Level Dataset.
We compare unscaled consensus ($\mathbf{s}^*$), scaled consensus ($\mathbf{s}^*_{scaled}$), model predictions ($\mathbf{s}_p$), and AtC output ($\widehat{\mathbf{s}}$).
**Bold** indicates the best performance among $\mathbf{s}^*_{scaled}$, $\mathbf{s}^*$, $\mathbf{s}_p$, and $\widehat{\mathbf{s}}$. The red denotes optimal performance across all methods.

| Stage-1 Method | Wasserstein↓ $\mathbf{s}^*_{scaled}$/ $\mathbf{s}^*$/ $\mathbf{s}_p$/ $\widehat{\mathbf{s}}$ | KS↓ $\mathbf{s}^*_{scaled}$/ $\mathbf{s}^*$/ $\mathbf{s}_p$/ $\widehat{\mathbf{s}}$ |
|---|---|---|
| HRA-G | 1.44 / 2.25 / 2.83 / **0.84** | 0.31 / 0.50 / 0.30 / **0.16** |
| HRA-E | 1.46 / 2.24 / 2.83 / **0.83** | 0.34 / 0.50 / 0.30 / **0.16** |
| HRA-N | 1.45 / 2.35 / 2.83 / **0.74** | 0.31 / 0.56 / 0.30 / **0.19** |
| BTL-MLE | 1.78 / 2.19 / 2.83 / **0.89** | 0.42 / 0.46 / **0.30** / **0.30** |
| CrowdBT | 1.72 / 2.15 / 2.83 / **0.84** | 0.40 / 0.46 / **0.30** / **0.27** |
| CrowdTCV | 1.76 / 2.27 / 2.83 / **1.02** | 0.42 / 0.51 / 0.30 / **0.31** |
| TCV-MLE | 1.78 / 2.34 / 2.83 / **0.91** | 0.43 / 0.55 / **0.30** / **0.30** |

### E.6 COMPREHENSIVE BASELINE COMPARISON ON REAL-WORLD DATASET

We present two evaluations on the Dots-activity dataset: (1) score-level performance analysis comparing human-only ($\mathbf{s}^*$), model-only ($\mathbf{s}_p$), and AtC ($\widehat{\mathbf{s}}$) assessments across all Stage-1 aggregation methods, and (2) end-to-end comparison against three established human-centered calibration baselines:

- **GPPL** (Chu & Ghahramani, 2005): Gaussian Process Preference Learning models pairwise preferences using GP regression with specialized covariance functions, learning latent utility functions from comparative judgments.
- **Rank-SVM** (Joachims, 2002): Support Vector Machine adaptation for ranking problems that learns from pairwise preferences by optimizing margin-based objectives, commonly used in information retrieval and clickthrough data analysis.
- **BARCW** (Li et al., 2022): Bayesian Approach for Ranking with Comparison Weights, a recent method designed to handle inconsistent preferences by modeling annotator-specific biases and reliabilities using Gaussian processes.

Table 4 demonstrates that AtC consistently outperforms both $\mathbf{s}_p$ and $\mathbf{s}^*$ across all seven aggregation methods. While the three baseline methods achieve competitive ranking accuracy (Kendall's $\tau \approx$ 0.92–0.94), they exhibit catastrophic failures in cardinal calibration: MSE deteriorates by 400–450× and Wasserstein distance by 24–26× compared to AtC. This stark contrast validates AtC's unique strength—simultaneously achieving accurate absolute score calibration and maintaining ordinal consistency with human consensus, a capability not demonstrated by existing human-centered assessment approaches. Moreover, our work is also related to methods that integrate human and machine decision-making to improve prediction quality (Li et al., 2025; Yang et al., 2025; Xie et al., 2025a).

### E.7 ROBUSTNESS RESULTS FOR ALL BASELINE METHODS ON REAL-WORLD DATASET.

This section provides supplementary results to complement the main analysis presented in Section 4. We present the complete set of radar plots for the six additional baseline aggregation methods: HRA-G, HRA-N, BTL, TCV, CrowdBT, and CrowdTCV. The experimental methodology, including the four types of image corruption and the evaluation metric (Kendall's Tau), is identical to the one used for the HRA-E analysis in the main body of the paper.

As illustrated in Figures 7 8 9 10 11 12, a consistent trend emerges across all baselines. The primary finding from the main text is strongly reinforced: the AtC framework, when calibrated

Table 4: Real-world results on Dots dataset.
**Bold** indicates the best performance among human-only assessment ($\mathbf{s}^*$), model-only assessment ($\mathbf{s}_p$), and AtC assessment ($\widehat{\mathbf{s}}$), respectively. The red denotes the optimal performance across all methods for the given metric.

| | Stage-1 Method | Kendall $\tau\uparrow$ $\mathbf{s}^*/\mathbf{s}_p/\widehat{\mathbf{s}}$ | Wasserstein$\downarrow$ $\mathbf{s}^*/\mathbf{s}_p/\widehat{\mathbf{s}}$ | KL$\downarrow$ $\mathbf{s}^*/\mathbf{s}_p/\widehat{\mathbf{s}}$ | MSE$\downarrow$ $\mathbf{s}^*/\mathbf{s}_p/\widehat{\mathbf{s}}$ |
|---|---|---|---|---|---|
| AtC | HRA-G | 0.917 / 0.923 / **0.940** | 6.97 / 2.53 / **2.53** | 123.13 / 0.881 / **0.860** | 65.08 / 11.75 / **9.61** |
| | HRA-E | 0.922 / 0.922 / **0.943** | 6.98 / 2.53 / **2.53** | 123.16 / 0.881 / **0.861** | 65.16 / 11.75 / **9.59** |
| | HRA-N | 0.917 / 0.923 / **0.934** | 7.05 / 2.53 / **2.53** | 125.50 / 0.881 / **0.859** | 66.44 / 11.75 / **9.75** |
| | CrowdBT | 0.894 / 0.923 / **0.927** | 5.32 / **2.53** / 2.55 | 72.84 / 0.881 / **0.865** | 39.28 / 11.75 / **10.24** |
| | CrowdTCV | 0.899 / 0.923 / **0.926** | 5.73 / **2.53** / 2.55 | 84.74 / 0.881 / **0.865** | 44.97 / 11.75 / **10.24** |
| | BTL | 0.917 / 0.923 / **0.937** | 6.72 / 2.53 / **2.53** | 115.37 / 0.881 / **0.860** | 60.51 / 11.75 / **9.73** |
| | TCV | 0.917 / 0.923 / **0.933** | 6.81 / **2.53** / 2.54 | 117.81 / 0.881 / **0.878** | 62.09 / 11.75 / **10.18** |
| GPPL | | − / − / 0.931 | − / − / 64.50 | − / − / 126.62 | − / − / 4220.36 |
| Rank-SVM | | − / − / 0.923 | − / − / 61.20 | − / − / 133.70 | − / − / 3814.49 |
| BARCW | | − / − / 0.940 | − / − / 64.62 | − / − / 24.09 | − / − / 4236.16 |

using aggregated rankings ($\widehat{\mathbf{s}}$ (Rank)), demonstrates superior robustness compared to all alternatives. In most cases, the performance of $\widehat{\mathbf{s}}$ (Rank) remains high and degrades gracefully, while the raw objective model scores ($\mathbf{s}_p$) are significantly more vulnerable to input corruptions. This consistent outperformance across a diverse set of aggregation algorithms underscores the general applicability and reliability of our proposed ranking-based calibration strategy.

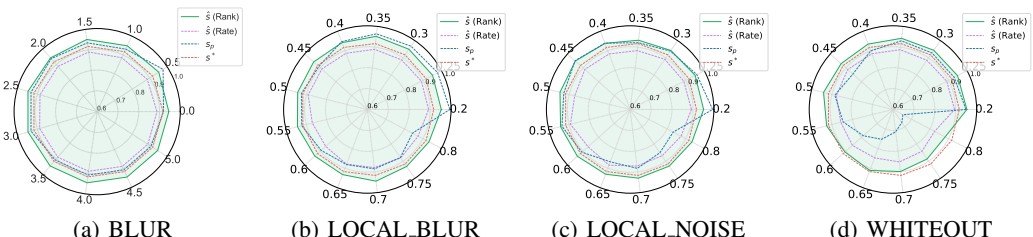

    (a) BLUR      (b) LOCAL_BLUR      (c) LOCAL_NOISE      (d) WHITEOUT

Figure 7: Radar plots under different noise conditions (HRA-G).

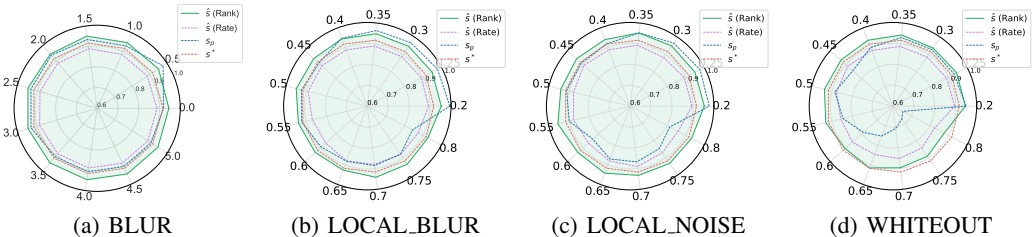

    (a) BLUR      (b) LOCAL_BLUR      (c) LOCAL_NOISE      (d) WHITEOUT

Figure 8: Radar plots under different noise conditions (HRA-N).

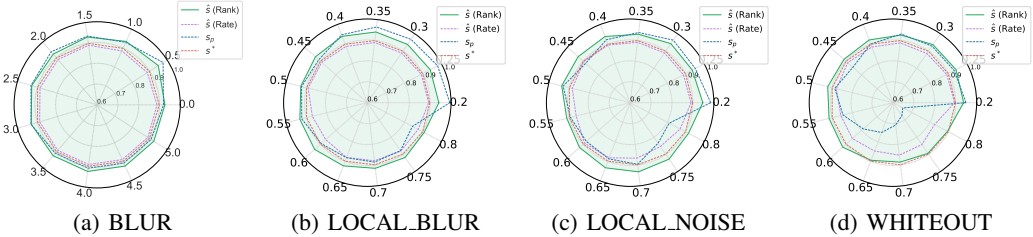

Figure 9: Radar plots under different noise conditions (CrowdTCV).

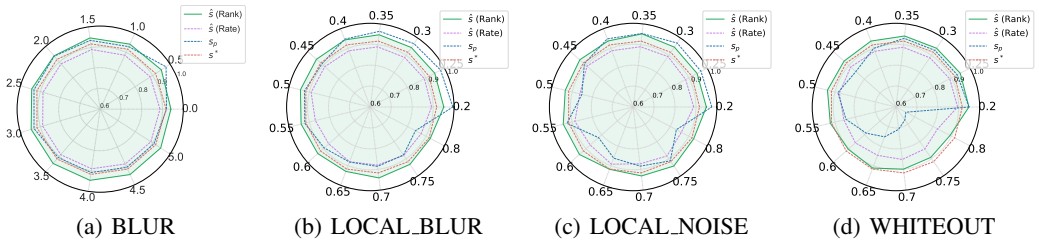

Figure 10: Radar plots under different noise conditions (BTL).

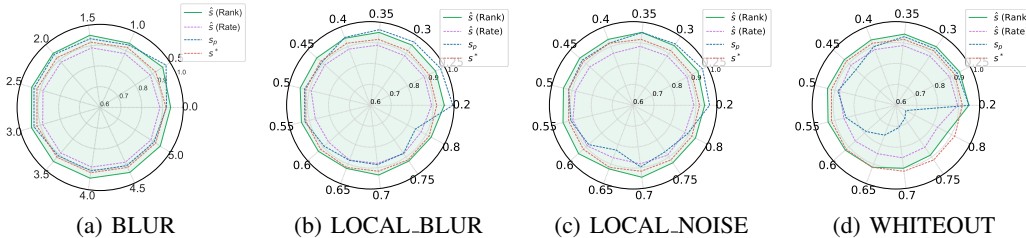

Figure 11: Radar plots under different noise conditions (TCV).

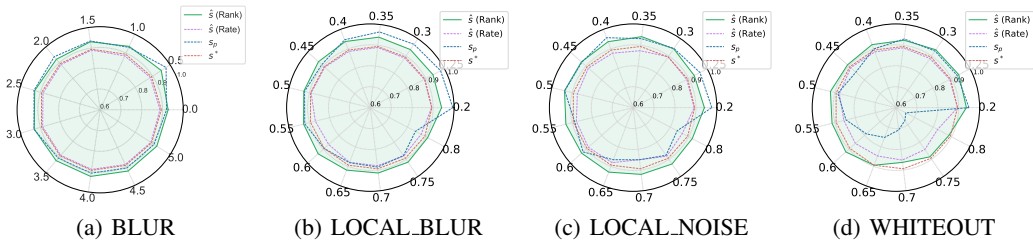

Figure 12: Radar plots under different noise conditions (CrowdBT).

