# OpenReview forum: "AtC: Aggregate-then-Calibrate for Human-centered Assessment"
_ICLR.cc/2026/Conference — ICLR 2026 Poster_

### Official Review · Reviewer_hr8J · 2025-10-31

**Soundness:** 3
**Presentation:** 2
**Contribution:** 2
**Rating:** 6
**Confidence:** 2

**Summary:**

This paper proposes a two-stage framework, Aggregate-then-Calibrate (AtC), for human-centered assessment tasks that lack verifiable ground truth. The authors provide theoretical results for supporting the modeling-selection, robustness, optimality.  Empirical evaluation on semi-synthetic and real datasets suggests that AtC improves robustness and accuracy over human-only or model-only baselines.

**Strengths:**

1. The paper focuses on an important research question—how to calibrate model's score for human-centered assessment tasks.
2. The interdisciplinary considerations are insightful, for example, the design of HTM under the consideration of the Weber–Fechner laws.
3. The theoretical analysis is sound under the modeling assumptions, and the framework performs consistently across multiple datasets, demonstrating its sound performance.

**Weaknesses:**

1. The paper lacks discussion of prior calibration works. The authors should better clarify how AtC differs from existing human-centered calibration approaches.
2. The methods' effectiveness depends on several strict assumptions, which may limit its general applicability.
3. Some key design choices in this paper lack validation through real human studies — for example, whether the HTM-inferred “consensus ranking” truly corresponds to actual human agreement. Moreover, it remains unclear whether the proposed AtC framework ultimately enhances decision-making performance in real-world applications.
4. The score-level performance comparison (Table 1) is conducted only on semi-synthetic experiments, which limits its generalizability. Moreover, the calibration baselines  include only human-only, model-only, and AtC variants with different stage-1 aggregation methods, which do not constitute rigorous baselines. More appropriate comparisons would involve other human-centered score calibration approaches, rather than what appear to be ablation-style baselines.

**Questions:**

The isotonic projection enforces the model’s scores to align with the human consensus ranking, implicitly assuming that this consensus ranking is correct. However, the model’s role is also to correct systematic errors in human ranking outcomes. How do you view the balance between improving objective correctness and maintaining adherence to human-centered consensus in the calibration process?

I especially feel that real human studies are important in this research area.

---

> ### Author Response · Authors · 2025-11-22
> **Response to Weaknesses and Questions (1/2)**
>
> Thank you for recognizing the importance of our research question, design, and theoretical analysis, as well as the evaluation performance. We have added experiments per your suggestions and addressed all concerns below. These updates have been incorporated into the revised manuscript.
>
> ---
>
> ### W1: Discussion of Prior Calibration Works
>
> Lines 498-522 provide a detailed discussion of existing calibration and human-ML complementarity work, explicitly contrasting our conceptual, design, and theoretical innovations. We have moved this content from the appendix to the main text in the revision.
>
> ---
>
> ### W2: Clarification of Modeling Assumptions
>
> Our assumptions can be summarized at three levels:
>
> 1. Conceptual Assumptions: This psychological foundation motivates using ordinal human judgments other than absolute scoring. We validate this assumption through RQ5 (Line 469-474) in the evaluation section.
>
> 2. Modeling Assumptions: (i) Distributional assumptions in the Heterogeneous Thurstone Model for data generation, and (ii) Independence assumptions for pairwise comparisons.
> We acknowledge these constraints on generalizability. However, similar distributional assumptions are standard in virtually all judgment/preference aggregation work (including our baselines)[1,2,3,4]. We validated robustness by testing three noise distributions (Gumbel, Normal, Exponential in HRA-G/N/E), all confirming effectiveness.
>
> 3. Theoretical Assumptions: Some conditions for Theorems 3.3, 3.5, and 3.9 (e.g., conditions in [5] for Theorem 3.3). These are standard in related theoretical work, as mentioned in the Related Work section [6,7,8].
>
> We welcome any specific questions about particular assumptions.
>
> [1] Pairwise ranking aggregation in a crowdsourced setting. In Proceedings of the sixth ACM international conference on Web search and data mining (pp. 193-202).
>
> [2] Mixtures of distance-based models for ranking data. Computational statistics & data analysis, 41(3-4), 645-655.
>
> [3] Maximum likelihood estimation of observer error‐rates using the EM algorithm. Journal of the Royal Statistical Society: Series C (Applied Statistics), 28(1), 20-28.
>
> [4] You are the best reviewer of your own papers: An owner-assisted scoring mechanism. Advances in Neural Information Processing Systems, 34, 27929-27939.
>
> [5] Halbert White. Maximum likelihood estimation of misspecified models. Econometrica: Journal of the econometric society, pp. 1–25, 1982.
>
> [6] On risk bounds in isotonic and other shape restricted regression problems. The Annals of Statistics, 1774-1800.
>
> [7] Sharp oracle inequalities for least squares estimators in shape restricted regression. The Annals of Statistics, 46(2), 745-780.
>
> [8] Risk bounds in isotonic regression. The Annals of Statistics, 30(2):528 – 555, 2002. doi: 10.1214/aos/1021379864.
>
> ---
>
> ### W3&Q: Validation Through Real Human Studies
>
> The real-world dots dataset represents a genuine human study, and RQ6 (Lines 460-475) specifically evaluates this. Since we focus on **system-level consensus judgments** (not individual preferences), we measure whether consensus ranking aligns with ground truth via Kendall's $\tau$ (Table 1 and our new real-world results Table 2 in revised version).
>
> **To our knowledge, our dots dataset experiment is the first open-source study using real human data in the related field.** Whether HTM-inferred consensus matches individual human agreement is not our current focus; our goal is accurate ground truth estimation for decision-making. We fully agree that this is an important direction and look forward to more suitable datasets enabling larger-scale future research through volunteer/crowdsourcing studies.

---

> ### Author Response · Authors · 2025-11-22
> **Response to Weaknesses and Questions (2/2)**
>
> ### W4: Additional Real-World Experiments and Rigorous Baselines
>
> We added two experiments: (1) score-level analysis on the real-world dots dataset, and (2) comparison against the most relevant human-centered calibration baselines we identified: **GPPL**[1], **Rank-SVM**[2], and **BARCW**[3].
>
> || Stage-1 Method | Kendall's $\tau$ | MSE | Wasserstein | KL |
> |--------|--------|----------|-----|-------------|-----|
> |**AtC**| | $\hat{s}$ / $s_p$ / $s^{\*}$ | $\hat{s}$ / $s_p$ / $s^{\*}$ | $\hat{s}$ / $s_p$ / $s^{\*}$ | $\hat{s}$ / $s_p$ / $s^{\*}$ |
> || **HRA-G** | **0.940** / 0.923 / 0.917 | **9.61** / 11.75 / 65.08 | **2.53** / 2.53 / 6.97 | **0.860** / 0.881 / 123.13 |
> || **HRA-E** | **0.943** / 0.923 / 0.922 | **9.59** / 11.75 / 65.16 | **2.53** / 2.53 / 6.98 | **0.861** / 0.881 / 123.16 |
> || **HRA-N** | **0.934** / 0.923 / 0.917 | **9.75** / 11.75 / 39.28 | **2.53** / 2.53 / 7.05 | **0.859** / 0.881 / 125.50 |
> || **CrowdBT** | **0.927** / 0.923 / 0.894 | **10.24** / 11.75 / 60.51 | 2.55 / **2.53** / 5.32 | **0.864** / 0.881 / 72.84 |
> || **BTL** | **0.937** / 0.923 / 0.917 | **9.73** / 11.75 / 44.97 | **2.53** / 2.53 / 6.72 | **0.860** / 0.881 / 115.37 |
> || **CrowdTCV** | **0.926** / 0.923 / 0.899 | **10.24** / 11.75 / 66.44 | 2.55 / **2.53** / 5.73 | **0.865** / 0.881 / 84.74 |
> || **TCV** | **0.933** / 0.923 / 0.917 | **10.18** / 11.75 / 62.09 | 2.54 / **2.53** / 6.81 | **0.878** / 0.881 / 117.81 |
> | **GPPL** || 0.931 / - / - | 4220.36 / - / - | 64.50 / - / - | 126.62 / - / - |
> | **Rank-SVM** || 0.923 / - / - | 3814.49 / - / - | 61.20 / - / - | 133.70 / - / - |
> | **BARCW** || 0.940 / - / - | 4236.16 / - / - | 64.62 / - / - | 24.09 / - / - |
>
> **Analysis**: AtC consistently outperforms $s_p$ and $s^{\*}$ across all aggregation methods. The three human-centered calibration baselines achieve comparable ranking accuracy ($\tau \approx 0.92-0.94$) but catastrophically fail on absolute error (MSE 400-450× worse than AtC) and distributional metrics (Wasserstein 24-26× worse). This validates AtC's unique strength: achieving accurate cardinal calibration while maintaining ordinal consistency.
>
> [1] Preference learning with Gaussian processes. In Proceedings of the 22nd international conference on Machine learning. 137–144.
>
> [2] Optimizing search engines using clickthrough data. In Proceedings of the eighth ACM SIGKDD international conference on Knowledge discovery and data mining.133–142.
>
> [3] Bayesian Analysis of Rank Data with Covariates and Heterogeneous Rankers, Statistical Science, Statist. Sci. 37(1), 1-23.
>
> ---
>
> ### Q: Balance Between Objective Correctness and Human Consensus
>
> Here may be a misunderstanding. AtC does not assume human consensus ranking is correct. It just uses the ranking as an ordinal constraint to calibrate potentially biased model scores $s_p$.
>
> Our framework recognizes:
> - Human consensus may contain ranking errors (captured by $\delta_1$ in Corollary 3.6).
> - Model predictions $s_p$ approximate ground truth but may violate human-derived order.
>
> The calibration balances **both** by finding $\hat{s}$ closest to $s_p$ (preserving the model's quantitative information) while respecting consensus order (incorporating collective human judgment). Theorem 3.5 shows this calibration remains robust even when consensus ranking is misspecified. The goal is not blind adherence to human consensus, but leveraging human ordinal structure to correct model's systematic errors.
>
> ---
>
> **We hope the substantial analytical and experimental results in our work will be taken into full consideration. Thank you very much for your thoughtful review.**

---

### Official Review · Reviewer_2TdT · 2025-10-31

**Soundness:** 3
**Presentation:** 2
**Contribution:** 3
**Rating:** 8
**Confidence:** 3

**Summary:**

For certain tasks, obtaining the ground-truth is costly, unobservable, or only gets clear in the future. This is the case for tasks such as estimating paper quality (only find out in the future after finding out its impact) or worker workload (though workers are asked to use the same scale, how the scale is applied is up to the individual).

This paper tackles the limitations of current practices that address such tasks by proposing the Aggregate-then-Calibrate framework (AtC). Aggregating over individual judgments that lead to proxy labels for the actual task does not allow the model to learn the actual features that are relevant for correct decision making. AtC combines both human and model judgments through two steps:
1. Aggregation step: They yield an ordinal consensus ranking of items from multiple annotators, using a heterogeneous rank aggregation approach.
2. Calibration step: The model’s initial predictions are calibrated using the consensus ranking through isotonic regression.

The paper first provides theoretical guarantees for the framework:
1. efficient in consensus w.r.t. annotator heterogeneity than homogeneous
2. robust under model misspecification
3. gives calibrated output that is more accurate than the uncalibrated model’s output

The authors evaluate the method on two datasets:
1. semi-synthetic: pairwise comparison reading difficulty level dataset where ground-truth scores are available but the annotators are synthetic
2. real-world: ranking and rating-based dots counting in an image, where the image can be obscured and made noisier

Results show that the framework outperforms baselines, especially when facing more noisy data.

**Strengths:**

1. The problem definition is novel, clear, and covers two aspects that heavily interact with each other but has (to the best of my knowledge) been rather understudied: unobservability of the ground truth that is estimated through noisy proxy labels, where humans can be inconsistent in their judgments. The latter adds another layer of complexity, which makes it an important task type to look at.
2. The authors provide theoretical guarantees for the framework, which then highlights the strength of the setup.
3. I also like the evaluation setup which is clearly mapped out with the 6 research questions. This helps the reader in getting a direct overview of what exactly will be evaluated and why. In addition, this helps in understanding the strengths of the AtC framework. The experiments also cover both a (semi-)synthetic setting and real-world setting.

**Weaknesses:**

1. I feel as though subjectivity can be a discussion point: I particularly view the inconsistency in how people use these rating scales as something subjectivee (e.g., people experience workloads differently) and the dots dataset also suffers from input ambiguity which leads to subjectivity.
2. A high-level figure of the entire pipeline would be helpful, Figure 1 is nice but the paper would really benefit from an extra figure that shows an example input and then how AtC leads to a better solution. This is especially helpful since there are so many intermediate steps.
3. Though the paper is nicely structured, I feel the reader can be taken a bit more by the hand, since the paper covers a lot of steps. I think an extra figure already helps with this. E.g., choice of evaluation metrics could be motivated a bit more, though I understand why, I think the reader can be taken a bit more by the hand here.
4. The conclusion and discussion can benefit from a clearer takeaway: in which scenarios would you suggest that a researcher or practitioner uses AtC?
5. Related Work is in the Appendix, which makes it difficult for the reader to adequately place the paper along existing papers. Some decision choices that I questioned before were only cleared up once I saw the Related Work in the Appendix. I understand that there are space constraints but it would be very helpful if at least the important papers are in the main body of the paper.

**Questions:**

Did you also control or verify whether the added noise does not to an extent actually change the ground-truth? I think as you add more obscurities, the actual number of dots could change. I would love to hear your thoughts on this or if I misunderstood any detail.

---

> ### Author Response · Authors · 2025-11-22
>
> Thank you for recognizing our novel problem definition, comprehensive theoretical guarantees, and clear experimental design with well-structured research questions. We particularly appreciate your specific suggestions for improving presentation, which we have integrated into the revised manuscript. Below are our responses to your concerns:
>
> ---
>
> ### W1: Discussion on Subjectivity and Annotator Inconsistency
>
> We appreciate this observation and welcome the opportunity to clarify our conceptual framework. The "inconsistency" we address is annotator heterogeneity in expertise, not fundamental subjectivity about preferences.
> In the example of the Delivery platform (Line 36), when different couriers judge the same delivery route's difficulty, their assessments differ; this is observed as an inconsistency. We don't "filter out" novice opinions. Instead, HTM learns each annotator's reliability from data and applies optimal weighting. This produces a consensus closer to ground truth (Theorem 3.3).
>
> **Task Distinction**: We focus on judgment tasks (objective ground truth exists but is unobservable, e.g., true workload, future paper impact) rather than preference tasks (no ground truth, e.g., favorite flavor). While novice and expert opinions are subjectively "legitimate," Theorems 3.3, 3.5, and 3.9 show expert judgments provide lower-variance, more robust estimators closer to ground truth.
>
> For the dots dataset, input ambiguity from corruption doesn't introduce subjectivity: humans still judge the objective number of dots in the original image, while models process degraded inputs.
>
> We are happy to discuss any specific aspects further if helpful.
>
> ---
>
> ### W2 & W3 & W5: Figure, Related Work, and Metric Clarity
>
> Thank you for this suggestion. We have added an enhanced Figure 1 with an detailed example, moved related work to the main pages, and provided clearer metric motivation in the main text:
> - Kendall's $\tau$: Measures ranking accuracy (ordinal correctness).
> - MSE: Measures absolute scoring error (cardinal accuracy).
> - Wasserstein & KS: Measure distributional alignment with ground truth.
>
> ---
>
> ### W4: When Should Practitioners Use AtC?
>
> AtC suits all human-centered assessment tasks where model-only evaluation faces limitations due to unobservable/costly ground truth, and human comparative judgments can provide ordinal constraints.
>
> Examples:
> - Human-centered assessment (Dots dataset): If models accessed original images, traditional CV tools suffice. When models receive corrupted inputs but humans know the ground truth, human judgments calibrate model predictions.
>
> - Human input is more cost-effective (Delivery platform, Line 36): Collecting courier pairwise route comparisons costs far less than deploying wearables for energy tracking at scale, making AtC practical for large-scale deployment.
>
> We have added this discussion to the Conclusion section.
>
> ---
>
> ### Question: Ground Truth Changes with Noise in Dots Dataset?
>
> Our experimental design assumes humans observe **original complete images** while models process **corrupted images**. The task is to estimate dot counts in original images, so the ground truth remains unchanged regardless of noise added to the model inputs.
>
> **Note**: This differs from the dataset's original use in [1]. We cleverly transformed it into a human-centered assessment scenario where model limitations necessitate human calibration. Given the scarcity of open real-world datasets in this domain, this represents our best validation approach on authentic data. We have clarified this setup in the revised version.
>
> [1] Enhancing Collective Estimates by Aggregating Cardinal and Ordinal Inputs. HCOMP, 2020.

---

### Official Review · Reviewer_Arz1 · 2025-11-03

**Soundness:** 3
**Presentation:** 3
**Contribution:** 3
**Rating:** 8
**Confidence:** 2

**Summary:**

This work introduces a notion of calibration and a technique to calibrate predicted model scores to fare well along this notion.
The paper is concerned with settings where human judgments is assumed to be noisy, but where there's assumed value in the aggregated information they provide about relative merit of alternative options.

The paper fits a statistical model for the aggregation of heterogeneous data, and this provides us with a permutation of the judged items (say, in ascending order of merit). The calibration technique involves projecting predicted model scores to the manifold of score vectors whose coordinates are sorted in compliance with the ranking induced by aggregation of human ratings. The paper proposes to use l2 projection.

The paper establishes a number of theoretical results and tests the proposed approach in meaningful settings, shedding light onto the significance of the theoretical results.

I find the paper rather clear, and a very interesting read (admittedly, however, I am not knowledgeable of the specific literature it talks to, so my review is that of an educated but non-expert reader),

**Strengths:**

1. The technique is well-motivated and rather simple
2. The paper analysis the technique on theoretical and empirical grounds, the experiments cover all claims (incl. analysis of the theoretical claims)
3. The paper is well-written and I found it rather clear

**Weaknesses:**

I don't see any major weaknesses, but I have a minor question (for the other box).

**Questions:**

What is the significance of the Euclidean projection? Or, asking differently, what happens if we project differently? Are there any interesting results (theoretical or of computational efficiency) that depend/vary as a function of this decision?

---

> ### Author Response · Authors · 2025-11-22
>
> Thank you for recognizing the motivation, theory, evaluation, and clarity of our work. We greatly appreciate your positive assessment. Below is our response to your question regarding the choice of Euclidean projection. We have incorporated into the 'Related Work' section.
>
> ---
> ### Q: What happens if we project differently?
>
> Among the norms commonly used in isotonic regression ($L_0$, $L_1$, $L_2$, $L_\infty$), we chose ​$L_2$ for strong theoretical, computational, and practical reasons.
>
> 1. Conceptually, let's briefly define the different $L_p$ norms in the context of our problem, where we seek to minimize the distance between the calibrated scores $y$ and the model's raw scores $s_p$:
> - $L_0$: $\|\|y - s_p\|\|\_0 = \sum_{i=1}^n \mathbb{I}(y_i \neq s_{p,i})$. This counts the number of points that are changed.
> Technically, $L_0$ is a pseudo-norm, and its objective is to minimize the number of modified points. This frames the task as a combinatorial optimization problem rather than one of geometric distance.
> - $L_1$: $\|\|y - s_p\|\|\_1 = \sum_{i=1}^n |y_i - s_{p,i}|$. It minimizes the sum of absolute errors.
> - $L_2$ (Euclidean): $\|\|y - s_p\|\|\_2 = \sqrt{\sum_{i=1}^n (y_i - s_{p,i})^2}$. It minimizes the sum of squared errors.
> - $L_\infty$: $\|\|y - s_p\|\|\_\infty = \max\_{i} |y_i - s_{p,i}|$. It minimizes the maximum individual error.
>
> Among these, the $L_2$ norm is the most common and standard choice for isotonic regression [2, page 1], which we adopted for strong theoretical and practical reasons.
>
>
> 2. Theoretically,
> - Our framework assumes Gaussian noise. Minimizing the $L_2$ norm is equivalent to the Maximum Likelihood Estimate (MLE) under this assumption, making it the most statistically consistent choice for our model.
> - $L_2$ norm is a strictly convex function, which guarantees a unique solution for the regression values. This is critical for an assessment system, which can return a single, deterministic score for each item. In contrast, the $L_1$ and $L_\infty$ norms are convex but not strictly convex, and their regression values are not necessarily unique [1, page 1]. Intuitively, the level sets of the $L_2$ norm are smooth spheres, which can only touch a convex constraint set at a single point. The level sets of $L_1$ (diamonds) and $L_\infty$ (squares), however, have "flat" sides and "corners," allowing them to align with the constraint set along an entire edge or face, resulting in multiple optimal solutions [3, page 145].
>
>
> 3. Computationally, for the linear order defined by our consensus ranking $\pi$, $L_2$ isotonic regression can be solved in optimal linear time, $\Theta(n)$, via the Pool-Adjacent-Violators algorithm (PAVA). While for a linear order, the complexity of the corresponding algorithms for $L_1$ and $L_\infty$ is typically $\Theta(n \log n)$ [2, page 12]. This performance difference arises because the $L_2$ solution's averaging property allows for a simple, greedy "pooling" strategy (PAVA). The median-finding nature of $L_1$ or the minimax structure of $L_\infty$ does not permit such a direct greedy approach, often requiring more complex data structures or iterative methods.
>
>
> [1] Stout Q F. Algorithms for L∞ isotonic regression[J]. 2011.
>
> [2] Stout Q F. $ L_p $ Isotonic Regression Algorithms Using an $ L_0 $ Approach[J]. arXiv preprint arXiv:2107.00251, 2021.
>
> [3] Jordan M, Kleinberg J, Schoelkopf B. Information science and statistics[J]. No Title, 2006.(P145, Fig 3.3)

---

### Official Review · Reviewer_Q8gb · 2025-11-04

**Soundness:** 2
**Presentation:** 3
**Contribution:** 2
**Rating:** 4
**Confidence:** 2

**Summary:**

This paper proposes an Aggregate-then-Calibrate (AtC) framework to estimate latent true scores when only noisy human judgments and biased model scores are available. The paper provides theoretical results, including an optimality guarantee showing that integrating human ordinal judgments with model-based scores improves accuracy. Experiments on two datasets show that AtC consistently outperforms human-only and model-only baselines.

**Strengths:**

- The paper is generally well written and easy to follow.

- AtC is a lightweight approach with theoretical guarantees, improving model scores by leveraging human labels.

**Weaknesses:**

- [Q1] The motivation is somewhat unclear. The adjusted score $\hat s$ preserves the same order as the estimated consensus score vector $s^∗$; that is, $\hat \pi$ is consistent between Stage-1 and Stage-2. This makes the need to compute $\hat s$ questionable. Please provide concrete use cases where the adjusted score is required while the estimated consensus score alone is insufficient.

- There are also some issues regarding the experiment.
  - [Q2] The Kendall $\tau$ is a rank correlation. If values in $s^*$ and $\hat s$ induce the same order, they should yield the same correlation wrt $s$. Am I missing something about how \tau  is computed here?
  - [Q3] In Figure 2, the $s^∗$ distribution appears standardized to zero mean and unit variance, which can inflate the distance between $s^∗$ and $s$. Would using a different scaling (for example, matching mean and variance via a simple estimate) make the distributions more comparable?
  - [Q4] As in Q1, across both datasets it remains unclear when the adjusted score is practically useful beyond the consensus outputs. Examples would help.

**Questions:**

See weaknesses

---

> ### Author Response · Authors · 2025-11-22
>
> Thank you for recognizing the clear presentation and theoretical contributions of our paper. We appreciate your constructive feedback. Below are our complete responses to your questions. We have incorporated into the revised manuscript.
>
> ---
>
> ### Q1 & Q4: Why is the adjusted score $\hat{s}$ necessary when it preserves the same order as $s^{\*}$? Please provide concrete examples.
>
> **Direct Answer**: The adjusted score $\hat{s}$ is essential because rankings alone are insufficient for systematic decision-making that requires cardinal/absolute values, not just ordinal comparisons. While $s^{*}$ and $\hat{s}$ share the same ordering $\hat{\pi}$, they differ critically in scale and calibration.
>
> Below are specific explanations:
>
> 1. Conceptually, $s^{\*}$ and $\hat{s}$ represent fundamentally different points in the constraint space $\hat{\mathcal{M}}$: (i) $s^{\*}$ is an interior point within $\hat{\mathcal{M}}$ derived solely from Stage-1 aggregation, and (ii) $\hat{s}$ is the projection of $s_p$ onto $\hat{\mathcal{M}}$'s surface.
> When the ground truth $s$ lies outside $\hat{\mathcal{M}}$ (due to Stage-1 ranking errors), if $s_p$ reasonably approximates $s$, then $\hat{s}$ has a higher probability of being closer to $s$ than $s^\*$ is. Theorem 3.5 in Section 3.2 formally quantifies this error decomposition. Additionally, Algorithm 2 helps mitigate these errors (detailed in our Q2 response).
>
> 2. Empirically, the ranking $\hat{\pi}$ is only one input to Algorithm 2; the objective model score $s_p$ is equally critical for scale consistency. Table 1 demonstrates that $\hat{s}$ consistently outperforms $s^{\*}$ across absolute error (MSE) and distributional metrics (KS, Wasserstein). Following your suggestion, we added experiments with scaled $s^{\*}$, and $\hat{s}$ maintains superior performance (see our Q3 response).
>
> 3. Examples (Q4):
> - **Dots Count**: Humans observe images and provide reliable comparisons; the model processes corrupted inputs. Scaling $s^{\*}$ alone preserves relative proportions but cannot determine absolute counts.
>
> -  **Delivery Platform Workload (Line 36)**: "All tasks easy" vs. "all tasks hard" yield identical rankings but require different compensation. Scaling $s^{\*}$ cannot distinguish whether the workload justifies 10 or 100 USD subsidies; only $\hat{s}$ integrates the model's absolute scale with human ordinal structure.
>
> ---
>
> ### Q2: If $s^{\*}$ and $\hat{s}$ induce the same order, shouldn't they yield identical Kendall's $\tau$?
>
> The key is that $\hat{s}$ often contains **ties** introduced by the Pool-Adjacent-Violators (PAVA) algorithm, which are absent in $s^{\*}$. When $s_p$ violates the consensus ranking, PAVA resolves conflicts by averaging violating scores, creating ties. This improves $\tau$ by converting discordant pairs into tied pairs, which are penalized less severely. For more relevant discussion, refer to [1].
>
> Here is an example:
>
> - Ground truth: $s = [10, 20, 30, 40]$ (order: A < B < C < D)
> - Stage-1: $s^{\*} = [1, 2, 5, 4]$ induces A < B < D < C (C,D inverted)
> - Model: $s_p = [12, 22, 29, 31]$ (correct order)
> - PAVA averages C,D: $\hat{s} = [12, 22, 30, 30]$
> - Result: $\tau(s^{\*}, s) = (5-1)/6 \approx 0.67$, $\tau(\hat{s}, s) = (5-0)/6 \approx 0.83$. Not equal.
>
>
> [1] Rank aggregation with ties: experiments and analysis. Proc. VLDB Endow. 8, 11 (July 2015), 1202–1213.
>
> ---
>
> ### Q3: Does standardization inflate distribution divergence between $s^{\*}$ and $\hat{s}$? Would matching mean/variance make distributions more comparable?
>
> Excellent suggestion. We added experiments scaling $s^{\*}$ to match the range of the ground truth $s$. The results show that scaled $s^{\*}$ substantially improves performance over $s^{\*}$, confirming that scale inconsistency is a major issue. However, $\hat{s}$ still outperforms $s^{\*}_{scaled}$, demonstrating that isotonic calibration provides value beyond simple rescaling. This validates that projection onto $\hat{\mathcal{M}}$'s surface yields superior calibration.
>
> | Algorithm | Wasserstein$\downarrow$ |  |  |  | KS $\downarrow$|  |  |  |
> |-----------|----------|----------|--------|-------|----------|----------|--------|-------|
> |           | **$s^\*_{scaled}$** | $s^\*$ | $s_p$ | $\hat{s}$ | **$s^\*_{scaled}$** | $s^\*$ | $s_p$ | $\hat{s}$ |
> | HRA-G     | 1.44 | 2.25 | 2.83 | **0.84** | 0.31 | 0.50 | 0.30 | **0.16** |
> | HRA-E     | 1.46 | 2.24 | 2.83 | **0.83** | 0.34 | 0.50 | 0.30 | **0.16** |
> | HRA-N     | 1.45 | 2.35 | 2.83 | **0.74** | 0.31 | 0.56 | 0.30 | **0.19** |
> | BTL-MLE   | 1.78 | 2.19 | 2.83 | **0.89** | 0.42 | 0.46 | 0.30 | **0.30** |
> | CrowdBT   | 1.72 | 2.15 | 2.83 | **0.84** | 0.40 | 0.46 | 0.30 | **0.27** |
> | CrowdTCV  | 1.76 | 2.27 | 2.83 | **1.02** | 0.42 | 0.51 | 0.30 | **0.31** |
> | TCV-MLE   | 1.78 | 2.34 | 2.83 | **0.91** | 0.43 | 0.55 | 0.30 | **0.30** |
>
> ---
>
> **If our responses adequately address your concerns, we would greatly appreciate your consideration in raising your scores. Thank you very much.**

---

### Author Response · Authors · 2025-11-25
**Summary of Rebuttal and Revisions**

We thank the reviewers for their valuable feedback and constructive suggestions. We have uploaded the revised manuscript with updates highlighted in $\textcolor[rgb]{1.0,0.13,0.32}{\text{crimson red}}$, and updated corresponding details in the rebuttal to reflect the current version. We hope the following summary can save reviewers' and AC's time for more thorough discussion.

Overall, the reviewers find our work addresses an important and/or well-defined problem in human-centered assessment [Q8gb, Arz1, 2TdT, hr8J], offers a well-motivated design and comprehensive theoretical analysis [Q8gb, Arz1, 2TdT, hr8J], provides clear and accessible writing [Q8gb, Arz1], and demonstrates reasonable experimental design with substantial performance improvements [Arz1, 2TdT, hr8J].

Based on reviewer feedback, we conducted additional experiments:

1. Scale inconsistency analysis [Q8gb]: We scaled $\mathbf{s}^{*}$ to match ground truth range to validate the necessity of isotonic calibration beyond simple rescaling. Details, results, and analysis are in $\textcolor[rgb]{1.0,0.13,0.32}{\text{Appendix E.5 (Page 32,33)}}$.

2. Comprehensive real-world evaluation [hr8J]: We extended the real-world dataset evaluation to include multiple rank aggregation methods (similar to Table 1 in semi-synthetic evaluation) and three human-centered assessment baselines for comparing the overall AtC framework. $\textcolor[rgb]{1.0,0.13,0.32}{\text{Table 2}}$ presents these results, with full description, results, and analysis in $\textcolor[rgb]{1.0,0.13,0.32}{\text{Appendix E.7 (Page 35)}}$.

We also added the following discussions addressing reviewers' suggestions:

1. [2TdT] Clarified the distinction between subjectivity and annotator inconsistency. $\textcolor[rgb]{1.0,0.13,0.32}{\text{(Page 3)}}$

2. [2TdT] Added illustrative example in Figure 1. $\textcolor[rgb]{1.0,0.13,0.32}{\text{(Page 4)}}$

3. [2TdT] Added brief description of Baselines and Metrics in main text with expanded appendix details. $\textcolor[rgb]{1.0,0.13,0.32}{\text{(Page 7,8)}}$

4. [Arz1, 2TdT, hr8J] Moved Related Work section from appendix to main text and explained benefits of $L_2$ projection. $\textcolor[rgb]{1.0,0.13,0.32}{\text{(Page 10)}}$

5. [2TdT, hr8J] Provided additional details on the real-world dataset and experimental setup. $\textcolor[rgb]{1.0,0.13,0.32}{\text{(Page 30)}}$

6. [Q8gb] Explained how PAVA's tie-creation mechanism leads to different Kendall's $\tau$ values between $\mathbf{s}^{*}$ and $\hat{\mathbf{s}}$. $\textcolor[rgb]{1.0,0.13,0.32}{\text{(Page 32)}}$

Please see our point-by-point responses to all reviewer comments below.

---

### Author Response · Authors · 2025-12-02
**Summary for AC**

Dear AC,

Below, we present a summary of our contributions and responses to the reviewers' concerns.

## Our Contribution
We introduce AtC, a framework for human-centered assessment, which provides:
- Conceptual Novelty: We formalize the problem of "human-centered assessment" and propose a general framework for human-AI complementarity.
- Theoretical Guarantees: We provide a comprehensive analysis with three key results: (Thm 3.3) The first formal proof that modeling annotator heterogeneity is strictly more efficient; (Thm 3.5) Novel risk bounds for isotonic regression; and (Thm 3.9) An optimality guarantee.
- Empirical Verification: Through experiments on semi-synthetic and real-world data, we demonstrate the superiority of AtC. Our work on the dots-activity dataset constitutes the first open-source study in this field.

## Reviewers' Recognition
Reviewers recognized the novelty, theoretical soundness, empirical validation, and clarity of our work:
- Novel, well-motivated, and important.
    - [Arz1] “The technique is well-motivated and rather simple”.
    - [2TdT] “The problem definition is novel, clear, and covers two aspects that heavily interact with each other but has (to the best of my knowledge) been rather understudied […], which makes it an important task type to look at”.
    - [hr8J] “The paper focuses on an important research question—how to calibrate model's score for human-centered assessment tasks.”
- Strong theoretical guarantees.
    - [Q8gb] “AtC is a lightweight approach with theoretical guarantees”.
    - [Arz1] “The paper analysis the technique on theoretical and empirical grounds”.
    - [2TdT] “The authors provide theoretical guarantees for the framework”.
    - [hr8J] “The theoretical analysis is sound under the modeling assumptions”.
- Comprehensive experiments.
    - [Arz1] “The experiments cover all claims (incl. analysis of the theoretical claims)”.
    - [2TdT] “The experiments also cover both a (semi-)synthetic setting and real-world setting”.
    - [hr8J] “The framework performs consistently across multiple datasets, demonstrating its sound performance”.
- Well-written.
    - [Q8gb] “The paper is generally well written and easy to follow”.
    - [Arz1] “The paper is well-written and I found it rather clear”.
    - [2TdT] “I also like the evaluation setup which is clearly mapped out with the 6 research questions. This helps the reader in getting a direct overview […]”.


## Responses to reviewers' concerns
We have addressed **all** weaknesses/questions for each reviewer.

-  **[Q8gb Q1, Q2, Q4] The Necessity of the Calibrated Score $\hat s$ beyond $s^\*$.**

First, we clarified that $\hat s$ and $s^\*$ lie on **fundamentally different** regions of the constraint set. Rankings alone cannot support decision-making tasks that require absolute values. Second, we explained that PAVA introduces ties when resolving violations, reducing penalties from discordant pairs. This mechanism changes how $\tau$ is computed. Third, we added a numerical example and incorporated the explanation into the revision.

- **[Q8gb Q3] Scale Inconsistency between $\hat s$ and $s^\*$.**

We added experiments where model scores $s^\*$ were scaled to match the ground-truth range. The results confirm that rescaling alone improves model-only performance, but **AtC still achieves the best accuracy**, demonstrating that isotonic projection provides benefits beyond simple normalization.

- **[Arz1 Q] Choice of Euclidean Projection.**

We clarified that: (1) $L\_2$ projection aligns with MLE under Gaussian noise; (2) strict convexity ensures a **unique** calibrated score; and (3) PAVA guarantees **optimal linear-time computation**.

- **[2TdT W1] Subjectivity and Annotator Inconsistency.**

We clarified the distinction between subjectivity (true, stable inter-person variability) and annotator inconsistency (noise that aggregation should correct), and revised the introduction to formalize how AtC handles these components differently within human-centered assessment.

- **[2TdT W2, W3, W4, W5; hr8J W1] Figure, Related work, and Metric.**

We have added an enhanced Figure 1, moved related work to the main pages, and provided clearer metric motivation in the main text.

- **[hr8J W2] Clarification of Modeling Assumptions.**

We summarized our assumptions at three levels (conceptual, modeling, and theoretical) and illustrated that these assumptions are **standard and appropriate** for our setting.

- **[hr8J W3, Q] Validation Through Real Human Studies.**

We expanded the description of the real-world Dots dataset. This demonstrates that AtC meaningfully improves score estimation in settings grounded in real human judgments.

- **[hr8J W4] Additional Experiments.**

We expanded the real-world dataset evaluation to include: (1) multiple rank aggregation baselines; (2) three additional human-centered assessment baselines. The results show that **AtC consistently outperforms $s\_p$ and $s^\*$ across all baselines.**

---

### Meta-Review · Area_Chair_x55a · 2025-12-19

**Summary:**

The paper develops a framework for decision making with human/AI interaction. The approach combines human rankings with an ML model's prediction. The procedure comes with theoretical guarantees, and is also evaluated experimentally.

I found the paper to be very interesting. The tackled problem is highly relevant, and developing approaches for combined human/AI decision making is very timely and relevant. I am impressed at the technical rigor of the approach, and also the comprehensive empirical evaluations.

I recommend the paper for acceptance. In their final version, I urge the authors to think more about how the paper can be made more accessible to a broad ICLR audience. The reviewers had some difficulty with the setup, and we say lower than average confidence scores for the paper.

**Reviewer Concerns:**

The main concern was raised by Reviewer Q8gb, who gave a score of 4. The reviewer was concerned about the motivation, and said that the ordering of the scores is preserved from the human judgement to the final rankings. The authors have adequately clarified this, the model's scores are necessary for determining the correct scale. The authors also explained this via examples.

Reviewer 2TdT raised some concerns about writing, these were adequately addressed by the authors.

**Reviewer Scores:**

Reviewer Q8gb could have increased their score. Rest of the scores are already positive.

---

### Decision · Program_Chairs · 2026-01-26

Accept (Poster)